# Last-Iterate Global Convergence of Policy Gradients for Constrained Reinforcement Learning

**Alessandro Montenegro**
Politecnico di Milano, Milan, Italy
`alessandro.montenegro@polimi.it`

**Marco Mussi**
Politecnico di Milano, Milan, Italy
`marco.mussi@polimi.it`

**Matteo Papini**
Politecnico di Milano, Milan, Italy
`matteo.papini@polimi.it`

**Alberto Maria Metelli**
Politecnico di Milano, Milan, Italy
`albertomaria.metelli@polimi.it`

## Abstract

*Constrained Reinforcement Learning* (CRL) tackles sequential decision-making problems where agents are required to achieve goals by maximizing the expected return while meeting domain-specific constraints, which are often formulated as expected costs. In this setting, *policy-based* methods are widely used since they come with several advantages when dealing with continuous-control problems. These methods search in the policy space with an *action-based* or *parameter-based* exploration strategy, depending on whether they learn directly the parameters of a stochastic policy or those of a stochastic hyperpolicy. In this paper, we propose a general framework for addressing CRL problems via *gradient-based primal-dual* algorithms, relying on an alternate ascent/descent scheme with dual-variable regularization. We introduce an exploration-agnostic algorithm, called `C-PG`, which exhibits global last-iterate convergence guarantees under (weak) gradient domination assumptions, improving and generalizing existing results. Then, we design `C-PGAE` and `C-PGPE`, the action-based and the parameter-based versions of C-PG, respectively, and we illustrate how they naturally extend to constraints defined in terms of *risk measures* over the costs, as it is often requested in safety-critical scenarios. Finally, we numerically validate our algorithms on constrained control problems, and compare them with state-of-the-art baselines, demonstrating their effectiveness.

## 1 Introduction

When applying Reinforcement Learning (RL, Sutton and Barto, 2018) to real-world scenarios, we are tasked with addressing large-scale continuous control problems where, in addition to reaching a goal, it is necessary to meet structural or utility-based constraints. For instance, an autonomous-driving car has its main objective of getting to the desired destination (i.e., goal) while avoiding collisions, ensuring the safety of people on the streets, adhering to traffic rules, and respecting the physical requirements of the engine to avoid damaging it (i.e., constraints) (Likmeta et al., 2020). To pursue such an objective, it is necessary to extend the RL problem formulation with the possibility to account for constraints. Constrained Reinforcement Learning (CRL, Uchibe and Doya, 2007) aims at solving this family of problems by employing RL techniques to tackle Constrained Markov Decision Processes (CMDPs, Altman, 1999), which provide an established and widely-used framework for modeling constrained control tasks. The conventional CRL framework primarily focuses on constraints related directly to *expected costs* (Stooke et al., 2020; Ding et al., 2020; Ying

et al., 2022; Ding et al., 2024). However, especially in safety-critical contexts, the expected cost may not represent a reliable index of safe behavior. In response to this issue, *chance constraints* were introduced to ensure that the probability of unsafe events is minimized. Nonetheless, employing chance constraints presents several challenges (Chow et al., 2017). To strike a balance between these two extremes, constraints are defined in terms of *risk measures* over the costs. Examples include the Conditional Value at Risk (CVaR, Rockafellar and Uryasev, 2000) and the Mean-Variance (MV, Markowitz and Todd, 2000; Li and Ng, 2000). These risk measures offer a generalization of the previous concepts, allowing for the consideration of uncertainties while preserving the focus on the cost. When incorporating constraints on risk measures, CRL is often referred to as Risk-CRL (Chow et al., 2017).

Among the RL methods applicable to CMDPs, Policy Gradients (PGs, Deisenroth et al., 2013) are particularly appealing. Indeed, PGs have demonstrably achieved impressive results in continuous-control problems due to several advantages that make them well-suited for real-world applications. These advantages include the ability to handle continuous state and action spaces (Peters and Schaal, 2006), resilience to sensor and actuator noise (Gravell et al., 2020), robustness in partially-observable environments (Azizzadenesheli et al., 2018), and the possibility of incorporating expert knowledge during policy design (Ghavamzadeh and Engel, 2006), which can simplify the learning process and improve the efficacy, safety, and interpretability of the learned policy (Likmeta et al., 2020). PGs can be categorized into two key families depending on the way exploration is carried out in the policy space (Montenegro et al., 2024). Following their taxonomy, we distinguish between the *action-based* and the *parameter-based* exploration paradigms. The former, employed by REINFORCE (Williams, 1992) and GPOMDP (Baxter and Bartlett, 2001), focuses on directly learning the parameters of a parametric stochastic *policy*. The latter, employed by PGPE (Sehnke et al., 2010), is tasked with learning the parameters of a parametric stochastic *hyperpolicy* from which the parameters of the actual policy (often deterministic) are sampled.

*Policy-based CRL* has gained significant popularity in solving constrained control problems (Achiam et al., 2017). Within this field, algorithms are primarily developed using *primal-dual* methods (Chow et al., 2017; Tessler et al., 2019; Ding et al., 2020, 2021; Bai et al., 2022), which can be formulated through *Lagrangian optimization* of the primal (i.e., policy parameters) and dual variable (i.e., Lagrange multipliers). Even though the distinction between the exploration paradigms is well known in the PG methods literature, the current state of the art in Policy-based CRL focuses only on the *action-based* exploration approach (Achiam et al., 2017; Stooke et al., 2020; Bai et al., 2023), while the *parameter-based* one remains unexplored. A critical challenge for policy-based Lagrangian optimization algorithms is ensuring convergence guarantees. Existing works have spent a notable effort in this direction (Ying et al., 2022; Gladin et al., 2023; Ding et al., 2024). Recently, (Ying et al., 2022; Gladin et al., 2023; Ding et al., 2024) manage to ensure *global last-iterate* convergence guarantees. However, these approaches are affected by some notable limitations: ($i$) the provided convergence rates depend on the problem dimension (e.g., the cardinality of the state and action spaces), limiting their applicability to tabular CMPDs and preventing scaling to realistic continuous control problems; ($ii$) they focus on *softmax* policies only, disregarding other more realistic policy models; ($iii$) (Ding et al., 2024) ensure convergence when a single constraint only is present.

**Original Contribution.** The goal of this work is to introduce a framework for solving constrained continuous control problems using policy-based primal-dual algorithms that operate in both the *action-based* and *parameter-based* policy gradient exploration scenarios, while providing global last-iterate convergence guarantees with general (hyper)policy parameterization. Specifically, the main contributions of this work can be summarized as follows:

- In Section 2, we introduce a general constrained optimization problem, which is agnostic w.r.t. both the *action-based* or *parameter-based* paradigm.
- In Section 3, we introduce `C-PG`, a general policy-based primal-dual algorithm optimizing the regularized Lagrangian function associated with the general constrained optimization problem shown in Section 2. We show that, under (weak) domination assumptions, it simultaneously achieves the following: ($i$) *last-iterate* convergence guarantees to a globally optimal feasible policy (i.e., satisfying all constraints); ($ii$) compatibility with CMDPs having *continuous state and action spaces*; ($iii$) the ability to handle *multiple constraints*.
- In Section 4, we introduce `C-PGAE` and `C-PGPE`, the *action-based* and *parameter-based* versions of C-PG, respectively. Both algorithms are designed to also handle constraints on risk measures, thus extending the applicability space of the father algorithm `C-PG`. This is achieved by employing

a parametric *unified risk measure* formulation, for which we show the mapping to several risk measures of the unified one and we present the specific form of all the estimators.

In Section 5, we numerically validate our proposals against state-of-the-art baselines in constrained control problems. Related works are discussed in Appendix B. The proofs of all the statements are reported in Appendix E.

## 2 Preliminaries

**Notation.** For a measurable set $\mathcal{X}$, we denote as $\Delta(\mathcal{X})$ the set of probability measures over $\mathcal{X}$. For $P \in \Delta(\mathcal{X})$, we denote with $p$ its density function and we will interchangeably use $x \sim P$ or $x \sim p$ to express that random variable $x$ is distributed according to $P$. For $n, m \in \mathbb{N}$ with $n \leqslant m$, we denote $[\![n]\!] := \{1, 2, \ldots, n\}$ and with $[\![n, m]\!] := \{n, n+1, \ldots, m\}$. For a vector $\boldsymbol{x} \in \mathbb{R}^d$, we denote as $x_i$ the $i$-th component of $\boldsymbol{x}$. For $a \in \mathbb{R}$, we define $(a)^+ := \max\{0, a\}$ and we extend the notation to vectors as $(\boldsymbol{x})^+ = ((x_1)^+, \ldots, (x_d)^+)^\top$. Given a set $\mathcal{X} \subseteq \mathbb{R}^d$, we denote with $\Pi_{\mathcal{X}}$ the Euclidean norm projection, i.e., $\Pi_{\mathcal{X}} \boldsymbol{x} \in \arg\min_{\boldsymbol{y} \in \mathcal{X}} \|\boldsymbol{y} - \boldsymbol{x}\|_2$ for any $\boldsymbol{x} \in \mathbb{R}^d$. For two vectors $\boldsymbol{x}, \boldsymbol{y} \in \mathbb{R}^d$, we denote with $\langle \boldsymbol{x}, \boldsymbol{y} \rangle$ their inner product. A function $f : \mathbb{R}^d \to \mathbb{R}$ is $L_1$-Lipschitz continuous if $|f(\boldsymbol{x}) - f(\boldsymbol{x}')| \leqslant L_1 \|\boldsymbol{x} - \boldsymbol{x}'\|_2$ and $L_2$-smooth if it is differentiable and $\|\nabla_{\boldsymbol{x}} f(\boldsymbol{x}) - \nabla_{\boldsymbol{x}} f(\boldsymbol{x}')\|_2 \leqslant L_2 \|\boldsymbol{x} - \boldsymbol{x}'\|_2$, for every $\boldsymbol{x}, \boldsymbol{x}' \in \mathbb{R}^d$.

**Constrained Markov Decision Processes.** A Constrained Markov Decision Process (CMDP, Altman, 1999) with $U$ constraints is represented by $\mathcal{M}_\mathcal{C} := \left(\mathcal{S}, \mathcal{A}, p, r, \{c_i\}_{i \in [\![U]\!]}, \{b_i\}_{i \in [\![U]\!]}, \mu_0, \gamma\right)$, where $\mathcal{S} \subseteq \mathbb{R}^{d_\mathcal{S}}$ and $\mathcal{A} \subseteq \mathbb{R}^{d_\mathcal{A}}$ are the measurable state and action spaces, $p : \mathcal{S} \times \mathcal{A} \to \Delta(\mathcal{S})$ is the transition model, where $p(\boldsymbol{s}'|\boldsymbol{s}, \boldsymbol{a})$ is the probability density of getting to state $\boldsymbol{s}' \in \mathcal{S}$ given that action $\boldsymbol{a} \in \mathcal{A}$ is taken in state $\boldsymbol{s} \in \mathcal{S}$, $r : \mathcal{S} \times \mathcal{A} \to [-1, 0]$ is the reward function, where $r(\boldsymbol{s}, \boldsymbol{a})$ is the instantaneous reward obtained by playing action $\boldsymbol{a}$ in state $\boldsymbol{s}$, $c_i : \mathcal{S} \times \mathcal{A} \to [0, 1]$ is the $i$-th cost function, where $c_i(\boldsymbol{s}, \boldsymbol{a})$ is the $i$-th instantaneous cost obtained by playing action $\boldsymbol{a}$ in state $\boldsymbol{s}$, $b_i \in [0, J_{\max}]$ with $J_{\max} := \frac{1 - \gamma^T}{1 - \gamma}$ is the threshold for the $i$-th cost for every $i \in [\![U]\!]$, $\mu_0 \in \Delta(\mathcal{S})$ is the initial state distribution, and $\gamma \in [0, 1]$ is the discount factor. A trajectory $\tau$ of length $T \in \mathbb{N} \cup \{+\infty\}$ is a sequence of $T$ state-action pairs: $\tau = (\boldsymbol{s}_{\tau,0}, \boldsymbol{a}_{\tau,0}, \ldots, \boldsymbol{s}_{\tau,T-1}, \boldsymbol{a}_{\tau,T-1})$. The *discounted return* over a trajectory $\tau$ is $R(\tau) := \sum_{t=0}^{T-1} \gamma^t r(\boldsymbol{s}_{\tau,t}, \boldsymbol{a}_{\tau,t})$, while the $i$-th *discounted cumulative cost* is $C_i(\tau) := \sum_{t=0}^{T-1} \gamma^t c_i(\boldsymbol{s}_{\tau,t}, \boldsymbol{a}_{\tau,t})$. We define the additional cost function $c_0(\boldsymbol{s}, \boldsymbol{a}) := -r(\boldsymbol{s}, \boldsymbol{a})$ and $C_0(\tau) := -R(\tau)$. Note that $R(\tau), C_i(\tau) \in [0, J_{\max}]$ for every $i \in [\![U]\!]$ and trajectory $\tau$.

**Action-based Policy Gradients.** Action-based (AB) PG methods focus on learning the parameters $\boldsymbol{\theta} \in \Theta \subseteq \mathbb{R}^{d_\Theta}$ of a parametric stochastic policy $\pi_{\boldsymbol{\theta}} : \mathcal{S} \to \Delta(\mathcal{A})$, where $\pi_{\boldsymbol{\theta}}(\boldsymbol{a}|\boldsymbol{s})$ represents the probability density of selecting action $\boldsymbol{a} \in \mathcal{A}$ being in state $\boldsymbol{s} \in \mathcal{S}$. At each step $t$ of the interaction with the environment, the stochastic policy is employed to sample an action $\boldsymbol{a}_t \sim \pi_{\boldsymbol{\theta}_t}(\cdot|\boldsymbol{s}_t)$. To assess the performance of $\pi_{\boldsymbol{\theta}}$ w.r.t. the $i$-th cost function, with $i \in [\![0, U]\!]$, we employ the *AB performance index* $J_{\mathrm{A},i} : \Theta \to \mathbb{R}$, which is defined as $J_{\mathrm{A},i}(\boldsymbol{\theta}) := \mathbb{E}_{\tau \sim p_{\mathrm{A}}(\cdot|\boldsymbol{\theta})}[C_i(\tau)]$, where $p_{\mathrm{A}}(\tau, \boldsymbol{\theta}) := \mu_0(\boldsymbol{s}_{\tau,0}) \prod_{t=0}^{T-1} \pi_{\boldsymbol{\theta}}(\boldsymbol{a}_{\tau,t}|\boldsymbol{s}_{\tau,t}) p(\boldsymbol{s}_{\tau,t+1}|\boldsymbol{s}_{\tau,t}, \boldsymbol{a}_{\tau,t})$ is the density of trajectory $\tau$ induced by policy $\pi_{\boldsymbol{\theta}}$.

**Parameter-based Policy Gradients.** Parameter-based (PB) PG methods focus on learning the parameters $\boldsymbol{\rho} \in \mathcal{R} \subseteq \mathbb{R}^{d_\mathcal{R}}$ of a parametric stochastic hyperpolicy $\nu_{\boldsymbol{\rho}} \in \Delta(\Theta)$. The hyperpolicy $\nu_{\boldsymbol{\rho}}$ is used to sample parameter configurations $\boldsymbol{\theta} \sim \nu_{\boldsymbol{\rho}}$ to be plugged into an underlying parametric policy $\pi_{\boldsymbol{\theta}}$, that will be then used for the interaction with the environment. Notice that $\pi_{\boldsymbol{\theta}}$ can also be deterministic. To assess the performance of $\nu_{\boldsymbol{\rho}}$ w.r.t. the $i$-th cost function, with $i \in [\![0, U]\!]$, we employ the *PB performance index* $J_{\mathrm{P},i} : \mathcal{R} \to \mathbb{R}$, which is defined as $J_{\mathrm{P},i}(\boldsymbol{\rho}) := \mathbb{E}_{\boldsymbol{\theta} \sim \nu_{\boldsymbol{\rho}}}[\mathbb{E}_{\tau \sim p_{\mathrm{A}}(\cdot|\boldsymbol{\theta})}[C_i(\tau)]]$.

**Constrained Optimization Problem.** Having introduced the AB and PB performance indices, we formulate a *constrained optimization problem* (COP) *agnostic* w.r.t. the exploration paradigm:

$$\min_{\boldsymbol{v} \in \mathcal{V}} J_{\dagger,0}(\boldsymbol{v}) \quad \text{s.t.} \quad J_{\dagger,i}(\boldsymbol{v}) \leqslant b_i, \quad \forall i \in [\![U]\!], \tag{1}$$

where $\dagger \in \{\mathrm{A}, \mathrm{P}\}$ and $\boldsymbol{v}$ is a generic parameter vector belonging to the parameter space $\mathcal{V}$. When $\dagger = \mathrm{A}$, we are considering the AB exploration paradigm, then $\mathcal{V} = \Theta$. On the other hand, when $\dagger = \mathrm{P}$, we are in the PB exploration paradigm, then $\mathcal{V} = \mathcal{R}$.

# 3 Last-Iterate Global Convergence of `C-PG`

In this section, we present `C-PG`, a general primal-dual algorithm that optimizes a regularized version of the Lagrangian function (Section 3.1) associated with the COP of Equation (1). After having introduced the necessary assumptions (Section 3.2), we show that `C-PG` exhibit *dimension-free*[1] *last-iterate global* convergence guarantees (Section 3.3). While `C-PG` is designed to be agnostic w.r.t. the exploration approach, we introduce two specific versions of `C-PG` in Section 4 for AB or PB, respectively. For notation convenience, in the rest of this section, we use $J_i$ in place of $J_{\dagger,i}$.

## 3.1 Regularized Lagrangian Approach

To solve the COP of Equation (1) we resort to the method of Lagrange multipliers (Bertsekas, 2014) introducing the Lagrangian function $\mathcal{L}_0(\boldsymbol{v}, \boldsymbol{\lambda}) := J_0(\boldsymbol{v}) + \sum_{i=1}^{U} \lambda_i (J_i(\boldsymbol{v}) - b_i) = J_0(\boldsymbol{v}) + \langle \boldsymbol{\lambda}, \mathbf{J}(\boldsymbol{v}) - \mathbf{b} \rangle$, where $\boldsymbol{v} \in \mathcal{V}$ is the primal variable and $\boldsymbol{\lambda} \in \mathbb{R}^{U}_{\geqslant 0}$ are the Lagrangian multipliers or dual variable, $\mathbf{J} = (J_1, \ldots, J_U)^\top$, and $\mathbf{b} = (b_1, \ldots, b_U)^\top$. This allows to rephrase the COP in Equation (1) as a min-max optimization problem $\min_{\boldsymbol{v} \in \mathcal{V}} \max_{\boldsymbol{\lambda} \in \mathbb{R}^{U}_{\geqslant 0}} \mathcal{L}_0(\boldsymbol{v}, \boldsymbol{\lambda})$ and we denote with $H_0(\boldsymbol{v}) := \max_{\boldsymbol{\lambda} \in \mathbb{R}^{U}_{\geqslant 0}} \mathcal{L}_0(\boldsymbol{v}, \boldsymbol{\lambda})$ the *primal function* and with $H_0^* := \min_{\boldsymbol{v} \in \mathcal{V}} H_0(\boldsymbol{v})$. To obtain a *last-iterate* convergence guarantee, we make use of a regularization approach. Specifically, let $\omega > 0$ be a regularization parameter, we define the $\omega$-*regularized Lagrangian function* as follows:

$$\mathcal{L}_\omega(\boldsymbol{v}, \boldsymbol{\lambda}) := J_0(\boldsymbol{v}) + \sum_{i=1}^{U} \lambda_i (J_i(\boldsymbol{v}) - b_i) - \frac{\omega}{2} \|\boldsymbol{\lambda}\|_2^2 = J_0(\boldsymbol{v}) + \langle \boldsymbol{\lambda}, \mathbf{J}(\boldsymbol{v}) - \mathbf{b} \rangle - \frac{\omega}{2} \|\boldsymbol{\lambda}\|_2^2. \quad (2)$$

The ridge regularization makes $\mathcal{L}_\omega(\boldsymbol{v}, \boldsymbol{\lambda})$ a strongly concave function of $\boldsymbol{\lambda}$ at the price of a bias that is quantified in Lemmas E.1, E.2 and E.3. Thus, we address the $\omega$-regularized min-max optimization problem $\min_{\boldsymbol{v} \in \mathcal{V}} \max_{\boldsymbol{\lambda} \in \Lambda} \mathcal{L}_\omega(\boldsymbol{v}, \boldsymbol{\lambda})$, where $\Lambda := \{\boldsymbol{\lambda} \in \mathbb{R}^{U}_{\geqslant 0} : \|\boldsymbol{\lambda}\|_2 \leqslant \omega^{-1} \sqrt{U} J_{\max}\}$, in replacement of the original (non-regularized) one. For this problem, we introduce the primal function $H_\omega(\boldsymbol{v}) := \max_{\boldsymbol{\lambda} \in \Lambda} \mathcal{L}_\omega(\boldsymbol{v}, \boldsymbol{\lambda})$, that, thanks to the ridge regularization, admits the closed-form expression $H_\omega(\boldsymbol{v}) = J_0(\boldsymbol{v}) + \frac{1}{2\omega} \sum_{i=1}^{U}((J_i(\boldsymbol{v}) - b_i)^+)^2 = J_0(\boldsymbol{v}) + \frac{1}{2\omega}\|(\mathbf{J}(\boldsymbol{v}) - \mathbf{b})^+\|_2^2$, where the optimal values of the Lagrange multipliers is given by $\boldsymbol{\lambda}^*(\boldsymbol{v}) = \Pi_\Lambda\left(\frac{1}{\omega}(\mathbf{J}(\boldsymbol{v}) - \mathbf{b})\right) = \frac{1}{\omega}(\mathbf{J}(\boldsymbol{v}) - \mathbf{b})^+$ that is guaranteed to have norm smaller than $\omega^{-1}\sqrt{U} J_{\max}$. Furthermore, we define $H_\omega^* := \min_{\boldsymbol{v} \in \mathcal{V}} H_\omega(\boldsymbol{v})$. `C-PG` updates the parameters $(\boldsymbol{v}_k, \boldsymbol{\lambda}_k)$ with an *alternate gradient descent-ascent* scheme for every $k \in \mathbb{N}$:

**Primal Update**     **Dual Update**
$$\boldsymbol{v}_{k+1} \leftarrow \Pi_\mathcal{V}\left(\boldsymbol{v}_k - \zeta_{\boldsymbol{v},k} \widehat{\nabla}_{\boldsymbol{v}} \mathcal{L}_\omega(\boldsymbol{v}_k, \boldsymbol{\lambda}_k)\right) \quad \boldsymbol{\lambda}_{k+1} \leftarrow \Pi_\Lambda\left(\boldsymbol{\lambda}_k + \zeta_{\boldsymbol{\lambda},k} \widehat{\nabla}_{\boldsymbol{\lambda}} \mathcal{L}_\omega(\boldsymbol{v}_{k+1}, \boldsymbol{\lambda}_k)\right),$$

where $\zeta_{\boldsymbol{v},k}, \zeta_{\boldsymbol{\lambda},k} > 0$ are the learning rates and $\widehat{\nabla}_{\boldsymbol{v}} \mathcal{L}_\omega(\boldsymbol{v}_k, \boldsymbol{\lambda}_k), \widehat{\nabla}_{\boldsymbol{\lambda}} \mathcal{L}_\omega(\boldsymbol{v}_k, \boldsymbol{\lambda}_k)$ are estimators of the gradients $\nabla_{\boldsymbol{v}} \mathcal{L}_\omega(\boldsymbol{v}_k, \boldsymbol{\lambda}_k), \nabla_{\boldsymbol{\lambda}} \mathcal{L}_\omega(\boldsymbol{v}_k, \boldsymbol{\lambda}_k)$ of the regularized Lagrangian function.

## 3.2 Assumptions

Before diving into the study of the convergence guarantees of `C-PG`, in this part, we list and motivate the assumptions necessary for our analysis.

**Assumption 3.1** (Existence of Saddle Points). *There exist $\boldsymbol{v}_0^* \in \mathcal{V}$ and $\boldsymbol{\lambda}_0^* \in \mathbb{R}^{U}_{\geqslant 0}$ such that $\mathcal{L}_0(\boldsymbol{v}_0^*, \boldsymbol{\lambda}_0^*) = \min_{\boldsymbol{v} \in \mathcal{V}} \max_{\boldsymbol{\lambda} \in \mathbb{R}^{U}_{\geqslant 0}} \mathcal{L}_0(\boldsymbol{v}, \boldsymbol{\lambda})$.*

Assumption 3.1 ensures that the value of the min-max problem is attained by a pair of primal-dual values $\boldsymbol{v}_0^* \in \mathcal{V}$ and $\boldsymbol{\lambda}_0^* \in \mathbb{R}^{U}_{\geqslant 0}$ which, consequently, satisfy $\mathcal{L}_0(\boldsymbol{v}_0^*, \boldsymbol{\lambda}) \leqslant \mathcal{L}_0(\boldsymbol{v}_0^*, \boldsymbol{\lambda}_0^*) \leqslant \mathcal{L}_0(\boldsymbol{v}, \boldsymbol{\lambda}_0^*)$ for every $\boldsymbol{v} \in \mathcal{V}$ and $\boldsymbol{\lambda} \in \mathbb{R}^{U}_{\geqslant 0}$. Analogous assumptions have been considered by Yang et al. (2020) and Ying et al. (2022). Thus, $(\boldsymbol{v}_0^*, \boldsymbol{\lambda}_0^*)$ is a saddle point of the Lagrangian function $\mathcal{L}_0$ and, consequently, *strong duality* holds. Alternatively, as commonly requested in CRL works, assuming

---

[1]The *dimension-free* property (Liu et al., 2021; Ding et al., 2020, 2022, 2024) is achieved when the convergence rates do not depend on the cardinality of the state and/or action spaces.

*Slater's condition* combined with the requirement that the policy space covers all Markovian policies ensures strong duality (e.g., Paternain et al., 2019; Ding et al., 2020, 2024).[2]

**Assumption 3.2** (Weak $\psi$-Gradient Domination). *Let $\psi \in [1, 2]$. There exist $\alpha_1 > 0$ and $\beta_1 \geqslant 0$ such that, for every $\boldsymbol{v} \in \mathcal{V}$ and $\boldsymbol{\lambda} \in \Lambda$, it holds that:*

$$\|\nabla_{\boldsymbol{v}} \mathcal{L}_0(\boldsymbol{v}, \boldsymbol{\lambda})\|_2^{\psi} \geqslant \alpha_1 \Big( \mathcal{L}_0(\boldsymbol{v}, \boldsymbol{\lambda}) - \min_{\boldsymbol{v}' \in \mathcal{V}} \mathcal{L}_0(\boldsymbol{v}', \boldsymbol{\lambda}) \Big) - \beta_1. \tag{3}$$

Assumption 3.2 is customary in the convergence analysis of policy gradient methods and it is usually enforced on the objective $J_0$ only (Yuan et al., 2022; Masiha et al., 2022; Fatkhullin et al., 2023). In particular, when $\beta_1 = 0$, we speak of (strong) $\psi$-gradient domination. In this form, for a generic exponent $\psi \in [1, 2]$, this assumption has been employed by Masiha et al. (2022). Particular cases are when $\psi = 1$, which corresponds to the standard (weak) *gradient domination* (GD), while for $\psi = 2$, we have the so-called *Polyak-Łojasiewicz* (PL) condition. Notice that Assumption 3.2 is enforced on the non-regularized Lagrangian function $\mathcal{L}_0$ (i.e., $\omega = 0$). However, it is easy to realize that it holds for the regularized one $\mathcal{L}_{\omega}$ by simply computing the terms of Equation (3) replacing $\mathcal{L}_0$ with $\mathcal{L}_{\omega}$.

**Remark 3.1** (When does Assumption 3.2 holds?). *As remarked by Ding et al. (2024), the Lagrangian function, for a fixed value of $\boldsymbol{\lambda}$, can be regarded as the expected return of a new reward function $-C_0 - \langle \boldsymbol{\lambda}, \mathbf{C} \rangle$, where $\mathbf{C} = (C_1, \ldots, C_U)^{\top}$. As a consequence, a sufficient condition for Assumption 3.2 is when the selected class of policies guarantees the $\psi$-gradient domination regardless of the reward function. For instance, in tabular environments with natural policy parametrization, i.e., $\pi_{\boldsymbol{\theta}}(s) = \boldsymbol{\theta}_s$ for every $s \in \mathcal{S}$, the PL condition ($\psi = 2$ and $\beta_1 = 0$) holds (Bhandari and Russo, 2024). Moreover, in tabular environments with softmax policy, i.e., $\pi_{\boldsymbol{\theta}}(a|s) \propto \exp(\theta(s, a))$, GD ($\psi = 1$ and $\beta_1 = 0$) holds (Mei et al., 2020). This enables a meaningful comparison of our results with resorting to softmax policies (e.g., Ding et al., 2020; Gladin et al., 2023; Ding et al., 2024). More in general, when (i) the Fisher information matrix induced by policy $\pi_{\boldsymbol{\theta}}$ is non-degenerate for every $\boldsymbol{\theta} \in \Theta$, i.e., $\mathbf{F}(\boldsymbol{\theta}) = \mathbb{E}_{\pi_{\boldsymbol{\theta}}}[\nabla_{\boldsymbol{\theta}} \log \pi_{\boldsymbol{\theta}}(\boldsymbol{a}|\boldsymbol{s}) \nabla_{\boldsymbol{\theta}} \log \pi_{\boldsymbol{\theta}}(\boldsymbol{a}|\boldsymbol{s})^{\top}] \succeq \mu_F \mathbf{I}$ for some $\mu_F > 0$ and (ii) a compatible function approximation bias bound holds, i.e., $\mathbb{E}_{\pi_{\boldsymbol{\theta}^*}}[(A^{\pi_{\boldsymbol{\theta}}}(\boldsymbol{s}, \boldsymbol{a}) - (1 - \gamma)\boldsymbol{u}^{\top} \nabla_{\boldsymbol{\theta}} \log \pi_{\boldsymbol{\theta}}(\boldsymbol{a}|\boldsymbol{s}))^2] \leqslant \epsilon_{bias}$ being $\boldsymbol{u} = \mathbf{F}(\boldsymbol{\theta})^{\dagger} \nabla_{\boldsymbol{\theta}} J_0(\boldsymbol{\theta})$ and the advantage function $A^{\pi_{\boldsymbol{\theta}}}$ computed w.r.t. reward $-C_0 - \langle \boldsymbol{\lambda}, \mathbf{C} \rangle$, the weak GD ($\psi = 1$) holds with $\alpha_1 = G\mu_F^{-1}$ and $\beta_1 = (1 - \gamma)^{-1}\sqrt{\epsilon_{bias}}$, where $G$ is such that $\|\nabla_{\boldsymbol{\theta}} \log \pi_{\boldsymbol{\theta}}(\boldsymbol{a}|\boldsymbol{s})\|_2 \leqslant G$ (Masiha et al., 2022).*

In principle, we could have enforced Assumption 3.2 on the primal function $H_{\omega}(\boldsymbol{v})$ only. However, this would come with two drawbacks: (i) the assumption would now depend explicitly on $\omega$; (ii) the considerations of Remark 3.1 would no longer hold. Nevertheless, in Lemma E.4, we prove that Assumption 3.2 induces an analogous property on the primal function $H_{\omega}(\boldsymbol{v})$ in the regularized case.

**Assumption 3.3** (Regularity of the Regularized Lagrangian $\mathcal{L}_{\omega}$). *There exists $L_1, L_2, L_3 > 0$ such that, for every $\boldsymbol{v}, \boldsymbol{v}' \in \mathcal{V}$, and for every $\boldsymbol{\lambda}, \boldsymbol{\lambda}' \in \Lambda$, the following hold:*

$$\nabla_{\boldsymbol{\lambda}} \mathcal{L}_0(\cdot, \boldsymbol{\lambda}) \ L_1\text{-Lipschitz w.r.t. } \boldsymbol{v}: \qquad \big\|\nabla_{\boldsymbol{\lambda}} \mathcal{L}_0(\boldsymbol{v}, \boldsymbol{\lambda}) - \nabla_{\boldsymbol{\lambda}} \mathcal{L}_0(\boldsymbol{v}', \boldsymbol{\lambda})\big\|_2 \leqslant L_1 \|\boldsymbol{v} - \boldsymbol{v}'\|_2, \qquad (4)$$

$$\mathcal{L}_0(\cdot, \boldsymbol{\lambda}) \ L_2\text{-Smooth w.r.t. } \boldsymbol{v}: \qquad \big\|\nabla_{\boldsymbol{v}} \mathcal{L}_0(\boldsymbol{v}, \boldsymbol{\lambda}) - \nabla_{\boldsymbol{v}} \mathcal{L}_0(\boldsymbol{v}', \boldsymbol{\lambda})\big\|_2 \leqslant L_2 \|\boldsymbol{v} - \boldsymbol{v}'\|_2, \qquad (5)$$

$$\nabla_{\boldsymbol{v}} \mathcal{L}_0(\boldsymbol{v}, \cdot) \ L_3\text{-Lipschitz w.r.t } \boldsymbol{\lambda}: \qquad \big\|\nabla_{\boldsymbol{v}} \mathcal{L}_0(\boldsymbol{v}, \boldsymbol{\lambda}) - \nabla_{\boldsymbol{v}} \mathcal{L}_0(\boldsymbol{v}, \boldsymbol{\lambda}')\big\|_2 \leqslant L_3 \|\boldsymbol{\lambda} - \boldsymbol{\lambda}'\|_2. \qquad (6)$$

Notice that, similarly to Assumption 3.2, we realize that if Assumption 3.3 holds for the non-regularized Lagrangian $\mathcal{L}_0$, it also holds (with the same constants) for the regularized one $\mathcal{L}_{\omega}$ for every $\omega > 0$. The regularity conditions of Assumption 3.3 are common in the literature (Yang et al., 2020) and mild when regarded from the policy optimization perspective. Equation (4) is satisfied whenever the constraint functions $J_i$ are Lipschitz continuous w.r.t. $\boldsymbol{v}$. Indeed, $\|\nabla_{\boldsymbol{\lambda}} \mathcal{L}_0(\boldsymbol{v}, \boldsymbol{\lambda}) - \nabla_{\boldsymbol{\lambda}} \mathcal{L}_0(\boldsymbol{v}', \boldsymbol{\lambda})\|_2 = \|\mathbf{J}(\boldsymbol{v}) - \mathbf{J}(\boldsymbol{v}')\|_2$. Equation (5) is fulfilled when the objective function $J_0$ and the constraint functions $J_i$ are smooth w.r.t. $\boldsymbol{v}$ and the Lagrange multipliers are bounded (guaranteed thanks to the projection $\Pi_{\Lambda}$), since $\|\nabla_{\boldsymbol{v}} \mathcal{L}_0(\boldsymbol{v}, \boldsymbol{\lambda}) - \nabla_{\boldsymbol{v}} \mathcal{L}_0(\boldsymbol{v}', \boldsymbol{\lambda})\|_2 \leqslant |\nabla_{\boldsymbol{v}} J_0(\boldsymbol{v}) - \nabla_{\boldsymbol{v}} J_0(\boldsymbol{v}')| + \sum_{i=1}^{U} \lambda_i |\nabla_{\boldsymbol{v}} J_i(\boldsymbol{v}) - \nabla_{\boldsymbol{v}} J_i(\boldsymbol{v})|$. Finally, Equation (6) is fulfilled whenever functions $J_i$ admit bounded gradients, since $\|\nabla_{\boldsymbol{v}} \mathcal{L}_0(\boldsymbol{v}, \boldsymbol{\lambda}) - \nabla_{\boldsymbol{v}} \mathcal{L}_0(\boldsymbol{v}, \boldsymbol{\lambda}')\|_2 \leqslant \|\nabla_{\boldsymbol{v}} \mathbf{J}(\boldsymbol{v})(\boldsymbol{\lambda} - \boldsymbol{\lambda}')\|_2$. It is worth noting that $L_2$ depends on the norm of the Lagrange multipliers and, consequently, due

---

[2]Assumption 3.1 combined with Slater's condition, i.e, the existence of a parametrization $\widetilde{\boldsymbol{v}} \in \mathcal{V}$ for which there exists $\xi > 0$ such that $J_i(\widetilde{\boldsymbol{v}}) - b < -\xi$ for all $i \in [\![U]\!]$ (strictly feasible), allows providing an upper bound to the Lagrange multipliers $\|\boldsymbol{\lambda}_0^*\|_2 \leqslant \xi^{-1}(J_0(\widetilde{\boldsymbol{v}}) - J_0(\boldsymbol{v}_0^*))$ using standard arguments (see Ying et al., 2022).

to the projection operator, we have that $L_2 = \mathcal{O}(\omega^{-1})$, whereas $L_1$ and $L_3$ are independent on $\omega$.[3] Explicit conditions on the constitutive elements of the MDP and (hyper)policies to ensure Lipshitzness and smoothness of these quantities are reported in (Montenegro et al., 2024, Appendix E) for both the AB and PB cases. These regularity properties enforced on $\mathcal{L}_\omega$ are inherited by the primal function $H_\omega$ which results to be $\left(L_2 + L_1^2 \omega^{-1}\right)$-smooth (Lemma E.7). Concerning the regularity of $\mathcal{L}_\omega$ w.r.t. $\boldsymbol{\lambda}$, we observe that it is a quadratic function and, therefore, it is $\omega$-smooth and satisfies the PL condition, i.e., Assumption 3.2 with $\psi = 2$, $\beta_1 = 0$, and with $\alpha_1 = \omega$ (Lemma E.5).

**Assumption 3.4** (Bounded Estimator Variance). *For every $\boldsymbol{v} \in \mathcal{V}$ and $\boldsymbol{\lambda} \in \Lambda$, the estimators $\widehat{\nabla}_{\boldsymbol{v}} \mathcal{L}_\omega(\boldsymbol{v}, \boldsymbol{\lambda})$ and $\widehat{\nabla}_{\boldsymbol{\lambda}} \mathcal{L}_\omega(\boldsymbol{v}, \boldsymbol{\lambda})$ are unbiased for $\nabla_{\boldsymbol{v}} \mathcal{L}_\omega(\boldsymbol{v}, \boldsymbol{\lambda}) = \nabla_{\boldsymbol{v}} J_0(\boldsymbol{v}) + \sum_{i=1}^U \lambda_i \nabla_{\boldsymbol{v}} J_i(\boldsymbol{v})$ and $\nabla_{\boldsymbol{\lambda}} \mathcal{L}_\omega(\boldsymbol{v}, \boldsymbol{\lambda}) = \mathbf{J}(\boldsymbol{v}) - \mathbf{b} - \omega\boldsymbol{\lambda}$ with bounded variance, i.e., there exist $V_{\boldsymbol{v}}, V_{\boldsymbol{\lambda}} < +\infty$ such that:*

$$\mathbb{V}\mathrm{ar}[\widehat{\nabla}_{\boldsymbol{v}} \mathcal{L}_\omega(\boldsymbol{v}, \boldsymbol{\lambda})] \leqslant V_{\boldsymbol{v}}, \qquad \mathbb{V}\mathrm{ar}[\widehat{\nabla}_{\boldsymbol{\lambda}} \mathcal{L}_\omega(\boldsymbol{v}, \boldsymbol{\lambda})] \leqslant V_{\boldsymbol{\lambda}}. \tag{7}$$

Note that $V_{\boldsymbol{v}}$ typically depends on the Lagrange multipliers and, for standard sample mean estimators, it is of order $V_{\boldsymbol{v}} = \mathcal{O}(\omega^{-2})$ thanks to the projection operator. Contrary, $V_{\boldsymbol{\lambda}}$ is usually not affected by $\omega$ since the term $\omega\boldsymbol{\lambda}$ is not estimated and, thus, it does not impact on the variance of the sample mean estimator. In Section 4, explicit estimators are provided for both the AB and PB cases. The variance of such estimators can be easily controlled by leveraging on the properties of the score function as done in previous works (see Papini et al. 2022 and Montenegro et al. 2024, Appendix E).

### 3.3 Convergence Analysis

We are now ready to attack the convergence analysis of C-PG to the global optimum of the COP of Equation (1). To this end, we study the *potential function* defined as $\mathcal{P}_k(\chi) := a_k + \chi b_k$, where $a_k := \mathbb{E}[H_\omega(\boldsymbol{v}_k) - H_\omega^*]$ and $b_k := \mathbb{E}[H_\omega(\boldsymbol{v}_k) - \mathcal{L}_\omega(\boldsymbol{v}_k, \boldsymbol{\lambda}_k)]$ and $\chi \in (0, 1)$ will be specified later. Since $a_k, b_k \geqslant 0$, intuitively, if $\mathcal{P}_k(\chi) \approx 0$ we have that both $a_k, b_k \approx 0$ and, consequently, convergence is achieved. Let us start relating $\mathcal{P}_k(\chi)$, with the solution of the COP in Equation (1).

**Theorem 3.1** (Objective Function Gap and Constraint Violation). *Let $\epsilon > 0$. Under Assumption 3.1, if $\mathcal{P}_k \leqslant \epsilon$, it holds that:*

$$\mathbb{E}[J_0(\boldsymbol{v}_k) - J_0(\boldsymbol{v}_0^*)] \leqslant \epsilon + \frac{\omega}{2}\|\boldsymbol{\lambda}_0^*\|_2^2, \qquad \mathbb{E}[(J_i(\boldsymbol{v}_k) - b_i)^+] \leqslant 4\epsilon + \omega\|\boldsymbol{\lambda}_0^*\|_2, \quad \forall i \in [\![U]\!]. \tag{8}$$

Theorem 3.1 justifies the study of the potential $\mathcal{P}_k$ as a technical tool to ensure convergence. Indeed, whenever $\mathcal{P}_k \leqslant \epsilon$ both the $(i)$ objective function gap and $(ii)$ the constraint violation scale linearly with $\epsilon$ and with the regularization parameter $\omega$ of the regularized Lagrangian $\mathcal{L}_\omega$ multiplied by the norm of the Lagrange multipliers of the non-regularized problem $\|\boldsymbol{\lambda}_0^*\|_2$, which are finite under Assumption 3.1. This expression also suggests a choice of $\omega = \mathcal{O}(\epsilon)$ to enforce an overall $\epsilon$ error on both quantities. Note that, from Theorem 3.1, it is immediate to employ a *conservative constraint* $(b_i' \approx b_i - 4\epsilon - \omega\|\boldsymbol{\lambda}_0^*\|_2^2)$ to achieve zero constraint violation with no modification of the algorithm.

We are now ready to state the convergence guarantees for the potential function.

**Theorem 3.2** (Convergence of $\mathcal{P}_K$). *Under Assumptions 3.2, 3.3, 3.4, for $\chi < 1/5$, sufficiently small $\epsilon$ and $\omega$, and a choice of constant learning rates $\zeta_{\boldsymbol{v}}, \zeta_{\boldsymbol{\lambda}}$, we have $\mathcal{P}_K(\chi) \leqslant \epsilon + \beta_1/\alpha_1$ whenever:[4]*

- $K = \mathcal{O}(\omega^{-1} \log(\epsilon^{-1}))$ *if $\psi = 2$ and the gradients are exact (i.e., $V_{\boldsymbol{v}} = V_{\boldsymbol{\lambda}} = 0$);*
- $K = \mathcal{O}(\omega^{-1} \epsilon^{-\frac{2}{\psi}-1})$ *if $\psi \in [1, 2)$ and the gradients are exact (i.e., $V_{\boldsymbol{v}} = V_{\boldsymbol{\lambda}} = 0$);*
- $K = \mathcal{O}(\omega^{-3} \epsilon^{-\frac{4}{\psi}+1})$ *if $\psi \in [1, 2]$ and the gradients are estimated (i.e., $V_{\boldsymbol{v}} = \mathcal{O}(\omega^{-2})$ and $V_{\boldsymbol{\lambda}} = \mathcal{O}(1)$).*

Some comments are in order. First, Theorem 3.2 holds for a specific choice of the constant $\chi \in (0, 1/5)$ defining the potential function $\mathcal{P}_K$. Second, the presented rates hold for sufficiently small values of $\epsilon$ and $\omega$. This is just for presentation purposes, as the sample complexity[5] can only improve if we increase the values of $\epsilon$ and $\omega$. Third, in the proof, an explicit expression of the learning rates is provided. Concerning their orders, for the case of exact gradients, we choose $\zeta_{\boldsymbol{\lambda}} = \omega^{-1}$ and $\zeta_{\boldsymbol{v}} = \mathcal{O}(\omega)$, whereas for the estimated gradient case, we choose $\zeta_{\boldsymbol{\lambda}} = \mathcal{O}(\omega\epsilon^{2/\psi})$ and $\zeta_{\boldsymbol{v}} = \mathcal{O}(\omega^3 \epsilon^{2/\psi})$.

---

[3]We highlight the dependences on $\omega$ since, as we shall see later, we will set $\omega = \mathcal{O}(\epsilon)$ having, consequently, an effect on the convergence rate.

[4]In the context of this statement, the $\mathcal{O}(\cdot)$ notation preserves dependences on $\epsilon$ and $\omega$ only.

[5]Theorem 3.2 provides an *iteration-complexity* guarantee. Concerning the *estimated gradient* case, this translates into a *sample complexity* guarantee since we are allowed to estimate gradients with a single sample.

| | Exact Gradients | | | Estimated Gradients | | |
|---|---|---|---|---|---|---|
| | $\psi{=}1$ (GD) | $\psi{\in}(1,2)$ | $\psi{=}2$ (PL) | $\psi{=}1$ (GD) | $\psi{\in}(1,2)$ | $\psi{=}2$ (PL) |
| **Fixed $\omega$** | $\omega^{-1}\epsilon^{-1}$ | $\omega^{-1}\epsilon^{-\frac{2}{\psi}+1}$ | $\omega^{-1}\log(\epsilon^{-1})$ | $\omega^{-3}\epsilon^{-3}\log(\epsilon^{-1})$ | $\omega^{-3}\epsilon^{-\frac{4}{\psi}+1}\log(\epsilon^{-1})$ | $\omega^{-3}\epsilon^{-1}\log(\epsilon^{-1})$ |
| $\omega{=}\mathcal{O}(\epsilon)$ | $\epsilon^{-2}$ | $\epsilon^{-\frac{2}{\psi}}$ | $\epsilon^{-1}\log(\epsilon^{-1})$ | $\epsilon^{-6}\log(\epsilon^{-1})$ | $\epsilon^{-\frac{4}{\psi}-2}\log(\epsilon^{-1})$ | $\epsilon^{-4}\log(\epsilon^{-1})$ |

Table 1: Summary of the sample complexity results of `C-PG` when either keeping $\omega$ fixed or setting it as $\omega = \mathcal{O}(\epsilon)$.

Assuming $\omega$ to be a constant, we observe that both learning rates display the same dependence on $\epsilon$ and, consequently, they are in *single-time scale*. However, as we have seen in Theorem 3.1, in order to obtain guarantees on the original non-regularized problem, we have to set $\omega = \mathcal{O}(\epsilon)$, leading to a *two-time scales* algorithm. Fourth, we observe that, for both exact and estimated gradients, the sample complexity degrades as the constant $\psi$ of the gradient domination moves from 2 to 1, delivering the smallest sample complexity when the PL condition holds. Finally, we highlight that `C-PG` jointly: ($i$) converges to the global optimum of the COP problem of Equation (1); ($ii$) delivers a *last-iterate* guarantee; ($iii$) has no dependence on the cardinality of the state or action spaces, making it completely *dimension-free*. Table 1 and Figure 1 summarize the results of Theorem 3.2.

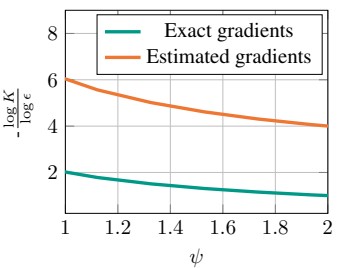

Figure 1: Plot of the exponents of $\epsilon^{-1}$ in the cases of Table 1.

# 4 Action-based and Parameter-based Primal-Dual Algorithms

In this section, we introduce `C-PGAE` and `C-PGPE`, the *action-based* and the *parameter-based* versions of `C-PG`, respectively. Both the algorithms have been designed to tackle *continuous risk-constrained optimization problems* (RCOP), generalizing the COP of Equation (1) and thus extending the applicability space of `C-PG`. This is done by employing a parametric *unified risk measure* formulation, which leads to having a framework for solving risk-constrained problems via policy-based primal-dual methods operating in both the *action-based* and *parameter-based* PGs exploration scenarios. Moreover, this framework allows to handle constraints enforced on several risk measures, which will be discussed below. We first introduce a *parametric unified risk measure* and formulate an exploration-agnostic RCOP (Section 4.1). Then, we present both `C-PGAE` and `C-PGPE` (Section 4.2).

## 4.1 Risk-Constrained Optimization Problem

We start presenting the notion of *unified risk measure* originally introduced by Bisi et al. (2022). For the AB case, to evaluate the performance of policy $\pi_{\boldsymbol{\theta}}$ w.r.t. the $i$-th cost, with $i \in [\![0, U]\!]$, given two functions $f_i : \mathbb{R}^2 \to \mathbb{R}$ and $g_i : \mathbb{R} \to \mathbb{R}$, we define the *AB-risk measure* as:

$$\min_{\eta_i \in \mathbb{R}} \mathcal{J}_{\mathrm{A},i}(\boldsymbol{\theta}, \eta_i) \quad \text{where} \quad \mathcal{J}_{\mathrm{A},i}(\boldsymbol{\theta}, \eta_i) := \underset{\tau \sim p_{\mathrm{A}}(\cdot|\boldsymbol{\theta})}{\mathbb{E}} \left[ f_i(C_i(\tau), \eta_i) \right] + g_i(\eta_i). \tag{9}$$

Similarly, for the PB case, to assess the performance of hyperpolicy $\nu_{\boldsymbol{\rho}}$ w.r.t. the $i$-th cost, with $i \in [\![0, U]\!]$, we define the *PB-risk measure* as:

$$\min_{\eta_i \in \mathbb{R}} \mathcal{J}_{\mathrm{P},i}(\boldsymbol{\rho}, \eta_i) \quad \text{where} \quad \mathcal{J}_{\mathrm{P},i}(\boldsymbol{\rho}, \eta_i) := \underset{\boldsymbol{\theta} \sim \nu_{\boldsymbol{\rho}}}{\mathbb{E}} \left[ \underset{\tau \sim p_{\mathrm{A}}(\cdot|\boldsymbol{\theta})}{\mathbb{E}} \left[ f_i(C_i(\tau), \eta_i) \right] \right] + g_i(\eta_i). \tag{10}$$

Some observations are in order. First, by selecting the functions $f_i$ and $g_i$, we generate different risk measures (Table 2). Details on the presented risk measures and on the mappings can be found in Appendix C. Second, the risk measure itself is defined as another minimization problem over the additional real variable $\eta_i$. In principle, if we replace the constraints on cost expectation of the COP in Equation (1) with the risk measures presented above, we are in the presence of a *bilevel* optimization problem. However, it is immediate to realize that we can merge variables $\eta_i$ with the primal variables $\boldsymbol{\upsilon}$ of the COP without changing the optimum. Finally, let us appreciate the semantic difference between enforcing risk-based constraints in the AB and PB cases. Indeed, while for AB the stochasticity inducing the risk is the one generated by the policy $\pi_{\boldsymbol{\theta}}$ and the environment, for PB we have the joint stochastic process of the hyperpolicy $\nu_{\boldsymbol{\rho}}$, policy $\pi_{\boldsymbol{\theta}}$ (when stochastic), and the

| Risk Measure | Parameter | Need for $\eta$ | $f_i(C_i(\tau), \eta)$ | $g_i(\eta)$ | **AB** GPOMDP-like Estimator |
|:---:|:---:|:---:|:---:|:---:|:---:|
| Expected Cost | - | ✗ | $C_i(\tau)$ | $0$ | Yes |
| Mean Variance | $\kappa$ | ✓ | $(1 - 2\kappa\eta)C_i(\tau) + \kappa C_i(\tau)^2$ | $\kappa\eta^2$ | Partial |
| CVaR$_\alpha$ | $\alpha$ | ✓ | $\frac{1}{1-\alpha}(C_i(\tau) - \eta)^+$ | $\eta$ | No |
| Chance | $n$ | ✗ | $\mathbb{1}\{C_i(\tau) \geqslant n\}$ | $0$ | No |

Table 2: Mapping of the unified risk measure to risk measures.

environment. To the best of our knowledge, this paper is the first proposing to enforce risk-based constraints for *parameter-based* exploration.

Having introduced the notions of AB and PB unified risk measures, we can formulate a *risk-constrained optimization problem* (RCOP) *agnostic* w.r.t. the exploration paradigm and the risk measure:[6]

$$\min_{\boldsymbol{v} \in \mathcal{V}, \boldsymbol{\eta} \in \mathbb{R}^U} \mathcal{J}_{\dagger,0}(\boldsymbol{v}, \eta_0) \quad \text{s.t.} \quad \mathcal{J}_{\dagger,i}(\boldsymbol{v}, \eta_i) \leqslant b_i, \quad \forall i \in [\![U]\!], \tag{11}$$

where $\dagger \in \{A, P\}$, $\boldsymbol{v} \in \mathcal{V}$ is the parameter and $\boldsymbol{\eta} = (\eta_1, \dots, \eta_U) \in \mathbb{R}^U$ are the auxiliary variables.

As for the COP of Equation (1), we define the regularized Lagrangian function for the RCOP (Equation 11) as follows:

$$\widetilde{\mathcal{L}}_{\dagger,\omega}(\boldsymbol{v}, \boldsymbol{\lambda}, \boldsymbol{\eta}) := \mathcal{J}_{\dagger,0}(\boldsymbol{v}, \eta_0) + \sum_{u=1}^{U} \lambda_u \left( \mathcal{J}_{\dagger,u}(\boldsymbol{v}, \eta_u) - b_u \right) - \frac{\omega}{2} \|\boldsymbol{\lambda}\|_2^2, \tag{12}$$

with $\dagger \in \{A, P\}$. We assume that $\widetilde{\mathcal{L}}_{\dagger,\omega}$ is *differentiable* w.r.t. $\boldsymbol{v}$ and *subdifferentiable*[7] w.r.t. $\boldsymbol{\eta}$, while, by construction, it is already differentiable w.r.t. $\boldsymbol{\lambda}$.

**Remark 4.1** (Do the Convergence Guarantees of Section 3 Apply to the RCOP?). *When all the $f_i$ and $g_i$ are selected in order to consider the expected values of the corresponding $C_i$ (i.e., $f_i(C_i(\tau), \eta_i) = C_i(\tau)$ and $g_i(\eta_i) = 0$), then $\widetilde{\mathcal{L}}_{\dagger,\omega}$ coincides with $\mathcal{L}_\omega$, thus, all the theoretical results presented in Section 3 hold. This is not true in general. For some risk measures $\widetilde{\mathcal{L}}_{\dagger,\omega}$ may not be smooth or even differentiable in $\boldsymbol{v}$ or $\boldsymbol{\eta}$, or it may violate the weak gradient domination assumption. Although out of the scope of the present paper, studying the preservation of the (weak) gradient domination for risk-based objectives is an appealing future research direction.*

## 4.2 Algorithms: `C-PGAE` and `C-PGPE`

**Algorithms.** Both algorithms, whose pseudo-codes are deferred to Appendix A, aim at solving the RCOP of Equation (11), finding the best feasible (hyper)policy parameterization. The *alternate ascent/descent* primal-dual prototypical algorithm is the same as described in Section 3. In particular, at each iteration $k \in [\![K]\!]$, the algorithms collect $N$ trajectories (employing the chose exploration paradigm) to update the primal and dual variables as:



**Primal Updates**[7]

$\boldsymbol{v}_{k+1} \leftarrow \boldsymbol{v}_k - \zeta_{\boldsymbol{v},k} \widehat{\nabla}_{\boldsymbol{v}} \widetilde{\mathcal{L}}_{\dagger,\omega}(\boldsymbol{v}_k, \boldsymbol{\lambda}_k, \boldsymbol{\eta}_k)$

$\boldsymbol{\eta}_{k+1} \leftarrow \boldsymbol{\eta}_k - \zeta_{\boldsymbol{\eta},k} \widehat{\nabla}_{\boldsymbol{\eta}} \widetilde{\mathcal{L}}_{\dagger,\omega}(\boldsymbol{v}_k, \boldsymbol{\lambda}_k, \boldsymbol{\eta}_k)$

**Dual Update**

$\boldsymbol{\lambda}_{k+1} \leftarrow \boldsymbol{\lambda}_k + \zeta_{\boldsymbol{\lambda},k} \widehat{\nabla}_{\boldsymbol{\lambda}} \widetilde{\mathcal{L}}_{\dagger,\omega}(\boldsymbol{v}_{k+1}, \boldsymbol{\lambda}_k, \boldsymbol{\eta}_{k+1}),$



where the values are considered to be projected into their spaces, and $\zeta_{\boldsymbol{v},k}, \zeta_{\boldsymbol{\lambda},k}, \zeta_{\boldsymbol{\eta},k} > 0$ are the learning rates at the $k$-th iteration for $\boldsymbol{v}, \boldsymbol{\lambda}$, and $\boldsymbol{\eta}$, respectively. Notice that, since we use *alternate ascent/descent*, the update for $\boldsymbol{\lambda}_{k+1}$ uses the updated primal values $\boldsymbol{v}_{k+1}$ and $\boldsymbol{\eta}_{k+1}$. In practice, this requires collecting a first batch of $N/2$ trajectories using the (hyper)policy $\boldsymbol{v}_k$ to perform the primal updates, computing $\boldsymbol{v}_{k+1}$ and $\boldsymbol{\eta}_{k+1}$. Then, it is necessary to collect a new batch of $N/2$ trajectories using the updated (hyper)policy $\boldsymbol{v}_{k+1}$ in order to perform the dual update, computing $\boldsymbol{\lambda}_{k+1}$, which also leverages $\boldsymbol{\eta}_{k+1}$.

---

[6]For the sake of generality, we admit risk optimization also for the objective function.

[7]As illustrated in Appendix D, some risk measures (e.g., CVaR$_\alpha$) deliver Lagrangian functions that admit subgradients and not gradients w.r.t. $\boldsymbol{\eta}$.

**Estimators.** Here, we provide the risk-agnostic form of the estimators used to perform the primal and dual updates. In particular, the gradient of $\widetilde{\mathcal{L}}_{\dagger,\omega}$ with respect to $\boldsymbol{\eta}$ is closely related to the choices of $f_i$ and $g_i$. Therefore, the discussion is deferred to Appendix D, where we also explicitly derive all the estimators for the risk measures presented in Table 2 and for both exploration paradigms. The estimator of the gradient of $\widetilde{\mathcal{L}}_{\dagger,\omega}$ w.r.t. $\boldsymbol{\lambda}$ has the same form for both C-PGAE and C-PGPE, that is:

$$\widehat{\nabla}_{\boldsymbol{\lambda}}\widetilde{\mathcal{L}}_{\dagger,\omega}(\boldsymbol{\upsilon},\boldsymbol{\lambda},\boldsymbol{\eta}) = \frac{1}{N}\sum_{i=1}^{N}\left(\boldsymbol{f}(\boldsymbol{C}(\tau_i),\boldsymbol{\eta}_{k+1}) - \boldsymbol{g}(\boldsymbol{\eta}_{k+1})\right) - \boldsymbol{b} - \omega\boldsymbol{\lambda}_k, \tag{13}$$

where the trajectories are collected with the AB or the PB exploration paradigm , $\boldsymbol{f}(\boldsymbol{C}(\tau),\boldsymbol{\eta}) \coloneqq (f_1(C_1(\tau),\eta_1),\ldots,f_U(C_U(\tau),\eta_U))^\top$ and $\boldsymbol{g}(\boldsymbol{\eta}) \coloneqq (g_1(\eta_1),\ldots,g_U(\eta_U))^\top$.

C-PGAE (Algorithm 1) is the *action-based* version of C-PG. At each iteration $k \in [\![K]\!]$, the agent collects $N$ trajectories by playing the policy $\pi_{\boldsymbol{\theta}_k}$, then it alternatively updates the policy parameter $\boldsymbol{\theta}$ and the risks parameter $\boldsymbol{\eta}$, or the Lagrange multipliers $\boldsymbol{\lambda}$. The $\boldsymbol{\theta}$ update relies on the estimator:

$$\widehat{\nabla}_{\boldsymbol{\theta}}\widetilde{\mathcal{L}}_{A,\omega}(\boldsymbol{\theta}_k,\boldsymbol{\lambda}_k,\boldsymbol{\eta}_k) \coloneqq \frac{1}{N}\sum_{i=1}^{N}\left(\sum_{t=0}^{T-1}\nabla_{\boldsymbol{\theta}}\log\pi_{\boldsymbol{\theta}_k}(\boldsymbol{a}_{\tau_i,t}|\boldsymbol{s}_{\tau_i,t})\right)\left(f_0(C_0(\tau_i),\eta_{k,0}) + \sum_{u=1}^{U}\lambda_{k,u}(f_u(C_u(\tau_i),\eta_{k,u}))\right),$$

which reduces to the prototypical *action-based* algorithm REINFORCE (Williams, 1992) in the risk-neutral case. When allowed by the choice of $f_i$, we switch to a GPOMDP-style estimator (Baxter and Bartlett, 2001) which suffers less variance. See Appendix D for details.

C-PGPE (Algorithm 2) is the *parameter-based* version of C-PG. At each iteration $k \in [\![K]\!]$, the algorithm samples $N$ parameter configurations $\{\boldsymbol{\theta}_i\}_{i\in[\![N]\!]}$ from the hyperpolicy $\nu_{\boldsymbol{\rho}_k}$, then collects a single trajectory $\tau_i$, obtained by playing the policy $\pi_{\boldsymbol{\theta}_i}$, for each sampled parameterization $\boldsymbol{\theta}_i$. The sampled trajectories and parameters are then used to alternatively update the hyperpolicy parameter vector $\boldsymbol{\rho}$ and the risks parameter vector $\boldsymbol{\eta}$, or the vector of Lagrange multipliers $\boldsymbol{\lambda}$. In particular, the $\boldsymbol{\rho}$ update relies on the following estimator:

$$\widehat{\nabla}_{\boldsymbol{\rho}}\widetilde{\mathcal{L}}_{P,\omega}(\boldsymbol{\rho}_k,\boldsymbol{\lambda}_k,\boldsymbol{\eta}_k) \coloneqq \frac{1}{N}\sum_{i=1}^{N}\nabla_{\boldsymbol{\rho}}\log\nu_{\boldsymbol{\rho}_k}(\boldsymbol{\theta}_i)\left(f_0(C_0(\tau_i),\eta_{k,0}) + \sum_{u=1}^{U}\lambda_u f_u(C_u(\tau_i),\eta_{k,u})\right),$$

which reduces to the *parameter-based* algorithm PGPE (Sehnke et al., 2010) in the risk-neutral case.

# 5    Numerical Validation

In this section, we empirically validate some of the theoretical results shown throughout this work. Experimental details and additional results are provided in Appendix H. The code to run the experiments in this paper is available at `https://github.com/MontenegroAlessandro/MagicRL`.

**Comparison in DGWW.** We compare our C-PGAE against the sample-based versions of NPG-PD (Ding et al., 2020, Appendix H) and RPG-PD (Ding et al., 2024, Appendix C.9) on a Discrete Grid World with Walls (DGWW, see Appendix H) with a horizon of $T = 100$, and with a single constraint on the average trajectory cost. The methods are learning the parameters of a tabular softmax policy and the learning phase considers constant step sizes. Figure 2a shows the average return and the average cost curves over the trajectories seen during the learning. As can be noticed, C-PGAE strikes the objective of the COP with fewer trajectories w.r.t. the competitors. Indeed, both NPG-PD and RPG-PD require additional $\mathcal{O}(|\mathcal{S}| + |\mathcal{S}||\mathcal{A}|)$ trajectories per iteration.

**Comparison in LQR.** We compare our C-PGAE and C-PGPE against the continuous sample-based version of NPG-PD2 (Ding et al., 2022, Algorithm 1), working with generic policy parameterizations. Additionally, we consider RPG-PD2, a ridge-regularized version of NPG-PD2. The environment is a bidimensional *CostLQR* (Appendix H) with a horizon $T = 50$ and a single cost that the algorithms should keep below $b = 0.2$ on average. C-PGAE, NPG-PD2, and RPG-PD2 learn the parameter of a *linear gaussian policy*, while C-PGPE the ones of a *gaussian hyperpolicy* over a *linear deterministic policy*. All the step sizes are chosen with Adam (Kingma and Ba, 2015) scheduler. Figure 2b reports the learning curves for the average return and the cost, confirming that our methods solve the COP with fewer trajectories. Indeed, being both NPG-PD2 and RPG-PD2 actor-critic methods, they suffer from the inner critic loop, which requires the collection of additional trajectories (500 here). We stress that the actor-critic methods were very sentitive to the hyperparameters selection, especially the length and the step size of the inner loop (see Appendix H).

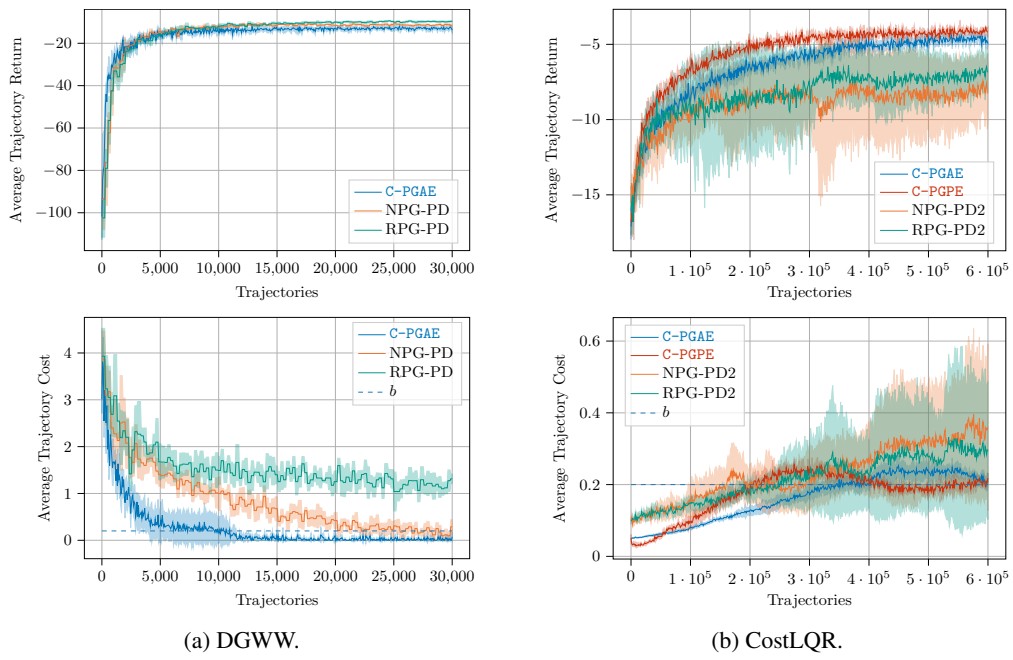

(a) DGWW.

(b) CostLQR.

Figure 2: Average return and cost in *CostLQR* and *DGWW* environments (5 runs, mean $\pm 95\%$ C.I.).

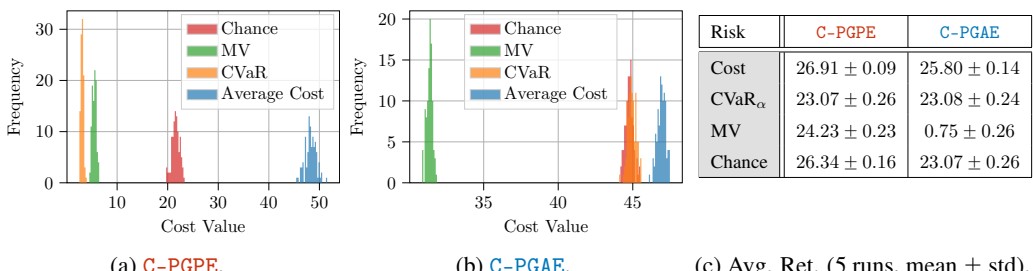

(a) C-PGPE.

(b) C-PGAE.

(c) Avg. Ret. (5 runs, mean $\pm$ std).

| Risk | C-PGPE | C-PGAE |
|---|---|---|
| Cost | $26.91 \pm 0.09$ | $25.80 \pm 0.14$ |
| CVaR$_\alpha$ | $23.07 \pm 0.26$ | $23.08 \pm 0.24$ |
| MV | $24.23 \pm 0.23$ | $0.75 \pm 0.26$ |
| Chance | $26.34 \pm 0.16$ | $23.07 \pm 0.26$ |

Figure 3: Cost distributions with (hyper)policies learned considering different risk measures (5 runs).

**Risk constraints on Swimmer.** In Figure 3, we show the empirical distributions of costs over 100 trajectories of the learned (hyper)policies via C-PGPE and C-PGAE. This experiment considers the cost-based version of the *Swimmer-v4* MuJoCo (Todorov et al., 2012) environment, with a single constraint over the actions (see Appendix H), for which we set $b = 50$. The experimental results show that C-PGPE learns a hyperpolicy paying less cost when using risk measures compared to average cost, with the smallest costs attained by CVaR$_\alpha$. C-PGAE shows similar results, although the difference between CVaR$_\alpha$ or the Chance constraints and average cost constraints are not very significant. Notice that, the minimum amount of cost is obtained using MV constraints even if the learned policy exhibits poor performances (Table 3c). In all the other cases, both C-PGPE and C-PGAE learns (hyper)policies exhibiting similar performance scores.

## 6 Conclusions

In this work, we proposed a general framework to address continuous CRL problems via *primal-dual policy-based* algorithms, leveraging on an alternate ascent-descent approach. Our *exploration-agnostic* proposal C-PG exhibits *dimension-free global last-iterate* convergence guarantees, under the standard (weak) gradient domination assumption. Furthermore, we introduced C-PGAE and C-PGPE, the action and parameter-based versions of C-PG which enable embedding risk-based constraints, enlarging the capabilities of our framework in addressing constrained real-world problems. Future works should focus on matching the sample complexity lower bound prescribed by (Vaswani et al., 2022) and devising algorithms with the same rates of C-PG with a single time-scale.

## Acknowledgements

The authors acknowledge the project FAIR that has been funded by the European Union – Next Generation EU within the project NRPP M4C2, Investment 1.3 DD. 341 - 15 March 2022 – FAIR – Future Artificial Intelligence Research – Spoke 4 - PE00000013 - D53C22002380006. The authors acknowledge the CINECA award under the ISCRA initiative, for the availability of high-performance computing resources and support. The authors acknowledge AI4REALNET that has received funding from European Union's Horizon Europe Research and Innovation programme under the Grant Agreement No 101119527. Views and opinions expressed are however those of the author(s) only and do not necessarily reflect those of the European Union. Neither the European Union nor the granting authority can be held responsible for them.

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

# Appendix

## Table of Contents

## A  Pseudo-codes

---

**Algorithm 1:** `C-PGAE`.

**Input:** Iterations $K$; batch size $N$; regularization $\omega$; initial parameters: $\boldsymbol{\theta}_0$, $\boldsymbol{\lambda}_0$, and $\boldsymbol{\eta}_0$; step sizes: $\zeta_{\boldsymbol{\theta},k}$, $\zeta_{\boldsymbol{\lambda},k}$, $\zeta_{\boldsymbol{\eta},k}$.

1 **for** $k \in [\![K]\!]$ **do**
2      Collect $N$ trajectories $\{\tau_i\}_{i \in [\![N]\!]}$ with $\pi_{\boldsymbol{\theta}_{k-1}}$
3      **if** $k$ is odd **then**
4         $\boldsymbol{\theta}_k \leftarrow \boldsymbol{\theta}_{k-1} - \zeta_{\boldsymbol{\theta},k-1}\widehat{\nabla}_{\boldsymbol{\theta}}\widetilde{\mathcal{L}}_{A,\omega}(\boldsymbol{\theta}_{k-1}, \boldsymbol{\lambda}_{k-1}, \boldsymbol{\eta}_{k-1})$
5         $\boldsymbol{\eta}_k \leftarrow \boldsymbol{\eta}_{k-1} - \zeta_{\boldsymbol{\eta},k-1}\widehat{\nabla}_{\boldsymbol{\eta}}\widetilde{\mathcal{L}}_{A,\omega}(\boldsymbol{\theta}_{k-1}, \boldsymbol{\lambda}_{k-1}, \boldsymbol{\eta}_{k-1})$
6      **else**
7         $\boldsymbol{\lambda}_k \leftarrow \boldsymbol{\lambda}_{k-1} + \zeta_{\boldsymbol{\lambda},k-1}\widehat{\nabla}_{\boldsymbol{\lambda}}\widetilde{\mathcal{L}}_{A,\omega}(\boldsymbol{\theta}_{k-1}, \boldsymbol{\lambda}_{k-1}, \boldsymbol{\eta}_{k-1})$
8      **end**
9 **end**
10 Return $\boldsymbol{\theta}_K$

---

**Algorithm 2:** `C-PGPE`.

**Input:** Iterations $K$; batch size $N$; regularization $\omega$; initial parameters: $\boldsymbol{\rho}_0$, $\boldsymbol{\lambda}_0$, and $\boldsymbol{\eta}_0$; step sizes: $\zeta_{\boldsymbol{\rho},k}$, $\zeta_{\boldsymbol{\lambda},k}$, $\zeta_{\boldsymbol{\eta},k}$.

1 **for** $k \in [\![K]\!]$ **do**
2      Sample $N$ parameters $\{\theta_i\}_{i \in [\![N]\!]}$ with $\nu_{\boldsymbol{\rho}_{k-1}}$
3      With each $\{\pi_{\theta_i}\}_{i \in [\![N]\!]}$ collect a trajectory $\tau_i$
4      **if** $k$ is odd **then**
5         $\boldsymbol{\rho}_k \leftarrow \boldsymbol{\rho}_{k-1} - \zeta_{\boldsymbol{\rho},k-1}\widehat{\nabla}_{\boldsymbol{\rho}}\widetilde{\mathcal{L}}_{P,\omega}(\boldsymbol{\rho}_{k-1}, \boldsymbol{\lambda}_{k-1}, \boldsymbol{\eta}_{k-1})$
6         $\boldsymbol{\eta}_k \leftarrow \boldsymbol{\eta}_{k-1} - \zeta_{\boldsymbol{\eta},k-1}\widehat{\nabla}_{\boldsymbol{\eta}}\widetilde{\mathcal{L}}_{P,\omega}(\boldsymbol{\rho}_{k-1}, \boldsymbol{\lambda}_{k-1}, \boldsymbol{\eta}_{k-1})$
7      **else**
8         $\boldsymbol{\lambda}_k \leftarrow \boldsymbol{\lambda}_{k-1} + \zeta_{\boldsymbol{\lambda},k-1}\widehat{\nabla}_{\boldsymbol{\lambda}}\widetilde{\mathcal{L}}_{P,\omega}(\boldsymbol{\rho}_{k-1}, \boldsymbol{\lambda}_{k-1}, \boldsymbol{\eta}_{k-1})$
9      **end**
10 **end**
11 Return $\boldsymbol{\rho}_K$

---

# B  Related Works

**Policy Optimization Approaches for Constrained Reinforcement Learning.** Policy Optimization-based algorithms for Constrained Reinforcement Learning mostly follow *primal*-only or *primal-dual* approaches. *Primal*-only algorithms (Dalal et al., 2018; Chow et al., 2018; Yu et al., 2019; Liu et al., 2020; Xu et al., 2021) avoid considering dual variables by focusing on the design of the objective function and by designing the update rules for the policy at hand incorporating the constraint satisfaction part.

The main benefit of employing primal-only algorithms lies in the fact that there is no need to consider another variable to learn, and therefore, no need to tune its learning rate. However, few of the existing methods establish global convergence to an optimal feasible solution. For instance, Xu et al. (2021) propose CRPO, an algorithm employing an *unconstrained* policy maximization update taking into account the reward when all the constraints are satisfied, while leveraging on-policy minimization updates in the direction of violated constraint functions. Moreover, it exhibits average global convergence guarantees for the tabular setting. On the other hand, *primal-dual* algorithms (Chow et al., 2017; Achiam et al., 2017; Tessler et al., 2019; Stooke et al., 2020; Ding et al., 2020, 2021; Bai et al., 2022; Ying et al., 2022; Bai et al., 2023; Gladin et al., 2023; Ding et al., 2024) are the most commonly used and investigated. Indeed, the effectiveness of using the primal-dual approach is justified by Paternain et al. (2019), which states that this kind of approach has zero duality gap under Slater's condition when optimizing over the space of all the possible stochastic policies. Among the reported works, Stooke et al. (2020) propose PID Lagrangian, a method to update the dual variable, smoothing the oscillations around the threshold value of the costs during the learning. The practical strength of such a method is that can be paired with any of the existing policy optimization methods. The other cited works are treated in details in the next paragraph.

**Lagrangian-based Policy Search Convergence Guarantees.** A lot of research effort has been spent in studying the convergence guarantees for primal-dual policy optimization methods. In this field, the goal is to ensure last-iterate convergence guarantees showing rates that are dimension-free, i.e., not relying on the state and action spaces' dimensions, and working with multiple constraints. In the rest of this paragraph, we talk about single time-scale algorithms when the methods at hand prescribe the usage of the same step sizes for both the primal and dual variables' updates. Vázquez-Abad et al. (2002) and Bhatnagar and Lakshmanan (2012) propose primal-dual policy gradient-based methods built upon distinct time-scales and relying on nested loops. Such methods only show *asymptotic* convergence guarantees. Chow et al. (2017) propose two primal-dual methods ensuring *asymptotic* convergence guarantees. The peculiarity of those methods lies in the fact that their notion of CMDP encapsulates risk-based constraints, introducing an additional learning variable. Their algorithms have guarantees of *asymptotic* convergence to stationary points. The recent works by Zheng et al. (2022) and Moskovitz et al. (2023) also propose methods ensuring *asymptotic* global convergence guarantees. These methods exploit occupancy-measure iterates rather than policy iterates. Ding et al. (2020) propose NPG-PD, which relies on a natural policy gradient approach and, under Slater's assumption, ensures dimension-dependent *average*-iterate global convergence guarantees in the single-constrained setting with a single time-scale and with exact gradients. This work has been extended by Ding et al. (2022), which strikes dimension-free rates, but still guaranteeing just *average*-rate convergence with exact gradients. However, sample-based versions of NPG-PD showing, under additional assumptions, the same convergence rates are provided by the authors. Another work ensuring an *average*-iterate rate is the one by Liu et al. (2021). The latter exhibits a convergence rate of order $\tilde{\mathcal{O}}(\epsilon^{-1})$, considering to act in tabular CMDPs with softmax policies and having access to exact graidents and to a generative model. Liu et al. (2021) propose also a sample-based version of their algorithm, keeping the same setting previously described, which ensures a convergence rate on *average* of order $\tilde{\mathcal{O}}(\epsilon^{-3})$. Both Ying et al. (2022) and Gladin et al. (2023) propose algorithms involving regularization. The proposed methods rely on natural policy-based subroutines and show dimension-dependent *last-iterate* global convergence guarantees, relying on two time-scales. These methods work also with multiple constraints. Finally, Ding et al. (2024) propose RPG-PD and OPG-PD, exhibiting *last-iterate* global convergence guarantees under Slater's condition in a single-constraint setting. The former is a regularized version of the algorithm proposed by Ding et al. (2020), showing *last-iterate* global convergence at a sublinear rate. The latter leverages on the optimistic gradient method (Hsieh et al., 2019) to unlock a faster linear convergence rate. These methods show single time-scale dimension-dependent rates and both leverage on exact gradients. However, for

| Algorithm | Dimension free | Setting | Exploration Type | Single time-scale | Gradients | Assumptions | Sample Complexity | Iteration Complexity |
|---|---|---|---|---|---|---|---|---|
| Dual Descent (Ying et al., 2022) | ✗ | $U \geqslant 1$ $T = \infty$ Softmax param. | AB | ✗ | Inexact | Slater Sufficient Exploration | $\mathcal{O}\left(\epsilon^{-2} \log^2 \epsilon^{-1}\right)$ | $\mathcal{O}\left(\log^2 \epsilon^{-1}\right)$ |
| Cutting-Plane (Gladin et al., 2023) | ✗ | $U \geqslant 1$ $T = \infty$ Softmax param. | AB | ✗ | Inexact | Slater Uniform Ergodicity Oracle | $\mathcal{O}\left(\epsilon^{-4} \log^3 \epsilon^{-1}\right)$ | $\mathcal{O}\left(\log^3 \epsilon^{-1}\right)$ |
| Exact RPG-PD (Ding et al., 2024) | ✗ | $U = 1$ $T = \infty$ Softmax param. | AB | ✓ | Exact | Slater | - | $\mathcal{O}\left(\epsilon^{-6} \log^2 \epsilon^{-1}\right)$ |
| Inexact RPG-PD (Ding et al., 2024) | ✗ | $U = 1$ $T = \infty$ Softmax param. | AB | ✓ | Inexact | Slater Stat. Err. Bounded Transf. Err. Bounded Cond. Num. $< +\infty$ | $\mathcal{O}\left(\epsilon^{-6} \log^2 \epsilon^{-1}\right)$ | $\mathcal{O}\left(\epsilon^{-6} \log^2 \epsilon^{-1}\right)$ |
| OPG-PD (Ding et al., 2024) | ✗ | $U = 1$ $T = \infty$ Softmax param. | AB | ✓ | Exact | Slater | - | $\mathcal{O}\left(\log^2 \epsilon^{-1}\right)$ |
| C-PG (This work) | ✓ | $U \geqslant 1$ $T \in \mathbb{N} \cup \{+\infty\}$ General param. | AB and PB | ✗ | Exact Inexact | Asm. 3.1, 3.2, 3.3, 3.4 | Table 1 | Table 1 |
| Lower Bound (Vaswani et al., 2022) | ✗ | $U = 1$ $T = \infty$ | - | - | Inexact | Slater | $\Omega\left(\epsilon^{-2}\right)$ | - |

Table 3: Comparison among primal-dual methods ensuring last-iterate global convergence guarantees.

RPG-PD there exists an *inexact* version showing, under additional assumptions on the statistical and transfer errors and the relative condition number (Ding et al., 2024, Assumption 2), the same guarantees of the exact one. It is worth noticing that all the mentioned works just consider the *action-based* exploration approach for policy optimization, while the *parameter-based* one remains unexplored. For the sake of clarity, Table 3 shows a detailed comparison among our approach and the other presented methods exhibiting *last-iterate* global convergence guarantees.

Furthermore, Vaswani et al. (2022) have recently proposed a dimension-dependent lower bound for the sample complexity of $\mathcal{O}\left(\epsilon^{-2}\right)$, assuming to be under the Slater condition and considering single-constrained CMDPs with finite state and action spaces.

**Risk-Constrained Reinforcement Learning.** For safety-critical problems, it is often insufficient to consider constraints solely on expected costs or utilities. Therefore, constraints are sometimes applied to risk measures over cost or utility functions (Chow and Pavone, 2013; Borkar and Jain, 2014; Chow et al., 2017; Zhang et al., 2024). The employed risk measures are those most commonly applied in risk-averse RL (Tamar et al., 2014; Bisi et al., 2020, 2022; Luo et al., 2023). For instance, Tamar et al. (2014) consider the Conditional Value at Risk (CVaR$_\alpha$, Rockafellar and Uryasev, 2000), while Bisi et al. (2022) consider a unified formulation embracing several risk measures, including the CVaR$_\alpha$ and the Mean-Variance (MV, Markowitz and Todd, 2000; Li and Ng, 2000). More details on risk measures are discussed in Appendix C. Among the works on Risk-Constrained RL, the one by Chow et al. (2017) considers both CVaR$_\alpha$ constraints and chance constraints over cost functions. In particular, the authors propose a trajectory-based policy gradient algorithm and an actor-critic one, both primal-dual methods. For what concerns the CVaR$_\alpha$-constrained problem, the authors incorporate into the constrained optimization problem the real-valued variable associated with the CVaR$_\alpha$. This leads to three variables to learn, requiring algorithms with *three* time-scales. Finally, the recent work by Zhang et al. (2024) considers an extension of the CPO algorithm (Achiam et al., 2017) imposing constraints over the CVaR$_\alpha$ of cost functions.

# C  On the Unified Risk Measure

In this section, we deepen the risk measures presented in our work, showing also how to select the functions $f$ and $g$ to obtain the desired risk measure. For what follows, we define $Z$ as a finite-mean random variable that has $F_Z(\cdot)$ as the cumulative distribution function. Formally:

$$\mathbb{E}\left[|Z|\right] < +\infty \quad \text{and} \quad F_Z(z) = \mathbb{P}(Z \leqslant z). \tag{14}$$

For what follows, let such a random variable represent a loss.

## C.1 Expected Cost and Mean-Variance

One of the most natural risk measures is represented directly by the *expected cost* on the trajectories induced by a (hyper)policy. This "risk neutral" risk measure was introduced in Section 2 for both the PG exploration paradigms as $J_{\text{AC},i}$ and $J_{\text{PC},i}$, where $i$ is the cost function index.

However, in some scenarios, it may be desirable to minimize jointly the expected cost and its variance. To this end, it is possible to consider the Mean-Variance (MV, Markowitz and Todd, 2000; Li and Ng, 2000) risk measure which models this kind of objective. Indeed, the MV is a combination of the mean cost and its variance over the trajectories. The MV with parameter $\kappa$ over the random variable $Z$, which represents a loss, is defined as follows:

$$\text{MV}_\kappa(Z) := \mathbb{E}\left[Z\right] + \kappa \, \mathbb{V}\text{ar}\left[Z\right]. \tag{15}$$

## C.2 Chance Constraints, Value at Risk, and Conditional Value at Risk

Value at Risk ($\text{VaR}_\alpha$) and Conditional Value at Risk ($\text{CVaR}_\alpha$) are two popular and strongly connected risk measures. In the following, we report their standard definition as provided by Rockafellar and Uryasev (2000) and Chow et al. (2017).

$\text{VaR}_\alpha$ measures risk as the maximum cost that might be incurred with respect to a given confidence level $\alpha \in (0, 1)$. Such a quantity models the *potential loss* and the *probability* that the loss will occur. In its classical definition, the $\text{VaR}_\alpha$ is the worst-case loss associated with a probability and a time horizon. This risk metric is particularly useful when there is a well defined failure state. Formally:

$$\text{VaR}_\alpha(Z) := \min\left\{z | F_Z(z) \geqslant \alpha\right\}. \tag{16}$$

$\text{CVaR}_\alpha$ measures risk as the expected cost given that such cost is greater than or equal to $\text{VaR}_\alpha$, and provides a number of theoretical and computational advantages. Such a quantity represents the *expected loss* if the worst-case threshold is passed (i.e., beyond the $\text{VaR}_\alpha$ breakpoint). Formally:

$$\text{CVaR}_\alpha(Z) := \min_{\eta \in \mathbb{R}}\left\{\eta + \frac{1}{1-\alpha} \mathbb{E}\left[(Z - \eta)^+\right]\right\}. \tag{17}$$

Another risk measure, similar to the $\text{VaR}_\alpha$ and the $\text{CVaR}_\alpha$, is the *chance* one. Considering *chance constraints* means to consider constraints to hold with high probability. Indeed, constraints are imposed on the probability that the loss random variable exceeds a certain threshold $n$:

$$\text{Chance}_n(Z) := \mathbb{P}(Z \geqslant n). \tag{18}$$

## C.3 Unified Risk Measure Mapping

Here we introduce an objective for writing an optimization problem for different risk measures (Bisi et al., 2022). For every $i \in [\![0, U]\!]$, we introduce the functions $f_i : \mathbb{R}^2 \to \mathbb{R}$ and $g_i : \mathbb{R} \to \mathbb{R}$ in order to express the unified risk-minimization objective function over a risk measure on $C_i$ as:

$$\min_{\eta_i \in \mathbb{R}}\left\{\min_{\boldsymbol{\theta} \in \Theta} \mathbb{E}_{\tau \sim p_{\text{A}}(\cdot|\boldsymbol{\theta})}\left[f_i(C_i(\tau), \eta_i)\right] + g_i(\eta_i)\right\}. \tag{19}$$

Thus, by fixing a policy parameter configuration $\boldsymbol{\theta}$, we obtain an AB unified risk measure over $C_i$ as follows:

$$\mathcal{Z}_{\text{A},i}(\boldsymbol{\theta}) := \min_{\eta_i \in \mathbb{R}}\left\{\mathbb{E}_{\tau \sim p_{\text{A}}(\cdot|\boldsymbol{\theta})}\left[f_i(C_i(\tau), \eta_i)\right] + g_i(\eta_i)\right\}. \tag{20}$$

In order to switch to the PB exploration paradigm, we need to consider also the expectation over the parameter configuration of the underlying policy. Indeed, by fixing a hyperparameter configuration $\boldsymbol{\rho}$, we can define:

$$\mathcal{Z}_{\text{P},i}(\boldsymbol{\rho}) := \min_{\eta_i \in \mathbb{R}}\left\{\mathbb{E}_{\boldsymbol{\theta} \sim \nu_{\boldsymbol{\rho}}}\left[\mathbb{E}_{\tau \sim p_{\text{A}}(\cdot|\boldsymbol{\theta})}\left[f_i(C_i(\tau), \eta_i)\right]\right] + g_i(\eta_i)\right\}. \tag{21}$$

By selecting the functions $f_i$ and $g_i$ we can consider distinct risk measures to minimize, as we show in Table 4.

The focus of this section is to derive the mapping between the functions $f$ and $g$ and the desired risk measure as shown in Table 4. In particular, we consider just the AB exploration scenario, since then passing to the PB one is straightforward. In this section, we consider a generic cost function $c : \mathcal{S} \times \mathcal{A} \to [0, 1]$ with its cumulative cost index over a trajectory $\tau$ which is $C(\tau) = \sum_{t=0}^{T-1} \gamma^t c(\boldsymbol{s}_{\tau,t}, \boldsymbol{a}_{\tau,t})$. In what follows, with a little abuse of notation, we will write the risk measures with also a dependence on the policy parameters $\boldsymbol{\theta}$ which induce trajectories $\tau$.

**Mean Cost.** This is the simplest case, indeed, we can express the minimization problem of the mean cost as:

$$\min_{\boldsymbol{\theta} \in \Theta} \mathbb{E}_{\tau \sim p_A(\cdot|\boldsymbol{\theta})} [C(\tau)]. \tag{22}$$

Thus, it suffices to select $f(C(\tau), \eta) = C(\tau)$ and $g(\eta) = 0$ to make the minimization problem fit with the form of the unified cost minimization formulation.

**Conditional Value at Risk.** For what concern the $\mathrm{CVaR}_\alpha$, we can start from its formulation:

$$\min_{\boldsymbol{\theta} \in \Theta} \mathrm{CVaR}_\alpha(\boldsymbol{\theta}) = \min_{\boldsymbol{\theta} \in \Theta} \left\{ \min_{\eta \in \mathbb{R}} \left\{ \eta + \frac{1}{1-\alpha} \mathbb{E}_{\tau \sim p_A(\cdot|\boldsymbol{\theta}} \left[ (C(\tau) - \eta)^+ \right] \right\} \right\} \tag{23}$$

$$= \min_{\eta \in \mathbb{R}} \left\{ \eta + \min_{\boldsymbol{\theta} \in \Theta} \mathbb{E}_{\tau \sim p_A(\cdot|\boldsymbol{\theta})} \left[ \frac{1}{1-\alpha} (C(\tau) - \eta)^+ \right] \right\}. \tag{24}$$

Thus, we need to select $f(C(\tau), \eta) = \frac{1}{1-\alpha}(C(\tau) - \eta)^+$ and $g(\eta) = \eta$ to complete the mapping to the unified cost minimization objective.

**Mean Variance.** For the $\mathrm{MV}_\kappa(\boldsymbol{\theta})$ objective, we need to make some preliminary observations. In particular, for a generic finite-mean random variable $X$, we have that:

$$\mathbb{V}\mathrm{ar}[X] = \mathbb{E}[X^2] - \mathbb{E}[X]^2. \tag{25}$$

Moreover, by Fenchel duality, we have the following:

$$\mathbb{E}[X]^2 = \max_{\eta \in \mathbb{R}} \left\{ 2\eta \mathbb{E}[X] - \eta^2 \right\}. \tag{26}$$

Given that, we can start the derivation from the minimization of the Mean Variance:

$$\min_{\boldsymbol{\theta} \in \Theta} \mathrm{MV}_\kappa(C, \boldsymbol{\theta}) \tag{27}$$

$$= \min_{\boldsymbol{\theta} \in \Theta} \left\{ \mathbb{E}_{\tau \sim p_A(\cdot|\boldsymbol{\theta})} [C(\tau)] + \kappa \mathbb{E}_{\tau \sim p_A(\cdot|\boldsymbol{\theta})} [C(\tau)^2] - \kappa \mathbb{E}_{\tau \sim p_A(\cdot|\boldsymbol{\theta})} [C(\tau)]^2 \right\} \tag{28}$$

$$= \min_{\boldsymbol{\theta} \in \Theta} \left\{ \mathbb{E}_{\tau \sim p_A(\cdot|\boldsymbol{\theta})} [C(\tau)] + \kappa \mathbb{E}_{\tau \sim p_A(\cdot|\boldsymbol{\theta})} [C(\tau)^2] - \kappa \max_{\eta \in \mathbb{R}} \left\{ 2\eta \mathbb{E}_{\tau \sim p_A(\cdot|\boldsymbol{\theta})} [C(\tau)] - \eta^2 \right\} \right\} \tag{29}$$

$$= \min_{\boldsymbol{\theta} \in \Theta} \left\{ \mathbb{E}_{\tau \sim p_A(\cdot|\boldsymbol{\theta})} [C(\tau)] + \kappa \mathbb{E}_{\tau \sim p_A(\cdot|\boldsymbol{\theta})} [C(\tau)^2] + \kappa \min_{\eta \in \mathbb{R}} \left\{ -2\eta \mathbb{E}_{\tau \sim p_A(\cdot|\boldsymbol{\theta})} [C(\tau)] + \eta^2 \right\} \right\} \tag{30}$$

$$= \min_{\eta \in \mathbb{R}} \left\{ \kappa\eta^2 + \min_{\boldsymbol{\theta} \in \Theta} \left\{ (1 - 2\kappa\eta) \mathbb{E}_{\tau \sim p_A(\cdot|\boldsymbol{\theta})} [C(\tau)] + \kappa \mathbb{E}_{\tau \sim p_A(\cdot|\boldsymbol{\theta})} [C(\tau)^2] \right\} \right\}. \tag{31}$$

Thus, to complete the mapping, we need to select

$$f(C(\tau), \eta) = (1 - 2\kappa\eta) \mathbb{E}_{\tau \sim p_A(\cdot|\boldsymbol{\theta})} [C(\tau)] + \kappa \mathbb{E}_{\tau \sim p_A(\cdot|\boldsymbol{\theta})} [C(\tau)^2]$$

and $g(\eta) = \kappa\eta^2$.

**Chance.** For characterizing the chance constraints, we start by expressing the probability of an event $A$ as the expected value of the indicator function:

$$\mathbb{P}(A) = \mathbb{E}[\mathbb{1}\{A\}]. \tag{32}$$

Thus, having to minimize a chance measure on the cost with a parameter $n$, we can rewrite it as:

$$\min_{\boldsymbol{\theta} \in \Theta} \mathrm{Chance}_n(C, \boldsymbol{\theta}) = \min_{\boldsymbol{\theta} \in \Theta} \mathbb{E}_{\tau \sim p_A(\cdot|\boldsymbol{\theta})} [\mathbb{P}(C(\tau) \geqslant n)] \tag{33}$$

$$= \min_{\boldsymbol{\theta} \in \Theta} \mathbb{E}_{\tau \sim p_A(\cdot|\boldsymbol{\theta})} [\mathbb{1}\{C(\tau) \geqslant n\}]. \tag{34}$$

Thus, to complete the mapping to the unified cost minimization objective, it suffices to select $f(C(\tau), \eta) = \mathbb{1}\{C(\tau) \geqslant n\}$ and $g(\eta) = 0$.

## D    Estimators

Here, we provide all the explicit estimators' forms for both `C-PGAE` and `C-PGPE` and for each risk measure. For the sake of clarity, we report the mapping between the $f_i$ and $g_i$ functions and the risk measures in Table 4.

| Risk Measure | Parameter | Need for $\eta$ | $f_i(C_i(\tau), \eta)$ | $g_i(\eta)$ | **AB** GPOMDP-like Estimator |
|:---:|:---:|:---:|:---:|:---:|:---:|
| Expected Cost | - | ✗ | $C_i(\tau)$ | $0$ | Yes |
| Mean Variance | $\kappa$ | ✓ | $(1 - 2\kappa\eta)C_i(\tau) + \kappa C_i(\tau)^2$ | $\kappa\eta^2$ | Partial |
| CVaR$_\alpha$ | $\alpha$ | ✓ | $\frac{1}{1-\alpha}(C_i(\tau) - \eta)^+$ | $\eta$ | No |
| Chance | $n$ | ✗ | $\mathbb{1}\{C_i(\tau) \geqslant n\}$ | $0$ | No |

Table 4: Mapping between $f_i$ and $g_i$ and the cost measures.

The general Lagrangian function for the problem in Equation (11) is the following:

$$\widetilde{\mathcal{L}}_{\dagger,\omega}(\boldsymbol{v}, \boldsymbol{\lambda}, \boldsymbol{\eta}) \coloneqq \mathcal{J}_{\dagger,0}(\boldsymbol{v}, \eta_0) + \sum_{u=1}^{U} \lambda_u \left( \mathcal{J}_{\dagger,u}(\boldsymbol{v}, \eta_u) - b_u \right) - \frac{\omega}{2} \|\boldsymbol{\lambda}\|_2^2. \tag{35}$$

For what follows we are going to compute the following gradients:

1. $\nabla_{\boldsymbol{v}}\widetilde{\mathcal{L}}_{\dagger,\omega}(\boldsymbol{v}, \boldsymbol{\lambda}, \boldsymbol{\eta})$;
2. $\nabla_{\boldsymbol{\lambda}}\widetilde{\mathcal{L}}_{\dagger,\omega}(\boldsymbol{v}, \boldsymbol{\lambda}, \boldsymbol{\eta})$;
3. $\nabla_{\boldsymbol{\eta}}\widetilde{\mathcal{L}}_{\dagger,\omega}(\boldsymbol{v}, \boldsymbol{\lambda}, \boldsymbol{\eta})$.

Before entering into the details of the estimators, we show the general forms of the gradients of the Lagrangian.

**General Gradient w.r.t. $\boldsymbol{v}$.**    Notice that the following holds:

$$\nabla_{\boldsymbol{v}}\widetilde{\mathcal{L}}_{\dagger,\omega}(\boldsymbol{v}, \boldsymbol{\lambda}, \boldsymbol{\eta}) = \nabla_{\boldsymbol{v}}\mathcal{J}_{\dagger,0}(\boldsymbol{v}, \eta_0) + \sum_{u=1}^{U} \lambda_u \nabla_{\boldsymbol{v}}\mathcal{J}_{\dagger,u}(\boldsymbol{v}, \eta_u), \tag{36}$$

thus, in what follows, for the gradient w.r.t. $\boldsymbol{v}$ we focus on the single terms $\nabla_{\boldsymbol{v}}\mathcal{J}_{\dagger,u}(\boldsymbol{v}, \eta_u)$ for every $u \in [\![0, U]\!]$.

**General Gradient w.r.t. $\boldsymbol{\lambda}$.**    The general gradient w.r.t. $\boldsymbol{\lambda}$ has the form:

$$\nabla_{\boldsymbol{\lambda}}\widetilde{\mathcal{L}}_{\dagger,\omega}(\boldsymbol{v}, \boldsymbol{\lambda}, \boldsymbol{\eta}) = \boldsymbol{\mathcal{J}}_{\dagger}(\boldsymbol{v}, \boldsymbol{\eta}) - \boldsymbol{b} + \omega\boldsymbol{\lambda}, \tag{37}$$

where $\boldsymbol{\mathcal{J}}_{\dagger}(\boldsymbol{v}, \boldsymbol{\eta}) = (\mathcal{J}_{\dagger,1}(\boldsymbol{v}, \eta_1), ..., \mathcal{J}_{\dagger,U}(\boldsymbol{v}, \eta_U))^{\top}$ and $\boldsymbol{b} = (b_1, ..., b_U)^{\top}$.

**General Gradient w.r.t. $\boldsymbol{\eta}$.**    As done for the gradient w.r.t. $\boldsymbol{v}$, the following holds:

$$\nabla_{\boldsymbol{\eta}}\widetilde{\mathcal{L}}_{\dagger,\omega}(\boldsymbol{v}, \boldsymbol{\lambda}, \boldsymbol{\eta}) = \nabla_{\boldsymbol{\eta}}\mathcal{J}_{\dagger,0}(\boldsymbol{v}, \eta_0) + \sum_{u=1}^{U} \lambda_u \nabla_{\boldsymbol{\eta}}\mathcal{J}_{\dagger,u}(\boldsymbol{v}, \eta_u). \tag{38}$$

Thus, also in this case, we will focus on the single terms $\nabla_{\boldsymbol{\eta}}\mathcal{J}_{\dagger,u}(\boldsymbol{v}, \eta_u)$ for every $u \in [\![0, U]\!]$.

### D.1    Parameter-based Algorithm: `C-PGPE`

In the following we consider a generic hyperpolicy $\nu_{\boldsymbol{\rho}}$ and a generic parameterization $\boldsymbol{\rho}$. Before starting with the derivations, we report the definition of $\mathcal{J}_{\mathrm{P},i}(\boldsymbol{\rho}, \eta_i)$:

$$\mathcal{J}_{\mathrm{P},i}(\boldsymbol{\rho}, \eta_i) \coloneqq \mathop{\mathbb{E}}_{\boldsymbol{\theta} \sim \nu_{\boldsymbol{\rho}}} \left[ \mathop{\mathbb{E}}_{\tau \sim p_{\mathrm{A}}(\cdot|\boldsymbol{\theta})} \left[ f_i(C_i(\tau), \eta_i) \right] \right] + g_i(\eta_i), \tag{39}$$

for every $i \in [\![0, U]\!]$.

### D.1.1 Gradients w.r.t. Parameters

The first step of the derivation can be done via the log-trick for the parameter-based exploration paradigm as stated by Sehnke et al. (2010):

$$\nabla_{\boldsymbol{\rho}} \mathcal{J}_{\mathrm{P},i}(\boldsymbol{\rho}, \eta_i) = \nabla_{\boldsymbol{\rho}} \underset{\boldsymbol{\theta} \sim \nu_{\boldsymbol{\rho}}}{\mathbb{E}} \left[ \underset{\tau \sim p_{\mathrm{A}}(\cdot|\boldsymbol{\theta})}{\mathbb{E}} \left[ f_i(C_i(\tau), \eta_i) \right] \right] \tag{40}$$

$$= \underset{\boldsymbol{\theta} \sim \nu_{\boldsymbol{\rho}}}{\mathbb{E}} \left[ \nabla_{\boldsymbol{\rho}} \log \nu_{\boldsymbol{\rho}}(\boldsymbol{\theta}) \underset{\tau \sim p_{\mathrm{A}}(\cdot|\boldsymbol{\theta})}{\mathbb{E}} \left[ f_i(C_i(\tau), \eta_i) \right] \right]. \tag{41}$$

To switch to the sample-based versions of all the gradients, we consider the behavior of `C-PGPE` described in Section 4. In particular, we have the following:

$$\widehat{\nabla}_{\boldsymbol{\rho}} \mathcal{J}_{\mathrm{P},i}(\boldsymbol{\rho}, \eta_i) = \frac{1}{N} \sum_{j=1}^{N} \nabla_{\boldsymbol{\rho}} \log \nu_{\boldsymbol{\rho}}(\boldsymbol{\theta}_j) f_i(C_i(\tau_j), \eta_i). \tag{42}$$

Thus, to obtain the estimators for all the risk measures, it suffices to map the selection of the $f_i$ functions in Equation (42).

### D.1.2 Gradients w.r.t. Lagrangian Multipliers

Given the general gradient w.r.t. $\boldsymbol{\lambda}$ of the regularized Lagrangian $\widetilde{\mathcal{L}}_{\mathrm{P},\omega}$ in the parameter-based scenario, the partial derivative w.r.t. $\lambda_i$ is the following:

$$\nabla_{\lambda_i} \widetilde{\mathcal{L}}_{\mathrm{P},\omega}(\boldsymbol{\rho}, \boldsymbol{\lambda}, \boldsymbol{\eta}) = \underset{\boldsymbol{\theta} \sim \nu_{\boldsymbol{\rho}}}{\mathbb{E}} \left[ \underset{\tau \sim p_{\mathrm{A}}(\cdot|\boldsymbol{\theta})}{\mathbb{E}} \left[ f_i(C_i(\tau), \eta_i) \right] \right] + g_i(\eta_i) - b_i + \omega \lambda_i. \tag{43}$$

This is defined for any $i \in [\![U]\!]$.

In order to switch to the sample-based version of the partial derivative, we consider the behavior of `C-PGPE` as described in Section 4. In particular, we have the following:

$$\widehat{\nabla}_{\lambda_i} \widetilde{\mathcal{L}}_{\mathrm{P},\omega}(\boldsymbol{\rho}, \boldsymbol{\lambda}, \boldsymbol{\eta}) = \frac{1}{N} \sum_{j=1}^{N} f_i(C_i(\tau_j), \eta_i) + g_i(\eta_i) - b_i + \omega \lambda_i. \tag{44}$$

Thus, to obtain the estimator for all the risk measures, it suffices to map the choices of $f_i$ and $g_i$ in Equation (44).

### D.1.3 Gradients w.r.t. Risk Parameters

For what concern the gradients w.r.t. $\boldsymbol{\eta}$, we have to enumerate the mappings shown in Table 4. Notice that, when the employed risk measure is the *expected cost* or the *chance*, the estimator of the gradient w.r.t. $\boldsymbol{\eta}$ is not needed. Indeed, in the unified risk measure formulation such mappings do not depend on $\boldsymbol{\eta}$. As shown at the beginning of this section, we can just focus on the terms $\nabla_{\boldsymbol{\eta}} \mathcal{J}_{\mathrm{P},i}(\boldsymbol{\rho}, \eta_i)$, and in particular on the partial derivative $\nabla_{\eta_i} \mathcal{J}_{\mathrm{P},i}(\boldsymbol{\rho}, \eta_i)$, that exhibits the common form:

$$\nabla_{\eta_i} \mathcal{J}_{\mathrm{P},i}(\boldsymbol{\rho}, \eta_i) = \nabla_{\eta_i} \underset{\boldsymbol{\theta} \sim \nu_{\boldsymbol{\rho}}}{\mathbb{E}} \left[ \underset{\tau \sim p_{\mathrm{A}}(\cdot|\boldsymbol{\theta})}{\mathbb{E}} \left[ f_i(C_i(\tau), \eta_i) \right] \right] + \nabla_{\eta_i} g_i(\eta_i) \tag{45}$$

$$= \underset{\boldsymbol{\theta} \sim \nu_{\boldsymbol{\rho}}}{\mathbb{E}} \left[ \underset{\tau \sim p_{\mathrm{A}}(\cdot|\boldsymbol{\theta})}{\mathbb{E}} \left[ \nabla_{\eta_i} f_i(C_i(\tau), \eta_i) \right] \right] + \nabla_{\eta_i} g_i(\eta_i). \tag{46}$$

**Mean Variance.**   In this case we have that $f_i(C_i(\tau), \eta_i) = (1 - 2\kappa_i \eta_i)C_i(\tau) + \kappa_i C_i(\tau)^2$ and that $g_i(\eta_i) = \kappa_i \eta_i^2$. Thus, from Equation (46), we get the following:

$$\nabla_{\eta_i} \mathcal{J}_{\mathrm{P},i}(\boldsymbol{\rho}, \eta_i) = \underset{\boldsymbol{\theta} \sim \nu_{\boldsymbol{\rho}}}{\mathbb{E}} \left[ \underset{\tau \sim p_{\mathrm{A}}(\cdot|\boldsymbol{\theta})}{\mathbb{E}} \left[ -2\kappa_i C_i(\tau) \right] \right] + 2\kappa_i \eta_i. \tag{47}$$

Considering the behavior of `C-PGPE` described in Section 4, we obtain the following sample-based version:

$$\widehat{\nabla}_{\eta_i} \mathcal{J}_{\mathrm{P},i}(\boldsymbol{\rho}, \eta_i) = -\frac{2\kappa_i}{N} \sum_{j=1}^{N} C_i(\tau_j) + 2\kappa_i \eta_i. \tag{48}$$

**Conditional Value at Risk.** In this case we have that $f_i(C_i(\tau), \eta_i) = \frac{1}{1-\alpha_i}\left(C_i(\tau) - \eta_i\right)^+$ and that $g_i(\eta_i) = \eta_i$. As also shown by Chow et al. (2017), due to the presence of the non-differentiable term $(\cdot)^+$, we need to resort to sub-differentiability theory. The following holds:

$$\partial_{\eta_i}(C_i(\tau) - \eta_i)^+ = \begin{cases} -1 & \text{if } C_i(\tau) > \eta_i \\ -q: \; q \in [0,1] & \text{if } C_i(\tau) = \eta_i \\ 0 & \text{elsewhere.} \end{cases} \tag{49}$$

Thus, from Equation (46), we can write what follows:

$$\partial_{\eta_i} \mathcal{J}_{\mathrm{P},i}(\boldsymbol{\rho}, \eta_i) \tag{50}$$

$$= \frac{1}{1-\alpha_i} \mathop{\mathbb{E}}_{\boldsymbol{\theta} \sim \nu_{\boldsymbol{\rho}}} \left[ \mathop{\mathbb{E}}_{\tau \sim p_{\mathrm{A}}(\cdot|\boldsymbol{\theta})} \left[ \partial_{\eta_i}\left(C_i(\tau) - \eta_i\right)^+ \right] \right] + 1 \tag{51}$$

$$= \frac{1}{1-\alpha_i} \mathop{\mathbb{E}}_{\boldsymbol{\theta} \sim \nu_{\boldsymbol{\rho}}} \left[ \int_\tau p_{\mathrm{A}}(\tau|\boldsymbol{\theta}) \partial_{\eta_i}\left(C_i(\tau) - \eta_i\right)^+ \mathrm{d}\tau \right] + 1 \tag{52}$$

$$= \frac{1}{1-\alpha_i} \mathop{\mathbb{E}}_{\boldsymbol{\theta} \sim \nu_{\boldsymbol{\rho}}} \left[ -\int_\tau p_{\mathrm{A}}(\tau|\boldsymbol{\theta}) q \mathbb{1}\left\{C_i(\tau) = \eta_i\right\} \mathrm{d}\tau - \int_\tau p_{\mathrm{A}}(\tau|\boldsymbol{\theta}) \mathbb{1}\left\{C_i(\tau) > \eta_i\right\} \mathrm{d}\tau \right] + 1, \tag{53}$$

with $q \in [0,1]$.

In particular, for $q = 1$, we obtain the following partial derivative:

$$\partial_{\eta_i} \mathcal{J}_{\mathrm{P},i}(\boldsymbol{\rho}, \eta_i) = \frac{1}{1-\alpha_i} \mathop{\mathbb{E}}_{\boldsymbol{\theta} \sim \nu_{\boldsymbol{\rho}}} \left[ \mathop{\mathbb{E}}_{\tau \sim p_{\mathrm{A}}(\cdot|\boldsymbol{\theta})} \left[ \mathbb{1}\left\{C_i(\tau) \geqslant \eta_i\right\} \right] \right] + 1. \tag{54}$$

Finally, according to the `C-PGPE` behavior described in Section 4, we obtain the following sample-based version:

$$\widehat{\partial}_{\eta_i} \mathcal{J}_{\mathrm{P},i}(\boldsymbol{\rho}, \eta_i) = \frac{1}{N} \sum_{j=1}^{N} \mathbb{1}\left\{C_i(\tau_j) \geqslant \eta_i\right\} + 1. \tag{55}$$

With a little abuse of notation, we will use $\widehat{\nabla}_{\eta_i} \mathcal{J}_{\mathrm{P},i}(\boldsymbol{\rho}, \eta_i) = \widehat{\partial}_{\eta_i} \mathcal{J}_{\mathrm{P},i}(\boldsymbol{\rho}, \eta_i)$ for the $\boldsymbol{\eta}$ update in the case in which the $\mathrm{CVaR}_\alpha$ risk measure is employed.

### D.2 Action-based Algorithm: `C-PGAE`

In the following we consider a generic policy $\pi_{\boldsymbol{\theta}}$ and a generic parameterization $\boldsymbol{\theta}$. Before starting with the derivations, we report the definition of $\mathcal{J}_{\mathrm{A},i}(\boldsymbol{\theta}, \eta_i)$:

$$\mathcal{J}_{\mathrm{A},i}(\boldsymbol{\theta}, \eta_i) := \mathop{\mathbb{E}}_{\tau \sim p_{\mathrm{A}}(\cdot|\boldsymbol{\theta})} \left[ f_i(C_i(\tau), \eta_i) \right] + g_i(\eta_i), \tag{56}$$

for every $i \in [\![0, U]\!]$.

#### D.2.1 Gradients w.r.t. Parameters

Also in this case, the first step of the derivation can be done via the log-trick, which provides an analogous result of the Policy Gradient Theorem (PGT, Sutton et al., 1999):

$$\nabla_{\boldsymbol{\theta}} \mathcal{J}_{\mathrm{A},i}(\boldsymbol{\theta}, \eta_i) = \nabla_{\boldsymbol{\theta}} \mathop{\mathbb{E}}_{\tau \sim p_{\mathrm{A}}(\cdot|\boldsymbol{\theta})} \left[ f_i(C_i(\tau), \eta_i) \right] \tag{57}$$

$$= \mathop{\mathbb{E}}_{\tau \sim p_{\mathrm{A}}(\cdot|\boldsymbol{\theta})} \left[ \nabla_{\boldsymbol{\theta}} \log p_{\mathrm{A}}(\tau|\boldsymbol{\theta}) f_i(C_i(\tau), \eta_i) \right] \tag{58}$$

$$= \mathop{\mathbb{E}}_{\tau \sim p_{\mathrm{A}}(\cdot|\boldsymbol{\theta})} \left[ \sum_{t=0}^{T-1} \nabla_{\boldsymbol{\theta}} \log \pi(\boldsymbol{a}_{\tau,t}, \boldsymbol{s}_{\tau,t}) f_i(C_i(\tau), \eta_i) \right]. \tag{59}$$

To switch to the sample-based versions of all the gradients, we generally resort to the Monte Carlo version of $\nabla_{\boldsymbol{\theta}} \mathcal{J}_{A,i}(\boldsymbol{\theta}, \eta_i)$, obtaining a REINFORCE-like (Williams, 1992) estimator, that is:

$$\widehat{\nabla}_{\boldsymbol{\theta}} \mathcal{J}_{A,i}(\boldsymbol{\theta}, \eta_i) = \frac{1}{N} \sum_{j=1}^{N} \left( \sum_{t=0}^{T-1} \nabla_{\boldsymbol{\theta}} \log \pi(\boldsymbol{a}_{\tau_j,t}, \boldsymbol{s}_{\tau_j,t}) \right) f_i(C_i(\tau_j), \eta_i). \tag{60}$$

Thus, to obtain the estimators for all the risk measures, it suffices to map the selection of the $f_i$ functions in Equation (60).

It is worth noticing that some of the choices for $f_i$ and $g_i$ allow to switch to a GPOMDP-like (Baxter and Bartlett, 2001) estimator, which suffers from less variance. This holds for the *expected cost* and *mean variance* risk measures.

**Expected Cost GPOMDP-like Estimator.** In this case $f_i(C_i(\tau), \eta_i) = C_i(\tau)$, thus we can obtain exactly the GPOMDP estimator:

$$\widehat{\nabla}_{\boldsymbol{\theta}} \mathcal{J}_{A,i}(\boldsymbol{\theta}, \eta_i) = \frac{1}{N} \sum_{j=1}^{N} \left( \sum_{t=0}^{T-1} \gamma^t c_i(\boldsymbol{s}_{\tau_j,t}, \boldsymbol{a}_{\tau_j,t}) \sum_{h=0}^{t} \nabla_{\boldsymbol{\theta}} \log \pi(\boldsymbol{a}_{\tau_j,h}, \boldsymbol{s}_{\tau_j,h}) \right). \tag{61}$$

**Mean Variance GPOMDP-like Estimator.** In this case $f_i(C_i(\tau), \eta_i) = (1 - 2\kappa_i \eta_i)C_i(\tau) + \kappa_i C_i(\tau)^2$, thus we can obtain the GPOMDP estimator just for the $C_i(\tau)$ part:

$$\widehat{\nabla}_{\boldsymbol{\theta}} \mathcal{J}_{A,i}(\boldsymbol{\theta}, \eta_i) \tag{62}$$

$$= \frac{1}{N} \sum_{j=1}^{N} \left( \sum_{t=0}^{T-1} \nabla_{\boldsymbol{\theta}} \log \pi(\boldsymbol{a}_{\tau_j,t}, \boldsymbol{s}_{\tau_j,t}) \right) \left( (1 - 2\kappa_i \eta_i)C_i(\tau_j) + \kappa_i C_i(\tau_j)^2 \right) \tag{63}$$

$$= \frac{1 - 2\kappa_i \eta_i}{N} \sum_{j=1}^{N} \left( \sum_{t=0}^{T-1} \gamma^t c_i(\boldsymbol{s}_{\tau_j,t}, \boldsymbol{a}_{\tau_j,t}) \sum_{h=0}^{t} \nabla_{\boldsymbol{\theta}} \log \pi(\boldsymbol{a}_{\tau_j,h}, \boldsymbol{s}_{\tau_j,h}) \right) \tag{64}$$

$$+ \frac{\kappa_i}{N} \sum_{j=1}^{N} \left( \sum_{t=0}^{T-1} \nabla_{\boldsymbol{\theta}} \log \pi(\boldsymbol{a}_{\tau_j,t}, \boldsymbol{s}_{\tau_j,t}) \right) C_i(\tau_j)^2. \tag{65}$$

### D.2.2 Gradients w.r.t. Lagrangian Multipliers

The result is the same as the one obtained for `C-PGPE`. The difference lies in the way in which trajectories are collected. The estimator for the partial derivative w.r.t. $\lambda_i$ is:

$$\widehat{\nabla}_{\lambda_i} \widetilde{\mathcal{L}}_{A,\omega}(\boldsymbol{\theta}, \boldsymbol{\lambda}, \boldsymbol{\eta}) = \frac{1}{N} \sum_{j=1}^{N} f_i(C_i(\tau_j), \eta_i) + g_i(\eta_i) - b_i + \omega \lambda_i. \tag{66}$$

To obtain the estimator for all the risk measures, it suffices to map the choices of $f_i$ and $g_i$ as prescribed by Table 4.

### D.2.3 Gradients w.r.t. Risk Parameters

As for the exploration-based case, also here we need to enumerate the mappings reported in Table 4. However, the *expected cost* and the *chance* risk measures do not depend on $\boldsymbol{\eta}$, thus they are not treated. As shown at the beginning of the section, we can just focus on the $\nabla_{\boldsymbol{\eta}} \mathcal{J}_{A,i}(\boldsymbol{\theta}, \eta_i)$ terms and, in particular, we consider the partial derivative $\nabla_{\eta_i} \mathcal{J}_{A,i}(\boldsymbol{\theta}, \eta_i)$. For it, we can recover the common form:

$$\nabla_{\eta_i} \mathcal{J}_{A,i}(\boldsymbol{\theta}, \eta_i) = \mathbb{E}_{\tau \sim p_A(\cdot|\boldsymbol{\theta})} \left[ \nabla_{\eta_i} f_i(C_i(\tau), \eta_i) \right] + \nabla_{\eta_i} g_i(\eta_i). \tag{67}$$

**Mean Variance.** In this case we have that $f_i(C_i(\tau), \eta_i) = (1 - 2\kappa_i \eta_i)C_i(\tau) + \kappa_i C_i(\tau)^2$ and that $g_i(\eta_i) = \kappa_i \eta_i^2$. Thus, from Equation (67), the following holds:

$$\nabla_{\eta_i} \mathcal{J}_{A,i}(\boldsymbol{\theta}, \eta_i) = \mathbb{E}_{\tau \sim p_A(\cdot|\boldsymbol{\theta})} \left[ -2\kappa_i C_i(\tau) \right] + 2\kappa_i \eta_i. \tag{68}$$

Considering the behavior of `C-PGAE` described in Section 4, we obtain the following sample-based version:

$$\widehat{\nabla}_{\eta_i} \mathcal{J}_{A,i}(\boldsymbol{\theta}, \eta_i) = -\frac{2\kappa_i}{N} \sum_{j=1}^{N} C_i(\tau_j) + 2\kappa_i \eta_i. \tag{69}$$

**Conditional Value at Risk.** In this case we have that $f_i(C_i(\tau), \eta_i) = \frac{1}{1-\alpha_i} \left(C_i(\tau) - \eta_i\right)^+$ and that $g_i(\eta_i) = \eta_i$. Here we face the same issues we have discussed in the corresponding `C-PGPE` part. With the same procedure, we obtain the following partial derivative:

$$\partial_{\eta_i} \mathcal{J}_{A,i}(\boldsymbol{\theta}, \eta_i) = \frac{1}{1-\alpha_i} \mathop{\mathbb{E}}_{\tau \sim p_A(\cdot|\boldsymbol{\theta})} \left[\mathbb{1}\left\{C_i(\tau) \geqslant \eta_i\right\}\right] + 1. \tag{70}$$

Finally, according to the `C-PGAE` behavior described in Section 4, we obtain the following sample-based version:

$$\widehat{\partial}_{\eta_i} \mathcal{J}_{A,i}(\boldsymbol{\theta}, \eta_i) = \frac{1}{N} \sum_{j=1}^{N} \mathbb{1}\left\{C_i(\tau_j) \geqslant \eta_i\right\} + 1. \tag{71}$$

Also in this case, we will use $\widehat{\nabla}_{\eta_i} \mathcal{J}_{A,i}(\boldsymbol{\rho}, \eta_i) = \widehat{\partial}_{\eta_i} \mathcal{J}_{A,i}(\boldsymbol{\rho}, \eta_i)$ for the $\boldsymbol{\eta}$ update in the case in which the $\text{CVaR}_\alpha$ risk measure is employed.

## E  Proofs

**Lemma E.1** (Regularization Bias on Saddle Points - 1). *Under Assumption 3.1, for every $\omega \geqslant 0$, let $(\boldsymbol{v}_\omega^*, \boldsymbol{\lambda}_\omega^*)$ be a saddle point of $\mathcal{L}_\omega$, it holds that:*

$$0 \leqslant \mathcal{L}_0(\boldsymbol{v}_0^*, \boldsymbol{\lambda}_0^*) - \mathcal{L}_0(\boldsymbol{v}_\omega^*, \boldsymbol{\lambda}_\omega^*) \leqslant \frac{\omega}{2} \left(\|\boldsymbol{\lambda}_0^*\|_2^2 - \|\boldsymbol{\lambda}_\omega^*\|_2^2\right).$$

*Proof.* From the fact that $(\boldsymbol{v}_\omega^*, \boldsymbol{\lambda}_\omega^*)$ is a saddle point of $\mathcal{L}_\omega$, we have for every $\boldsymbol{v} \in \mathcal{V}$ and $\boldsymbol{\lambda} \in \Lambda$:

$$\mathcal{L}_\omega(\boldsymbol{v}, \boldsymbol{\lambda}_\omega^*) \geqslant \mathcal{L}_\omega(\boldsymbol{v}_\omega^*, \boldsymbol{\lambda}_\omega^*) \geqslant \mathcal{L}_\omega(\boldsymbol{v}_\omega^*, \boldsymbol{\lambda}) \tag{72}$$

$$\iff \mathcal{L}_0(\boldsymbol{v}, \boldsymbol{\lambda}_\omega^*) - \frac{\omega}{2}\|\boldsymbol{\lambda}_\omega^*\|_2^2 \geqslant \mathcal{L}_0(\boldsymbol{v}_\omega^*, \boldsymbol{\lambda}_\omega^*) - \frac{\omega}{2}\|\boldsymbol{\lambda}_\omega^*\|_2^2 \geqslant \mathcal{L}_0(\boldsymbol{v}_\omega^*, \boldsymbol{\lambda}) - \frac{\omega}{2}\|\boldsymbol{\lambda}\|_2^2 \tag{73}$$

$$\iff \mathcal{L}_0(\boldsymbol{v}, \boldsymbol{\lambda}_\omega^*) \geqslant \mathcal{L}_0(\boldsymbol{v}_\omega^*, \boldsymbol{\lambda}_\omega^*) \geqslant \mathcal{L}_0(\boldsymbol{v}_\omega^*, \boldsymbol{\lambda}) + \frac{\omega}{2}\left(\|\boldsymbol{\lambda}_\omega^*\|_2^2 - \|\boldsymbol{\lambda}\|_2^2\right). \tag{74}$$

From the fact that $(\boldsymbol{v}_0^*, \boldsymbol{\lambda}_0^*)$ is a saddle point of $\mathcal{L}_0$, we have for every $\boldsymbol{v} \in \mathcal{V}$ and $\boldsymbol{\lambda} \in \Lambda$:

$$\mathcal{L}_0(\boldsymbol{v}, \boldsymbol{\lambda}_0^*) \geqslant \mathcal{L}_0(\boldsymbol{v}_0^*, \boldsymbol{\lambda}_0^*) \geqslant \mathcal{L}_0(\boldsymbol{v}_0^*, \boldsymbol{\lambda}). \tag{75}$$

By setting $(\boldsymbol{v}, \boldsymbol{\lambda}) \leftarrow (\boldsymbol{v}_\omega^*, \boldsymbol{\lambda}_\omega^*)$ in Equation (75) and $(\boldsymbol{v}, \boldsymbol{\lambda}) \leftarrow (\boldsymbol{v}_0^*, \boldsymbol{\lambda}_0^*)$ in Equation (74), we obtain:

$$\mathcal{L}_0(\boldsymbol{v}_\omega^*, \boldsymbol{\lambda}_0^*) \geqslant \mathcal{L}_0(\boldsymbol{v}_0^*, \boldsymbol{\lambda}_0^*) \geqslant \mathcal{L}_0(\boldsymbol{v}_0^*, \boldsymbol{\lambda}_\omega^*) \tag{76}$$

$$= \mathcal{L}_0(\boldsymbol{v}_0^*, \boldsymbol{\lambda}_\omega^*) \geqslant \mathcal{L}_0(\boldsymbol{v}_\omega^*, \boldsymbol{\lambda}_\omega^*) \geqslant \mathcal{L}_0(\boldsymbol{v}_\omega^*, \boldsymbol{\lambda}_0^*) + \frac{\omega}{2}\left(\|\boldsymbol{\lambda}_\omega^*\|_2^2 - \|\boldsymbol{\lambda}_0^*\|_2^2\right) \tag{77}$$

$$\geqslant \mathcal{L}_0(\boldsymbol{v}_0^*, \boldsymbol{\lambda}_0^*) + \frac{\omega}{2}\left(\|\boldsymbol{\lambda}_\omega^*\|_2^2 - \|\boldsymbol{\lambda}_0^*\|_2^2\right), \tag{78}$$

thus:

$$\mathcal{L}_0(\boldsymbol{v}_0^*, \boldsymbol{\lambda}_0^*) \geqslant \mathcal{L}_0(\boldsymbol{v}_\omega^*, \boldsymbol{\lambda}_\omega^*) \geqslant \mathcal{L}_0(\boldsymbol{v}_0^*, \boldsymbol{\lambda}_0^*) + \frac{\omega}{2}\left(\|\boldsymbol{\lambda}_\omega^*\|_2^2 - \|\boldsymbol{\lambda}_0^*\|_2^2\right). \tag{79}$$

$\square$

**Lemma E.2** (Regularization Bias on Saddle Points - 2). *Under Assumption 3.1, for every $\omega \geqslant 0$, it holds that:*

$$0 \leqslant \min_{\boldsymbol{v} \in \mathcal{V}} \max_{\boldsymbol{\lambda} \in \Lambda} \mathcal{L}_0(\boldsymbol{v}, \boldsymbol{\lambda}) - \min_{\boldsymbol{v} \in \mathcal{V}} \max_{\boldsymbol{\lambda} \in \Lambda} \mathcal{L}_\omega(\boldsymbol{v}, \boldsymbol{\lambda}) \leqslant \frac{\omega}{2}\|\boldsymbol{\lambda}_0^*\|_2^2.$$

*Proof.* The first inequality follows from the observation that $\mathcal{L}_0(\boldsymbol{v}, \boldsymbol{\lambda}) \geqslant \mathcal{L}_\omega(\boldsymbol{v}, \boldsymbol{\lambda})$ for every $\omega \geqslant 0$. For the second inequality, let us denote as $(\boldsymbol{v}_\omega^*, \boldsymbol{\lambda}_\omega^*)$ the saddle point for $\mathcal{L}_\omega$ and let $\Lambda^* = \{\boldsymbol{\lambda}_0^*, \boldsymbol{\lambda}_\omega^*\}$. We have:

$$\mathcal{L}_0(\boldsymbol{v}_0^*, \boldsymbol{\lambda}_0^*) - \mathcal{L}_\omega(\boldsymbol{v}_\omega^*, \boldsymbol{\lambda}_\omega^*) = \min_{\boldsymbol{v} \in \mathcal{V}} \max_{\boldsymbol{\lambda} \in \Lambda^*} \mathcal{L}_0(\boldsymbol{v}, \boldsymbol{\lambda}) - \min_{\boldsymbol{v} \in \mathcal{V}} \max_{\boldsymbol{\lambda} \in \Lambda^*} \mathcal{L}_\omega(\boldsymbol{v}, \boldsymbol{\lambda}) \tag{80}$$

$$\leqslant \max_{\boldsymbol{v} \in \mathcal{V}} \left| \max_{\boldsymbol{\lambda} \in \Lambda^*} \mathcal{L}_0(\boldsymbol{v}, \boldsymbol{\lambda}) - \max_{\boldsymbol{\lambda} \in \Lambda^*} \mathcal{L}_\omega(\boldsymbol{v}, \boldsymbol{\lambda}) \right| \tag{81}$$

$$= \max_{\boldsymbol{v} \in \mathcal{V}, \boldsymbol{\lambda} \in \Lambda^*} |\mathcal{L}_0(\boldsymbol{v}, \boldsymbol{\lambda}) - \mathcal{L}_\omega(\boldsymbol{v}, \boldsymbol{\lambda})| \tag{82}$$

$$= \frac{\omega}{2} \max \left\{ \|\boldsymbol{\lambda}_0^*\|_2^2; \ \|\boldsymbol{\lambda}_\omega^*\|_2^2 \right\} \tag{83}$$

$$= \frac{\omega}{2} \|\boldsymbol{\lambda}_0^*\|_2^2, \tag{84}$$

where we used Lemma E.1 to conclude that $\|\boldsymbol{\lambda}_0^*\|_2^2 \geqslant \|\boldsymbol{\lambda}_\omega^*\|_2^2$. $\qquad\square$

**Lemma E.3** (Objective bound and Constraint violation). *Under Assumption 3.1, for every $\omega \geqslant 0$, letting $(\boldsymbol{v}_\omega^*, \boldsymbol{\lambda}_\omega^*)$ be a saddle point of $\mathcal{L}_\omega$, it holds that:*

$$0 \leqslant J_0(\boldsymbol{v}_0^*) - J_0(\boldsymbol{v}_\omega^*) \leqslant \omega \|\boldsymbol{\lambda}_0^*\|_2^2, \tag{85}$$

$$\|(\mathbf{J}(\boldsymbol{v}_\omega^*) - \mathbf{b})^+\|_2 \leqslant \omega \|\boldsymbol{\lambda}_0^*\|_2. \tag{86}$$

*Proof.* Since $(\boldsymbol{v}_0^*, \boldsymbol{\lambda}_0^*)$ is a saddle point of $\mathcal{L}_0$, it holds that $\boldsymbol{v}_0^*$ is feasible and, consequently, $\mathcal{L}_0(\boldsymbol{v}_0^*, \boldsymbol{\lambda}_0^*) = J_0(\boldsymbol{v}_0^*)$. Moreover, let $\omega > 0$: since $(\boldsymbol{v}_\omega^*, \boldsymbol{\lambda}_\omega^*)$ is a saddle point of $\mathcal{L}_\omega$ it holds that $\boldsymbol{\lambda}_\omega^* = \boldsymbol{\lambda}^*(\boldsymbol{v}_\omega^*) = \Pi_\Lambda \left( \frac{1}{\omega}(\mathbf{J}(\boldsymbol{v}_\omega^*) - \mathbf{b}) \right) = \frac{1}{\omega}(\mathbf{J}(\boldsymbol{v}_\omega^*) - \mathbf{b})^+$, since $\frac{1}{\omega}\|(\mathbf{J}(\boldsymbol{v}_\omega^*) - \mathbf{b})^+\|_2 \leqslant \omega^{-1}\sqrt{U}J_{\max}$. Thus, we have:

$$\mathcal{L}_0(\boldsymbol{v}_\omega^*, \boldsymbol{\lambda}_\omega^*) = J_0(\boldsymbol{v}_\omega^*) + \langle \boldsymbol{\lambda}_\omega^*, \mathbf{J}(\boldsymbol{v}_\omega^*) - \mathbf{b} \rangle = J_0(\boldsymbol{v}_\omega^*) + \frac{1}{\omega}\|(\mathbf{J}(\boldsymbol{v}_\omega^*) - \mathbf{b})^+\|_2^2. \tag{87}$$

From Lemma E.1, we have:

$$0 \leqslant J_0(\boldsymbol{v}_0^*) - J_0(\boldsymbol{v}_\omega^*) - \frac{1}{\omega}\|(\mathbf{J}(\boldsymbol{v}_\omega^*) - \mathbf{b})^+\|_2^2 \leqslant \frac{\omega}{2}\|\boldsymbol{\lambda}_0^*\|_2^2 - \frac{1}{2\omega}\|(\mathbf{J}(\boldsymbol{v}_\omega^*) - \mathbf{b})^+\|_2^2. \tag{88}$$

By summing $\frac{1}{\omega}\|(\mathbf{J}(\boldsymbol{v}_\omega^*) - \mathbf{b})^+\|_2^2$ to all members, we have:

$$\frac{1}{\omega}\|(\mathbf{J}(\boldsymbol{v}_\omega^*) - \mathbf{b})^+\|_2^2 \leqslant J_0(\boldsymbol{v}_0^*) - J_0(\boldsymbol{v}_\omega^*) \leqslant \frac{\omega}{2}\|\boldsymbol{\lambda}_0^*\|_2^2 + \frac{1}{2\omega}\|(\mathbf{J}(\boldsymbol{v}_\omega^*) - \mathbf{b})^+\|_2^2. \tag{89}$$

Now taking the first and last member, we conclude:

$$\|(\mathbf{J}(\boldsymbol{v}_\omega^*) - \mathbf{b})^+\|_2^2 \leqslant \omega^2 \|\boldsymbol{\lambda}_0^*\|_2^2. \tag{90}$$

Since $\frac{1}{\omega}\|(\mathbf{J}(\boldsymbol{v}_\omega^*) - \mathbf{b})^+\|_2^2 \geqslant 0$ and plugging the latter inequality into the third member of (89) we obtain:

$$0 \leqslant J_0(\boldsymbol{v}_0^*) - J_0(\boldsymbol{v}_\omega^*) \leqslant \omega \|\boldsymbol{\lambda}_0^*\|_2^2. \tag{91}$$

$\qquad\square$

**Lemma E.4** (Weak $\psi$-Gradient Domination on $H_\omega(\boldsymbol{v})$). *Under Assumption 3.2, if $\omega > 0$, for every $\boldsymbol{v} \in \mathcal{V}$ and $\boldsymbol{\lambda} \in \Lambda$, it holds that:*

$$\|\nabla_{\boldsymbol{v}} H_\omega(\boldsymbol{v})\|_2^\psi \geqslant \alpha_1 \left( H_\omega(\boldsymbol{v}) - \min_{\boldsymbol{v}' \in \mathcal{V}} H_\omega(\boldsymbol{v}') \right) - \beta_1. \tag{92}$$

*Proof.* If $\omega > 0$, the dual variable exist finite since the maximization problem over $\boldsymbol{\lambda}$ is concave:

$$\boldsymbol{\lambda}^*(\boldsymbol{v}) = \arg\max_{\boldsymbol{\lambda} \in \Lambda} \mathcal{L}_\omega(\boldsymbol{v}, \boldsymbol{\lambda}).$$

Thus, we have from Lemma E.7 that $\nabla_{\boldsymbol{v}} H_\omega(\boldsymbol{v}) = \nabla_{\boldsymbol{v}} \mathcal{L}_\omega(\boldsymbol{v}, \boldsymbol{\lambda})|_{\boldsymbol{\lambda}=\boldsymbol{\lambda}^*(\boldsymbol{v})}$ and by Assumption 3.2 we have the following:

$$\|\nabla_{\boldsymbol{v}} H_\omega(\boldsymbol{v})\|_2 = \left\| \nabla_{\boldsymbol{v}} \mathcal{L}_\omega(\boldsymbol{v}, \boldsymbol{\lambda})|_{\boldsymbol{\lambda}=\boldsymbol{\lambda}^*(\boldsymbol{v})} \right\|_2 \tag{93}$$

$$\geqslant \alpha_1 \left( \mathcal{L}_\omega(\boldsymbol{v}, \boldsymbol{\lambda}^*(\boldsymbol{v})) - \min_{\boldsymbol{v}' \in \mathcal{V}} \mathcal{L}_\omega(\boldsymbol{v}', \boldsymbol{\lambda}^*(\boldsymbol{v})) \right) - \beta_1 \tag{94}$$

$$\geqslant \alpha_1 \left( H_\omega(\boldsymbol{v}) - \min_{\boldsymbol{v}' \in \mathcal{V}} \max_{\boldsymbol{\lambda} \in \Lambda} \mathcal{L}_\omega(\boldsymbol{v}', \boldsymbol{\lambda}) \right) - \beta_1 \tag{95}$$

$$= \alpha_1 (H_\omega(\boldsymbol{v}) - H_\omega^*) - \beta_1. \tag{96}$$

$\qquad\square$

**Lemma E.5.** *Let $\omega > 0$ and $\boldsymbol{v} \in \mathcal{V}$. The following statements hold:*

- $\mathcal{L}_\omega(\boldsymbol{v}, \cdot)$ is $\omega$-smooth, i.e., for every $\boldsymbol{\lambda}, \boldsymbol{\lambda}' \in \Lambda$ it holds that:
$$\left|\nabla_{\boldsymbol{\lambda}}\mathcal{L}_\omega(\boldsymbol{v}, \boldsymbol{\lambda}') - \nabla_{\boldsymbol{\lambda}}\mathcal{L}_\omega(\boldsymbol{v}, \boldsymbol{\lambda})\right| \leqslant \omega \left\|\boldsymbol{\lambda} - \boldsymbol{\lambda}'\right\|_2^2$$

- $\mathcal{L}_\omega(\boldsymbol{v}, \cdot)$ satisfies the PL condition, i.e., for every $\boldsymbol{\lambda} \in \Lambda$ it holds that:
$$\|\nabla_{\boldsymbol{\lambda}}\mathcal{L}_\omega(\boldsymbol{v}, \boldsymbol{\lambda})\|_2^2 \geqslant \omega \left(\max_{\boldsymbol{\lambda}' \in \Lambda} \mathcal{L}_\omega(\boldsymbol{v}, \boldsymbol{\lambda}') - \mathcal{L}_\omega(\boldsymbol{v}, \boldsymbol{\lambda})\right).$$

- $\mathcal{L}_\omega(\boldsymbol{v}, \cdot)$ satisfies the error bound *(EB)* condition, i.e., for every $\boldsymbol{\lambda}, \boldsymbol{\lambda}' \in \Lambda$ it holds that:
$$\|\nabla_{\boldsymbol{\lambda}}\mathcal{L}_\omega(\boldsymbol{v}, \boldsymbol{\lambda})\| \geqslant \frac{\omega}{2}\|\boldsymbol{\lambda}^*(\boldsymbol{v}) - \boldsymbol{\lambda}\|_2,$$
where $\boldsymbol{\lambda}^*(\boldsymbol{v}) = \arg\max_{\boldsymbol{\lambda} \in \Lambda} \mathcal{L}_\omega(\boldsymbol{v}, \boldsymbol{\lambda})$.

- $\mathcal{L}_\omega(\boldsymbol{v}, \cdot)$ satisfies the quadratic growth *(QG)* condition, i.e., for every $\boldsymbol{\lambda}, \boldsymbol{\lambda}' \in \Lambda$ it holds that:
$$H_\omega(\boldsymbol{v}) - \mathcal{L}_\omega(\boldsymbol{v}, \boldsymbol{\lambda}) \geqslant \frac{\omega}{4}\|\boldsymbol{\lambda}^*(\boldsymbol{v}) - \boldsymbol{\lambda}\|_2,$$
where $\boldsymbol{\lambda}^*(\boldsymbol{v}) = \arg\max_{\boldsymbol{\lambda} \in \Lambda} \mathcal{L}_\omega(\boldsymbol{v}, \boldsymbol{\lambda})$.

*Proof.* For the first property, it is enough to observe that $\mathcal{L}_\omega$ is twice differentiable in $\boldsymbol{\lambda}$ and that its Hessian is $\omega\mathbf{I}$. For the second property, we observe that $\mathcal{L}_\omega$ is quadratic in $\boldsymbol{\lambda}$ and, consequently it satisfies the PL condition with parameter $\omega$:

$$\|\nabla_{\boldsymbol{\lambda}}\mathcal{L}_\omega(\boldsymbol{v}, \boldsymbol{\lambda})\|_2^2 \geqslant \omega \left(\max_{\boldsymbol{\lambda}' \in \mathbb{R}^U} \mathcal{L}_\omega(\boldsymbol{v}, \boldsymbol{\lambda}') - \mathcal{L}_\omega(\boldsymbol{v}, \boldsymbol{\lambda})\right) \geqslant \omega \left(\max_{\boldsymbol{\lambda}' \in \Lambda} \mathcal{L}_\omega(\boldsymbol{v}, \boldsymbol{\lambda}') - \mathcal{L}_\omega(\boldsymbol{v}, \boldsymbol{\lambda})\right).$$

For the third and fourth properties, we refer to Lemma A.1 of Yang et al. (2020). $\square$

**Lemma E.6.** *Let $\omega > 0$. For every $\boldsymbol{v} \in \mathcal{V}$, it holds that:*
$$H_\omega(\boldsymbol{v}) - H_\omega^* \geqslant \frac{\omega}{4}\|\boldsymbol{\lambda}^*(\boldsymbol{v}) - \boldsymbol{\lambda}_\omega^*\|_2. \tag{97}$$

*Proof.* Let us consider the following derivation:
$$H_\omega(\boldsymbol{v}) - H_\omega^* = H_\omega(\boldsymbol{v}) - \mathcal{L}_\omega(\boldsymbol{v}_\omega^*, \boldsymbol{\lambda}_\omega^*) \tag{98}$$
$$\geqslant H_\omega(\boldsymbol{v}) - \mathcal{L}_\omega(\boldsymbol{v}, \boldsymbol{\lambda}_\omega^*) \tag{99}$$
$$\geqslant \frac{\omega}{4}\|\boldsymbol{\lambda}^*(\boldsymbol{v}) - \boldsymbol{\lambda}_\omega^*\|_2. \tag{100}$$
having exploited the fact that, from the saddle point property, $\mathcal{L}_\omega(\boldsymbol{v}_\omega^*, \boldsymbol{\lambda}_\omega^*) \leqslant \mathcal{L}_\omega(\boldsymbol{v}, \boldsymbol{\lambda}_\omega^*)$ and, then, Lemma E.5. $\square$

**Lemma E.7.** *Let $\omega > 0$. The following statements hold:*

- $H_\omega$ is $L_H$-smooth, i.e., for every $\boldsymbol{v}, \boldsymbol{v}' \in \mathcal{V}$, it holds that:
$$\|\nabla_{\boldsymbol{v}}H_\omega(\boldsymbol{v}') - \nabla_{\boldsymbol{v}}H_\omega(\boldsymbol{v})\|_2 \leqslant L_H\|\boldsymbol{v}' - \boldsymbol{v}\|_2.$$
where $L_H := L_2 + \frac{L_1^2}{\omega}$.

- *For every $\boldsymbol{v}, \boldsymbol{v}' \in \mathcal{V}$ we have $\nabla_{\boldsymbol{v}}H_\omega(\boldsymbol{v}) = \nabla_{\boldsymbol{v}}\mathcal{L}_\omega(\boldsymbol{v}, \boldsymbol{\lambda})|_{\boldsymbol{\lambda}=\boldsymbol{\lambda}^*(\boldsymbol{v})}$, where $\boldsymbol{\lambda}^*(\boldsymbol{v}) = \arg\max_{\boldsymbol{\lambda} \in \Lambda} \mathcal{L}_\omega(\boldsymbol{v}, \boldsymbol{\lambda})$.*

*Proof.* The first and second statements follow from Lemma A.5 of Nouiehed et al. (2019). $\square$

**Theorem 3.1** (Objective Function Gap and Constraint Violation). *Let $\epsilon > 0$. Under Assumption 3.1, if $\mathcal{P}_k \leqslant \epsilon$, it holds that:*

$$\mathbb{E}[J_0(\boldsymbol{v}_k) - J_0(\boldsymbol{v}_0^*)] \leqslant \epsilon + \frac{\omega}{2}\|\boldsymbol{\lambda}_0^*\|_2^2, \qquad \mathbb{E}[(J_i(\boldsymbol{v}_k) - b_i)^+] \leqslant 4\epsilon + \omega\|\boldsymbol{\lambda}_0^*\|_2, \quad \forall i \in [\![U]\!]. \tag{8}$$

*Proof.* Since $\mathcal{P}_k \leqslant \epsilon$, it follows that $a_k \leqslant \epsilon$ and, consequently, $0 \leqslant \mathbb{E}[H_\omega(\boldsymbol{v}_k) - H_\omega^*] \leqslant \epsilon$. We start by bounding the norm of the dual variables:
$$\|\boldsymbol{\lambda}^*(\boldsymbol{v}_k)\|_2 \leqslant \|\boldsymbol{\lambda}_\omega^*\|_2 + \|\boldsymbol{\lambda}^*(\boldsymbol{v}_k) - \boldsymbol{\lambda}_\omega^*\|_2 \tag{101}$$

$$\leqslant \|\boldsymbol{\lambda}_\omega^*\|_2 + \frac{4}{\omega}(H_\omega(\boldsymbol{v}_k) - H_\omega^*). \tag{102}$$

where we applied the triangular inequality and Lemma E.6. The projection $\Pi_\Lambda$ is such that $\boldsymbol{\lambda}^*(\boldsymbol{v}) = \Pi_\Lambda\left(\frac{1}{\omega}(\mathbf{J}(\boldsymbol{v}) - \mathbf{b})\right) = \frac{1}{\omega}(\mathbf{J}(\boldsymbol{v}) - \mathbf{b})^+$ and, consequently, we have:

$$\|(\mathbf{J}(\boldsymbol{v}_k) - \mathbf{b})^+\|_2 - \|(\mathbf{J}(\boldsymbol{v}_\omega^*) - \mathbf{b})^+\|_2 \leqslant 4(H_\omega(\boldsymbol{v}_k) - H_\omega^*). \tag{103}$$

By the last inequality, together with Lemma E.3, having applied the expectation on both sides:

$$\mathbb{E}[\|(\mathbf{J}(\boldsymbol{v}_k) - \mathbf{b})^+\|_2] \leqslant \|(\mathbf{J}(\boldsymbol{v}_\omega^*) - \mathbf{b})^+\|_2 + 4\,\mathbb{E}[H_\omega(\boldsymbol{v}_k) - H_\omega^*] \tag{104}$$

$$\leqslant \omega\|\boldsymbol{\lambda}_0^*\|_2 + 4\epsilon. \tag{105}$$

Recalling that:

$$\mathbb{E}[\|(\mathbf{J}(\boldsymbol{v}_k) - \mathbf{b}))^+\|_2] \geqslant \left\|\mathbb{E}[(\mathbf{J}(\boldsymbol{v}_k) - \mathbf{b})^+]\right\|_2 \geqslant \|\mathbb{E}[(\mathbf{J}(\boldsymbol{v}_k) - \mathbf{b})^+]\|_\infty. \tag{106}$$

For the objective function bound, let us consider the following derivation. By definition of $H_\omega(\boldsymbol{v})$ and $\boldsymbol{\lambda}^*(\boldsymbol{v})$ we have:

$$J_0(\boldsymbol{v}_k) - J_0(\boldsymbol{v}_\omega^*) = H_\omega(\boldsymbol{v}_k) - H_\omega^* - \frac{\omega}{2}\left(\|\boldsymbol{\lambda}^*(\boldsymbol{v}_k)\|_2^2 - \|\boldsymbol{\lambda}_\omega^*\|_2^2\right). \tag{107}$$

Taking the expectation on both sides and upper bounding $\|\boldsymbol{\lambda}_\omega^*\|$ with $\|\boldsymbol{\lambda}_0^*\|$ from Lemma E.1:

$$\mathbb{E}[J_0(\boldsymbol{v}_k) - J_0(\boldsymbol{v}_\omega^*)] = \mathbb{E}[H_\omega(\boldsymbol{v}_k) - H_\omega^*] - \frac{\omega}{2}\,\mathbb{E}[\|\boldsymbol{\lambda}^*(\boldsymbol{v}_k)\|_2^2 - \|\boldsymbol{\lambda}_\omega^*\|_2^2] \tag{108}$$

$$\leqslant \mathbb{E}[H_\omega(\boldsymbol{v}_k) - H_\omega^*] + \frac{\omega}{2}\|\boldsymbol{\lambda}_\omega^*\|_2^2 \tag{109}$$

$$\leqslant \epsilon + \frac{\omega}{2}\|\boldsymbol{\lambda}_0^*\|_2^2. \tag{110}$$

The result is obtained by applying Lemma E.3 as follows:

$$\mathbb{E}[J_0(\boldsymbol{v}_k) - J_0(\boldsymbol{v}_0^*)] = \mathbb{E}[J_0(\boldsymbol{v}_k) - J_0(\boldsymbol{v}_\omega^*)] + \underbrace{J_0(\boldsymbol{v}_\omega^*) - J_0(\boldsymbol{v}_0^*)}_{\leqslant 0}. \tag{111}$$

$\square$

**Theorem 3.2** (Convergence of $\mathcal{P}_K$). *Under Assumptions 3.2, 3.3, 3.4, for $\chi < 1/5$, sufficiently small $\epsilon$ and $\omega$, and a choice of* constant *learning rates $\zeta_{\boldsymbol{v}}, \zeta_{\boldsymbol{\lambda}}$, we have $\mathcal{P}_K(\chi) \leqslant \epsilon + \beta_1/\alpha_1$ whenever:*[8]

- $K = \mathcal{O}(\omega^{-1}\log(\epsilon^{-1}))$ *if $\psi = 2$ and the gradients are exact (i.e., $V_{\boldsymbol{v}} = V_{\boldsymbol{\lambda}} = 0$);*
- $K = \mathcal{O}(\omega^{-1}\epsilon^{-\frac{2}{\psi}-1})$ *if $\psi \in [1,2)$ and the gradients are exact (i.e., $V_{\boldsymbol{v}} = V_{\boldsymbol{\lambda}} = 0$);*
- $K = \mathcal{O}(\omega^{-3}\epsilon^{-\frac{4}{\psi}+1})$ *if $\psi \in [1,2]$ and the gradients are estimated (i.e., $V_{\boldsymbol{v}} = \mathcal{O}(\omega^{-2})$ and $V_{\boldsymbol{\lambda}} = \mathcal{O}(1)$).*

*Proof.* The proof is subdivided into several parts. We will omit the $\omega$ subscript for notational easiness. Let us focus on a specific iteration $k \in \mathbb{N}$.

**Part I: bounding the $a_k$ term.** Let us start with the $a_k$ term:

$$H_\omega(\boldsymbol{v}_{k+1}) - H^* \leqslant H_\omega(\boldsymbol{v}_k) - H^* + \langle \boldsymbol{v}_{k+1} - \boldsymbol{v}_k, \nabla_{\boldsymbol{v}} H_\omega(\boldsymbol{v}_k)\rangle + \frac{L_H}{2}\|\boldsymbol{v}_{k+1} - \boldsymbol{v}_k\|_2^2 \tag{112}$$

$$\leqslant H_{\boldsymbol{v}}(\boldsymbol{v}_k) - H^* - \zeta_{\boldsymbol{v},k}\left\langle \hat{\nabla}_{\boldsymbol{v}}\mathcal{L}_\omega(\boldsymbol{v}_k, \boldsymbol{\lambda}_k), \nabla_{\boldsymbol{v}} H_\omega(\boldsymbol{v}_k)\right\rangle \tag{113}$$

$$+ \frac{L_H}{2}\zeta_{\boldsymbol{v},k}^2\left\|\hat{\nabla}_{\boldsymbol{v}}\mathcal{L}_\omega(\boldsymbol{v}_k, \boldsymbol{\lambda}_k)\right\|_2^2, \tag{114}$$

where the first line is due to the fact that the function $H_\omega$ is $L_H$-smooth (Lemma E.7), the last inequality is due to the update rule of $\boldsymbol{v}$. Now, we apply the expected value on both sides of the inequality and we use the fact that the gradient estimation is unbiased and has variance bounded by $V_{\boldsymbol{v}}$:

$$\mathbb{E}\left[H_\omega(\boldsymbol{v}_{k+1})|\mathcal{F}_{k-1}\right] - H^* \leqslant H_\omega(\boldsymbol{v}_k) - H^* - \zeta_{\boldsymbol{v},k}\langle \nabla_{\boldsymbol{v}}\mathcal{L}_\omega(\boldsymbol{v}_k, \boldsymbol{\lambda}_k), \nabla_{\boldsymbol{v}} H_\omega(\boldsymbol{v}_k)\rangle \tag{115}$$

$$+ \frac{L_H}{2}\zeta_{\boldsymbol{v},k}^2\,\mathbb{E}\left[\left\|\hat{\nabla}_{\boldsymbol{v}}\mathcal{L}_\omega(\boldsymbol{v}_k, \boldsymbol{\lambda}_k)\right\|_2^2|\mathcal{F}_{k-1}\right], \tag{116}$$

---

[8]In the context of this statement, the $\mathcal{O}(\cdot)$ notation preserves dependences on $\epsilon$ and $\omega$ only.

where $\mathcal{F}_{k-1}$ is the filtration associated with all events realized up to interaction $k-1$. We recall that:

$$\mathbb{E}\left[\left\|\widehat{\nabla}_{\boldsymbol{v}}\mathcal{L}_{\omega}(\boldsymbol{v}_k, \boldsymbol{\lambda}_k)\right\|_2^2 \Big| \mathcal{F}_{k-1}\right] = \mathbb{V}\mathrm{ar}\left[\widehat{\nabla}_{\boldsymbol{v}}\mathcal{L}_{\omega}(\boldsymbol{v}_k, \boldsymbol{\lambda}_k)|\mathcal{F}_{k-1}\right] + \|\nabla_{\boldsymbol{v}}\mathcal{L}_{\omega}(\boldsymbol{v}_k, \boldsymbol{\lambda}_k)\|_2^2, \quad (117)$$

and that $\mathbb{V}\mathrm{ar}\left[\widehat{\nabla}_{\boldsymbol{v}}\mathcal{L}_{\omega}(\boldsymbol{v}_k, \boldsymbol{\lambda}_k)\right] \leqslant V_{\boldsymbol{v}}$ by Assumption 3.4. Thus, selecting $\zeta_{\boldsymbol{v},k} \leqslant 1/L_H$, we have that:

$$\mathbb{E}\left[H_{\omega}(\boldsymbol{v}_{k+1})|\mathcal{F}_{k-1}\right] - H^* \tag{118}$$

$$\leqslant H_{\omega}(\boldsymbol{v}_k) - H^* - \zeta_{\boldsymbol{v},k}\left\langle \nabla_{\boldsymbol{v}}\mathcal{L}_{\omega}(\boldsymbol{v}_k, \boldsymbol{\lambda}_k)\,\nabla_{\boldsymbol{v}}H_{\omega}(\boldsymbol{v}_k)\right\rangle \tag{119}$$

$$+ \frac{\zeta_{\boldsymbol{v},k}}{2}\|\nabla_{\boldsymbol{v}}\mathcal{L}_{\omega}(\boldsymbol{v}_k, \boldsymbol{\lambda}_k)\|_2^2 + \frac{L_H}{2}\zeta_{\boldsymbol{v},k}^2 V_{\boldsymbol{v}} \tag{120}$$

$$= H_{\omega}(\boldsymbol{v}_k) - H^* - \zeta_{\boldsymbol{v},k}\left\langle \nabla_{\boldsymbol{v}}\mathcal{L}_{\omega}(\boldsymbol{v}_k, \boldsymbol{\lambda}_k)\,\nabla_{\boldsymbol{v}}H_{\omega}(\boldsymbol{v}_k)\right\rangle \tag{121}$$

$$+ \frac{\zeta_{\boldsymbol{v},k}}{2}\|\nabla_{\boldsymbol{v}}\mathcal{L}_{\omega}(\boldsymbol{v}_k, \boldsymbol{\lambda}_k) \pm \nabla_{\boldsymbol{v}}H_{\omega}(\boldsymbol{v}_k)\|_2^2 + \frac{L_H}{2}\zeta_{\boldsymbol{v},k}^2 V_{\boldsymbol{v}}. \tag{122}$$

Consider that:

$$\frac{\zeta_{\boldsymbol{v},k}}{2}\|\nabla_{\boldsymbol{v}}\mathcal{L}_{\omega}(\boldsymbol{v}_k, \boldsymbol{\lambda}_k) - \nabla_{\boldsymbol{v}}H_{\omega}(\boldsymbol{v}_k) + \nabla_{\boldsymbol{v}}H_{\omega}(\boldsymbol{v}_k)\|_2^2 \tag{123}$$

$$= \frac{\zeta_{\boldsymbol{v},k}}{2}\|\nabla_{\boldsymbol{v}}\mathcal{L}_{\omega}(\boldsymbol{v}_k, \boldsymbol{\lambda}_k) - \nabla_{\boldsymbol{v}}H_{\omega}(\boldsymbol{v}_k)\|_2^2 - \frac{\zeta_{\boldsymbol{v},k}}{2}\|\nabla_{\boldsymbol{v}}H_{\omega}(\boldsymbol{v}_k)\|_2^2 \tag{124}$$

$$+ \zeta_{\boldsymbol{v},k}\left\langle \nabla_{\boldsymbol{v}}\mathcal{L}_{\omega}(\boldsymbol{v}_k, \boldsymbol{\lambda}_k),\ \nabla_{\boldsymbol{v}}H_{\omega}(\boldsymbol{v}_k)\right\rangle. \tag{125}$$

Thus, the following holds:

$$\mathbb{E}\left[H_{\omega}(\boldsymbol{v}_{k+1})|\mathcal{F}_{k-1}\right] - H^* \tag{126}$$

$$\leqslant H_{\omega}(\boldsymbol{v}_k) - H^* - \zeta_{\boldsymbol{v},k}\left\langle \nabla_{\boldsymbol{v}}\mathcal{L}_{\omega}(\boldsymbol{v}_k, \boldsymbol{\lambda}_k),\ \nabla_{\boldsymbol{v}}H_{\omega}(\boldsymbol{v}_k)\right\rangle + \frac{L_H}{2}\zeta_{\boldsymbol{v},k}^2 V_{\boldsymbol{v}} \tag{127}$$

$$+ \frac{\zeta_{\boldsymbol{v},k}}{2}\|\nabla_{\boldsymbol{v}}\mathcal{L}_{\omega}(\boldsymbol{v}_k, \boldsymbol{\lambda}_k) - \nabla_{\boldsymbol{v}}H_{\omega}(\boldsymbol{v}_k)\|_2^2 - \frac{\zeta_{\boldsymbol{v},k}}{2}\|\nabla_{\boldsymbol{v}}H_{\omega}(\boldsymbol{v}_k)\|_2^2 \tag{128}$$

$$+ \zeta_{\boldsymbol{v},k}\left\langle \nabla_{\boldsymbol{v}}\mathcal{L}_{\omega}(\boldsymbol{v}_k, \boldsymbol{\lambda}_k),\ \nabla_{\boldsymbol{v}}H_{\omega}(\boldsymbol{v}_k)\right\rangle \tag{129}$$

$$= H_{\omega}(\boldsymbol{v}_k) - H^* - \frac{\zeta_{\boldsymbol{v},k}}{2}\|\nabla_{\boldsymbol{v}}H_{\omega}(\boldsymbol{v}_k)\|_2^2 + \frac{\zeta_{\boldsymbol{v},k}}{2}\|\nabla_{\boldsymbol{v}}\mathcal{L}_{\omega}(\boldsymbol{v}_k, \boldsymbol{\lambda}_k) - \nabla_{\boldsymbol{v}}H_{\omega}(\boldsymbol{v}_k)\|_2^2 \tag{130}$$

$$+ \frac{L_H}{2}\zeta_{\boldsymbol{v},k}^2 V_{\boldsymbol{v}}. \tag{131}$$

Thus, we have obtained:

$$Ⓐ := \mathbb{E}\left[H_{\omega}(\boldsymbol{v}_{k+1})|\mathcal{F}_{k-1}\right] - H^* \tag{132}$$

$$\leqslant H_{\omega}(\boldsymbol{v}_k) - H^* - \frac{\zeta_{\boldsymbol{v},k}}{2}\|\nabla_{\boldsymbol{v}}H_{\omega}(\boldsymbol{v}_k)\|_2^2 + \frac{\zeta_{\boldsymbol{v},k}}{2}\|\nabla_{\boldsymbol{v}}\mathcal{L}_{\omega}(\boldsymbol{v}_k, \boldsymbol{\lambda}_k) - \nabla_{\boldsymbol{v}}H_{\omega}(\boldsymbol{v}_k)\|_2^2 \tag{133}$$

$$+ \frac{L_H}{2}\zeta_{\boldsymbol{v},k}^2 V_{\boldsymbol{v}}, \tag{134}$$

holding via the selection of $\zeta_{\boldsymbol{v},k} \leqslant 1/L_H$. Notice that, from Ⓐ the following directly follows:

$$Ⓓ := \mathbb{E}\left[H_{\omega}(\boldsymbol{v}_{k+1})|\mathcal{F}_{k-1}\right] - H_{\omega}(\boldsymbol{v}_k) \tag{135}$$

$$\leqslant -\frac{\zeta_{\boldsymbol{v},k}}{2}\|\nabla_{\boldsymbol{v}}H_{\omega}(\boldsymbol{v}_k)\|_2^2 + \frac{\zeta_{\boldsymbol{v},k}}{2}\|\nabla_{\boldsymbol{v}}\mathcal{L}_{\omega}(\boldsymbol{v}_k, \boldsymbol{\lambda}_k) - \nabla_{\boldsymbol{v}}H_{\omega}(\boldsymbol{v}_k)\|_2^2 + \frac{L_H}{2}\zeta_{\boldsymbol{v},k}^2 V_{\boldsymbol{v}}. \tag{136}$$

**Part II: bounding the $b_k$ term.** We are ready to analyze the $b_k$ term. Recall that for ridge regularization of the Lagrangian function presented in the main paper, we have that $\mathcal{L}_{\omega}$ is $\omega$-smooth and fulfills the PL condition with constant $\omega$, as shown in Lemma E.5. Since $\mathcal{L}$ is a quadratic function of $\boldsymbol{\lambda}$ and $\boldsymbol{\lambda}^*(\boldsymbol{v}_{k+1}) \in \Lambda$, we have that considering the non-projected $\boldsymbol{\lambda}_{k+1}$ can only increase the distance. Thus, we will ignore projection for the rest of the proof. We have:

$$H_{\omega}(\boldsymbol{v}_{k+1}) - \mathcal{L}_{\omega}(\boldsymbol{v}_{k+1}, \boldsymbol{\lambda}_{k+1}) \tag{137}$$

$$\leqslant H_{\omega}(\boldsymbol{v}_{k+1}) - \mathcal{L}_{\omega}(\boldsymbol{v}_{k+1}, \boldsymbol{\lambda}_k) - \left\langle \boldsymbol{\lambda}_{k+1} - \boldsymbol{\lambda}_k,\ \nabla_{\boldsymbol{\lambda}}\mathcal{L}_{\omega}(\boldsymbol{v}_{k+1}, \boldsymbol{\lambda}_k)\right\rangle + \frac{\omega}{2}\|\boldsymbol{\lambda}_{k+1} - \boldsymbol{\lambda}_k\|_2^2 \tag{138}$$

$$= H_{\omega}(\boldsymbol{v}_{k+1}) - \mathcal{L}_{\omega}(\boldsymbol{v}_{k+1}, \boldsymbol{\lambda}_k) - \zeta_{\boldsymbol{\lambda},k}\left\langle \widehat{\nabla}_{\boldsymbol{\lambda}}\mathcal{L}_{\omega}(\boldsymbol{v}_{k+1}, \boldsymbol{\lambda}_k),\ \nabla_{\boldsymbol{\lambda}}\mathcal{L}_{\omega}(\boldsymbol{v}_{k+1}, \boldsymbol{\lambda}_k)\right\rangle \tag{139}$$

$$+ \frac{\omega}{2} \zeta_{\boldsymbol{\lambda},k}^2 \left\| \widehat{\nabla}_{\boldsymbol{\lambda}} \mathcal{L}_\omega(\boldsymbol{v}_{k+1}, \boldsymbol{\lambda}_k) \right\|_2^2, \tag{140}$$

that is possible under Assumption 3.3 (i.e., $\mathcal{L}_\omega$ is $L_2$-smooth) and due to the update rules we are considering. Now, by applying the expectation on both sides, we obtain the following:

$$\mathbb{E}\left[ H_\omega(\boldsymbol{v}_{k+1}) - \mathcal{L}_\omega(\boldsymbol{v}_{k+1}, \boldsymbol{\lambda}_{k+1}) | \mathcal{F}_{k-1} \right] \tag{141}$$

$$\leqslant \mathbb{E}\left[ H_\omega(\boldsymbol{v}_{k+1}) - \mathcal{L}_\omega(\boldsymbol{v}_{k+1}, \boldsymbol{\lambda}_k) | \mathcal{F}_{k-1} \right] - \zeta_{\boldsymbol{\lambda},k} \left\| \nabla_{\boldsymbol{\lambda}} \mathcal{L}_\omega(\boldsymbol{v}_{k+1}, \boldsymbol{\lambda}_k) \right\|_2^2 \tag{142}$$

$$+ \frac{\omega}{2} \zeta_{\boldsymbol{\lambda},k}^2 \mathbb{E}\left[ \left\| \widehat{\nabla}_{\boldsymbol{\lambda}} \mathcal{L}_\omega(\boldsymbol{v}_{k+1}, \boldsymbol{\lambda}_k) \right\|_2^2 | \mathcal{F}_{k-1} \right] \tag{143}$$

$$\leqslant \mathbb{E}\left[ H_\omega(\boldsymbol{v}_{k+1}) - \mathcal{L}_\omega(\boldsymbol{v}_{k+1}, \boldsymbol{\lambda}_k) | \mathcal{F}_{k-1} \right] - \zeta_{\boldsymbol{\lambda},k} \left\| \nabla_{\boldsymbol{\lambda}} \mathcal{L}_\omega(\boldsymbol{v}_{k+1}, \boldsymbol{\lambda}_k) \right\|_2^2 + \frac{\omega}{2} \zeta_{\boldsymbol{\lambda},k}^2 V_{\boldsymbol{\lambda}} \tag{144}$$

$$+ \frac{\omega}{2} \zeta_{\boldsymbol{\lambda},k}^2 \left\| \widehat{\nabla}_{\boldsymbol{\lambda}} \mathcal{L}_\omega(\boldsymbol{v}_{k+1}, \boldsymbol{\lambda}) \right\|_2^2 \tag{145}$$

$$\leqslant \mathbb{E}\left[ H_\omega(\boldsymbol{v}_{k+1}) - \mathcal{L}_\omega(\boldsymbol{v}_{k+1}, \boldsymbol{\lambda}_k) | \mathcal{F}_{k-1} \right] - \frac{\zeta_{\boldsymbol{\lambda},k}}{2} \left\| \nabla_{\boldsymbol{\lambda}} \mathcal{L}_\omega(\boldsymbol{v}_{k+1}, \boldsymbol{\lambda}_k) \right\|_2^2 + \frac{\omega}{2} \zeta_{\boldsymbol{\lambda},k}^2 V_{\boldsymbol{\lambda}}, \tag{146}$$

where the last line follows by selecting $\zeta_{\boldsymbol{\lambda},k} \leqslant 1/\omega$. Since $\mathcal{L}_\omega$ enjoys the PL condition w.r.t $\boldsymbol{\lambda}$ with constant $\omega$, for every pair $(\boldsymbol{v}, \boldsymbol{\lambda})$ we have:

$$\left\| \nabla_{\boldsymbol{\lambda}} \mathcal{L}_\omega(\boldsymbol{v}, \boldsymbol{\lambda}) \right\|_2^2 \geqslant \omega \left( \max_{\overline{\boldsymbol{\lambda}} \in \mathbb{R}^U} \mathcal{L}_\omega(\boldsymbol{v}, \overline{\boldsymbol{\lambda}}) - \mathcal{L}_\omega(\boldsymbol{v}, \boldsymbol{\lambda}) \right) \geqslant \omega \left( \max_{\overline{\boldsymbol{\lambda}} \in \Lambda} \mathcal{L}_\omega(\boldsymbol{v}, \overline{\boldsymbol{\lambda}}) - \mathcal{L}_\omega(\boldsymbol{v}, \boldsymbol{\lambda}) \right). \tag{147}$$

By applying the PL condition:

$$\mathbb{E}\left[ H_\omega(\boldsymbol{v}_{k+1}) - \mathcal{L}_\omega(\boldsymbol{v}_{k+1}, \boldsymbol{\lambda}_{k+1}) | \mathcal{F}_{k-1} \right] \tag{148}$$

$$\leqslant \mathbb{E}\left[ H_\omega(\boldsymbol{v}_{k+1}) - \mathcal{L}_\omega(\boldsymbol{v}_{k+1}, \boldsymbol{\lambda}_k) | \mathcal{F}_{k-1} \right] - \frac{\zeta_{\boldsymbol{\lambda},k}}{2} \left\| \nabla_{\boldsymbol{\lambda}} \mathcal{L}_\omega(\boldsymbol{v}_{k+1}, \boldsymbol{\lambda}_k) \right\|_2^2 + \frac{\omega}{2} \zeta_{\boldsymbol{\lambda},k}^2 V_{\boldsymbol{\lambda}} \tag{149}$$

$$\leqslant \mathbb{E}\left[ H_\omega(\boldsymbol{v}_{k+1}) - \mathcal{L}_\omega(\boldsymbol{v}_{k+1}, \boldsymbol{\lambda}_k) | \mathcal{F}_{k-1} \right] - \frac{\zeta_{\boldsymbol{\lambda},k}}{2} \omega \, \mathbb{E}\left[ H_\omega(\boldsymbol{v}_{k+1}) - \mathcal{L}_\omega(\boldsymbol{v}_{k+1}, \boldsymbol{\lambda}_k) | \mathcal{F}_{k-1} \right] \tag{150}$$

$$+ \frac{\omega}{2} \zeta_{\boldsymbol{\lambda},k}^2 V_{\boldsymbol{\lambda}} \tag{151}$$

$$= \left( 1 - \frac{\zeta_{\boldsymbol{\lambda},k}}{2} \omega \right) \mathbb{E}\left[ H_\omega(\boldsymbol{v}_{k+1}) - \mathcal{L}_\omega(\boldsymbol{v}_{k+1}, \boldsymbol{\lambda}_k) | \mathcal{F}_{k-1} \right] + \frac{\omega}{2} \zeta_{\boldsymbol{\lambda},k}^2 V_{\boldsymbol{\lambda}}, \tag{152}$$

where we enforce $1 - \frac{\zeta_{\boldsymbol{\lambda},k}}{2} \omega \geqslant 0$, i.e., $\zeta_{\boldsymbol{\lambda},k} \leqslant 2/\omega$. However, we do not have a proper recursive term, thus consider the following:

$$H_\omega(\boldsymbol{v}_{k+1}) - \mathcal{L}_\omega(\boldsymbol{v}_{k+1}, \boldsymbol{\lambda}_k) = \underbrace{H_\omega(\boldsymbol{v}_k) - \mathcal{L}_\omega(\boldsymbol{v}_k, \boldsymbol{\lambda}_k)}_{\text{Recursive Term}} + \underbrace{\mathcal{L}_\omega(\boldsymbol{v}_k, \boldsymbol{\lambda}_k) - \mathcal{L}_\omega(\boldsymbol{v}_{k+1}, \boldsymbol{\lambda}_k)}_{\text{\textcircled{C}}} \tag{153}$$

$$+ \underbrace{H_\omega(\boldsymbol{v}_{k+1}) - H_\omega(\boldsymbol{v}_k)}_{\text{\textcircled{D}}}. \tag{154}$$

A bound on \textcircled{D} has already been derived, so let us bound the term \textcircled{C}:

$$\mathcal{L}_\omega(\boldsymbol{v}_k, \boldsymbol{\lambda}_k) - \mathcal{L}_\omega(\boldsymbol{v}_{k+1}, \boldsymbol{\lambda}_k) \leqslant - \langle \boldsymbol{v}_{k+1} - \boldsymbol{v}_k, \ \nabla_{\boldsymbol{v}} \mathcal{L}_\omega(\boldsymbol{v}_k, \boldsymbol{\lambda}_k) \rangle + \frac{L_2}{2} \left\| \boldsymbol{v}_{k+1} - \boldsymbol{v}_k \right\|_2^2 \tag{155}$$

$$\leqslant \zeta_{\boldsymbol{v},k} \left\langle \widehat{\nabla}_{\boldsymbol{v}} \mathcal{L}_\omega(\boldsymbol{v}_k, \boldsymbol{\lambda}_k), \ \nabla_{\boldsymbol{v}} \mathcal{L}_\omega(\boldsymbol{v}_k, \boldsymbol{\lambda}_k) \right\rangle + \frac{L_2}{2} \zeta_{\boldsymbol{v},k}^2 \left\| \widehat{\nabla}_{\boldsymbol{v}} \mathcal{L}_\omega(\boldsymbol{v}_k, \boldsymbol{\lambda}_k) \right\|_2^2, \tag{156}$$

again because of Assumption 3.3 and the update rule. Now, as usual, we consider the expectation conditioned to the filtration $\mathcal{F}_{k-1}$ and the properties of the variance, to obtain:

$$\textcircled{C} := \mathcal{L}_\omega(\boldsymbol{v}_k, \boldsymbol{\lambda}_k) - \mathbb{E}\left[ \mathcal{L}_\omega(\boldsymbol{v}_{k+1}, \boldsymbol{\lambda}_k) | \mathcal{F}_{k-1} \right] \tag{157}$$

$$\leqslant \zeta_{\boldsymbol{v},k} \left( 1 + \frac{L_2}{2} \zeta_{\boldsymbol{v},k} \right) \left\| \nabla_{\boldsymbol{v}} \mathcal{L}_\omega(\boldsymbol{v}_k, \boldsymbol{\lambda}_k) \right\|_2^2 + \frac{L_2}{2} \zeta_{\boldsymbol{v},k}^2 V_{\boldsymbol{v}}, \tag{158}$$

having set $\zeta_{\boldsymbol{v},k} \leqslant 1/L_2$. We are finally able to conclude the bound of the term \textcircled{B}:

$$\textcircled{B} := \mathbb{E}\left[ H_\omega(\boldsymbol{v}_{k+1}) - \mathcal{L}_\omega(\boldsymbol{v}_{k+1}, \boldsymbol{\lambda}_{k+1}) | \mathcal{F}_{k-1} \right] \tag{159}$$

$$\leqslant \left( 1 - \frac{\zeta_{\boldsymbol{\lambda},k}}{2} \omega \right) \mathbb{E}\left[ H_\omega(\boldsymbol{v}_{k+1}) - \mathcal{L}_\omega(\boldsymbol{v}_{k+1}, \boldsymbol{\lambda}_k) | \mathcal{F}_{k-1} \right] + \frac{\omega}{2} \zeta_{\boldsymbol{\lambda},k}^2 V_{\boldsymbol{\lambda}} \tag{160}$$

$$= \left(1 - \frac{\zeta_{\boldsymbol{\lambda},k}}{2}\omega\right)(H_\omega(\boldsymbol{v}_k) - \mathcal{L}_\omega(\boldsymbol{v}_k, \boldsymbol{\lambda}_k)) \tag{161}$$

$$+ \left(1 - \frac{\zeta_{\boldsymbol{\lambda},k}}{2}\omega\right)(\mathcal{L}_\omega(\boldsymbol{v}_k, \boldsymbol{\lambda}_k) - \mathbb{E}\left[\mathcal{L}_\omega(\boldsymbol{v}_{k+1}, \boldsymbol{\lambda}_k)|\mathcal{F}_{k-1}\right]) \tag{162}$$

$$+ \left(1 - \frac{\zeta_{\boldsymbol{\lambda},k}}{2}\omega\right)(\mathbb{E}\left[H_\omega(\boldsymbol{v}_{k+1})|\mathcal{F}_{k-1}\right] - H_\omega(\boldsymbol{v}_k)) + \frac{\omega}{2}\zeta_{\boldsymbol{\lambda},k}^2 V_{\boldsymbol{\lambda}}. \tag{163}$$

Now we apply the bounds on $\mathbb{C}$ and $\mathbb{D}$ (the latter is from Eq. 136), obtaining:

$$\mathbb{E}\left[H_\omega(\boldsymbol{v}_{k+1}) - \mathcal{L}_\omega(\boldsymbol{v}_{k+1}, \boldsymbol{\lambda}_{k+1})|\mathcal{F}_{k-1}\right] \tag{164}$$

$$\leqslant \left(1 - \frac{\zeta_{\boldsymbol{\lambda},k}}{2}\omega\right)(H_\omega(\boldsymbol{v}_k) - \mathcal{L}_\omega(\boldsymbol{v}_k, \boldsymbol{\lambda}_k)) \tag{165}$$

$$+ \left(1 - \frac{\zeta_{\boldsymbol{\lambda},k}}{2}\omega\right)\left(\zeta_{\boldsymbol{v},k}\left(1 + \frac{L_2}{2}\zeta_{\boldsymbol{v},k}\right)\|\nabla_{\boldsymbol{v}}\mathcal{L}_\omega(\boldsymbol{v}_k, \boldsymbol{\lambda}_k)\|_2^2 + \frac{L_2}{2}\zeta_{\boldsymbol{v},k}^2 V_{\boldsymbol{v}}\right) \tag{166}$$

$$+ \left(1 - \frac{\zeta_{\boldsymbol{\lambda},k}}{2}\omega\right)\left(-\frac{\zeta_{\boldsymbol{v},k}}{2}\|\nabla_{\boldsymbol{v}}H_\omega(\boldsymbol{v}_k)\|_2^2 + \frac{\zeta_{\boldsymbol{v},k}}{2}\|\nabla_{\boldsymbol{v}}\mathcal{L}_\omega(\boldsymbol{v}_k, \boldsymbol{\lambda}_k) - \nabla_{\boldsymbol{v}}H_\omega(\boldsymbol{v}_k)\|_2^2 \tag{167}$$

$$+ \frac{L_H}{2}\zeta_{\boldsymbol{v},k}^2 V_{\boldsymbol{v}}\right) + \frac{\omega}{2}\zeta_{\boldsymbol{\lambda},k}^2 V_{\boldsymbol{\lambda}}, \tag{168}$$

that is the second fundamental term.

**Part III: bounding the potential function $P_k(\chi)$.** Before going on, we recall that so far we enforced: $\zeta_{\boldsymbol{v},k} \leqslant 1/L_H$ (since $L_H \geqslant L_2$) and $\zeta_{\boldsymbol{\lambda},k} \leqslant 1/\omega$, for every $t \in [\![K]\!]$. What we want to bound here is the potential function $P_{k+1}(\chi) = a_{k+1} + \chi b_{k+1}$. Using the final results of Part I and Part II:

$$a_{k+1} + \chi b_{k+1} = \mathbb{E}\left[H_\omega(\boldsymbol{v}_{k+1}) - H^*\right] + \chi \mathbb{E}\left[H_\omega(\boldsymbol{v}_{k+1}) - \mathcal{L}_\omega(\boldsymbol{v}_{k+1}, \boldsymbol{\lambda}_{k+1})\right] \tag{169}$$

$$\leqslant \mathbb{E}\left[H_\omega(\boldsymbol{v}_k) - H^*\right] - \frac{\zeta_{\boldsymbol{v},k}}{2}\mathbb{E}\left[\|\nabla_{\boldsymbol{v}}H_\omega(\boldsymbol{v}_k)\|_2^2\right] \tag{170}$$

$$+ \frac{\zeta_{\boldsymbol{v},k}}{2}\mathbb{E}\left[\|\nabla_{\boldsymbol{v}}\mathcal{L}_\omega(\boldsymbol{v}_k, \boldsymbol{\lambda}_k) - \nabla_{\boldsymbol{v}}H_\omega(\boldsymbol{v}_k)\|_2^2\right] + \frac{L_H}{2}\zeta_{\boldsymbol{v},k}^2 V_{\boldsymbol{v}} \tag{171}$$

$$+ \chi\left(1 - \frac{\zeta_{\boldsymbol{\lambda},k}}{2}\omega\right)\mathbb{E}\left[H_\omega(\boldsymbol{v}_k) - \mathcal{L}_\omega(\boldsymbol{v}_k, \boldsymbol{\lambda}_k)\right] \tag{172}$$

$$+ \chi\left(1 - \frac{\zeta_{\boldsymbol{\lambda},k}}{2}\omega\right)\left(\zeta_{\boldsymbol{v},k}\left(1 + \frac{L_2}{2}\zeta_{\boldsymbol{v},k}\right)\mathbb{E}\left[\|\nabla_{\boldsymbol{v}}\mathcal{L}_\omega(\boldsymbol{v}_k, \boldsymbol{\lambda}_k)\|_2^2\right] + \frac{L_2}{2}\zeta_{\boldsymbol{v},k}^2 V_{\boldsymbol{v}}\right) \tag{173}$$

$$+ \chi\left(1 - \frac{\zeta_{\boldsymbol{\lambda},k}}{2}\omega\right)\left(-\frac{\zeta_{\boldsymbol{v},k}}{2}\mathbb{E}\left[\|\nabla_{\boldsymbol{v}}H_\omega(\boldsymbol{v}_k)\|_2^2\right]\right. \tag{174}$$

$$\left.+ \frac{\zeta_{\boldsymbol{v},k}}{2}\mathbb{E}\left[\|\nabla_{\boldsymbol{v}}\mathcal{L}_\omega(\boldsymbol{v}_k, \boldsymbol{\lambda}_k) - \nabla_{\boldsymbol{v}}H_\omega(\boldsymbol{v}_k)\|_2^2\right] + \frac{L_H}{2}\zeta_{\boldsymbol{v},k}^2 V_{\boldsymbol{v}}\right) \tag{175}$$

$$+ \chi\frac{\omega}{2}\zeta_{\boldsymbol{\lambda},k}^2 V_{\boldsymbol{\lambda}} \tag{176}$$

$$= a_k + \chi\left(1 - \frac{\zeta_{\boldsymbol{\lambda},k}}{2}\omega\right)b_k \tag{177}$$

$$- \frac{\zeta_{\boldsymbol{v},k}}{2}\left(1 + \chi\left(1 - \frac{\zeta_{\boldsymbol{\lambda},k}}{2}\omega\right)\right)\mathbb{E}\left[\|\nabla_{\boldsymbol{v}}H_\omega(\boldsymbol{v}_k)\|_2^2\right] \tag{178}$$

$$+ \frac{\zeta_{\boldsymbol{v},k}}{2}\left(1 + \chi\left(1 - \frac{\zeta_{\boldsymbol{\lambda},k}}{2}\omega\right)\right)\mathbb{E}\left[\|\nabla_{\boldsymbol{v}}\mathcal{L}_\omega(\boldsymbol{v}_k, \boldsymbol{\lambda}_k) - \nabla_{\boldsymbol{v}}H_\omega(\boldsymbol{v}_k)\|_2^2\right] \tag{179}$$

$$+ \zeta_{\boldsymbol{v},k}\left(1 + \frac{L_2}{2}\zeta_{\boldsymbol{v},k}\right)\chi\left(1 - \frac{\zeta_{\boldsymbol{\lambda},k}}{2}\omega\right)\mathbb{E}\left[\|\nabla_{\boldsymbol{v}}\mathcal{L}_\omega(\boldsymbol{v}_k, \boldsymbol{\lambda}_k)\|_2^2\right] \tag{180}$$

$$+ \frac{\zeta_{\boldsymbol{v},k}^2}{2}\left(L_H + \chi\left(1 - \frac{\zeta_{\boldsymbol{\lambda},k}}{2}\omega\right)(L_H + L_2)\right)V_{\boldsymbol{v}} + \chi\frac{\omega}{2}\zeta_{\boldsymbol{\lambda},k}^2 V_{\boldsymbol{\lambda}}. \tag{181}$$

Now we can re-arrange the terms by noticing that:

$$\|\nabla_{\boldsymbol{v}}\mathcal{L}_\omega(\boldsymbol{v}_k,\boldsymbol{\lambda}_k)\|_2^2 = \|\nabla_{\boldsymbol{v}}\mathcal{L}_\omega(\boldsymbol{v}_k,\boldsymbol{\lambda}_k) - \nabla_{\boldsymbol{v}}H_\omega(\boldsymbol{v}_k) + \nabla_{\boldsymbol{v}}H_\omega(\boldsymbol{v}_k)\|_2^2 \tag{182}$$

$$= \|\nabla_{\boldsymbol{v}}\mathcal{L}_\omega(\boldsymbol{v}_k,\boldsymbol{\lambda}_k) - \nabla_{\boldsymbol{v}}H_\omega(\boldsymbol{v}_k)\|_2^2 \tag{183}$$

$$+ \|\nabla_{\boldsymbol{v}}H_\omega(\boldsymbol{v}_k)\|_2^2 + 2\left\langle\nabla_{\boldsymbol{v}}\mathcal{L}_\omega(\boldsymbol{v}_k,\boldsymbol{\lambda}_k) - \nabla_{\boldsymbol{v}}H_\omega(\boldsymbol{v}_k),\ \nabla_{\boldsymbol{v}}H_\omega(\boldsymbol{v}_k)\right\rangle \tag{184}$$

$$\leqslant 2\|\nabla_{\boldsymbol{v}}\mathcal{L}_\omega(\boldsymbol{v}_k,\boldsymbol{\lambda}_k) - \nabla_{\boldsymbol{v}}H_\omega(\boldsymbol{v}_k)\|_2^2 + 2\|\nabla_{\boldsymbol{v}}H_\omega(\boldsymbol{v}_k)\|_2^2, \tag{185}$$

where the last inequality holds by Young's inequality. Then we can write what follows:

$$a_{k+1} + \chi b_{k+1} \tag{186}$$

$$\leqslant a_k + \chi\left(1 - \frac{\zeta_{\boldsymbol{\lambda},k}}{2}\omega\right)b_k \tag{187}$$

$$+ \left(2\zeta_{\boldsymbol{v},k}\left(1 + \frac{L_2}{2}\zeta_{\boldsymbol{v},k}\right)\chi\left(1 - \frac{\zeta_{\boldsymbol{\lambda},k}}{2}\omega\right)\right. \tag{188}$$

$$\left. - \frac{\zeta_{\boldsymbol{v},k}}{2}\left(1 + \chi\left(1 - \frac{\zeta_{\boldsymbol{\lambda},k}}{2}\omega\right)\right)\right)\mathbb{E}\left[\|\nabla_{\boldsymbol{v}}H_\omega(\boldsymbol{v}_k)\|_2^2\right] \tag{189}$$

$$+ \left(2\zeta_{\boldsymbol{v},k}\left(1 + \frac{L_2}{2}\zeta_{\boldsymbol{v},k}\right)\chi\left(1 - \frac{\zeta_{\boldsymbol{\lambda},k}}{2}\omega\right) + \frac{\zeta_{\boldsymbol{v},k}}{2}\left(1 + \chi\left(1 - \frac{\zeta_{\boldsymbol{\lambda},k}}{2}\omega\right)\right)\right) \tag{190}$$

$$\cdot\mathbb{E}\left[\|\nabla_{\boldsymbol{v}}\mathcal{L}_\omega(\boldsymbol{v}_k,\boldsymbol{\lambda}_k) - \nabla_{\boldsymbol{v}}H_\omega(\boldsymbol{v}_k)\|_2^2\right] \tag{191}$$

$$+ \frac{\zeta_{\boldsymbol{v},k}^2}{2}\left(L_H + \chi\left(1 - \frac{\zeta_{\boldsymbol{\lambda},k}}{2}\omega\right)(L_H + L_2)\right)V_{\boldsymbol{v}} + \chi\frac{\omega}{2}\zeta_{\boldsymbol{\lambda},k}^2 V_{\boldsymbol{\lambda}}. \tag{192}$$

Let us now proceed to bound $\|\nabla_{\boldsymbol{v}}\mathcal{L}_\omega(\boldsymbol{v}_k,\boldsymbol{\lambda}_k) - \nabla_{\boldsymbol{v}}H_\omega(\boldsymbol{v}_k)\|_2^2$. By Lemma E.7, we have that $\nabla_{\boldsymbol{v}}H_\omega(\boldsymbol{v}) = \nabla_{\boldsymbol{v}}\mathcal{L}_\omega(\boldsymbol{v},\boldsymbol{\lambda}^*(\boldsymbol{v}))$ for every $\boldsymbol{\lambda}^*(\boldsymbol{v}) \in \arg\max_{\overline{\boldsymbol{\lambda}}\in\Lambda}\mathcal{L}_\omega(\boldsymbol{v},\overline{\boldsymbol{\lambda}})$, thus we can write:

$$\|\nabla_{\boldsymbol{v}}\mathcal{L}_\omega(\boldsymbol{v}_k,\boldsymbol{\lambda}_k) - \nabla_{\boldsymbol{v}}H_\omega(\boldsymbol{v}_k)\|_2^2 = \|\nabla_{\boldsymbol{v}}\mathcal{L}_\omega(\boldsymbol{v}_k,\boldsymbol{\lambda}_k) - \nabla_{\boldsymbol{v}}\mathcal{L}_\omega(\boldsymbol{v}_k,\boldsymbol{\lambda}^*(\boldsymbol{v}_k))\|_2^2 \tag{193}$$

$$\leqslant L_3^2\|\boldsymbol{\lambda}^*(\boldsymbol{v}_k) - \boldsymbol{\lambda}_k\|_2^2, \tag{194}$$

since Assumption 3.3 holds.

For a fixed value of $\boldsymbol{v}$, by Lemma E.5 it follows that $\mathcal{L}_\omega(\boldsymbol{v},\cdot)$ satisfies the quadratic growth condition (since it satisfies the PL condition), for which the following holds:

$$\|\boldsymbol{\lambda}^*(\boldsymbol{v}_k) - \boldsymbol{\lambda}_k\|_2^2 \leqslant \frac{4}{\omega}\left(H_\omega(\boldsymbol{v}_k) - \mathcal{L}_\omega(\boldsymbol{v}_k,\boldsymbol{\lambda}_k)\right), \tag{195}$$

and thus we have:

$$\|\nabla_{\boldsymbol{v}}\mathcal{L}_\omega(\boldsymbol{v}_k,\boldsymbol{\lambda}_k) - \nabla_{\boldsymbol{v}}H_\omega(\boldsymbol{v}_k)\|_2^2 \leqslant \frac{4L_3^2}{\omega}\left(H_\omega(\boldsymbol{v}_k) - \mathcal{L}_\omega(\boldsymbol{v}_k,\boldsymbol{\lambda}_k)\right). \tag{196}$$

By applying the total expectation, it trivially follows:

$$\mathbb{E}\left[\|\nabla_{\boldsymbol{v}}\mathcal{L}_\omega(\boldsymbol{v}_k,\boldsymbol{\lambda}_k) - \nabla_{\boldsymbol{v}}H_\omega(\boldsymbol{v}_k)\|_2^2\right] \leqslant \frac{4L_3^2}{\omega}\mathbb{E}\left[H_\omega(\boldsymbol{v}_k) - \mathcal{L}_\omega(\boldsymbol{v}_k,\boldsymbol{\lambda}_k)\right] = \frac{4L_3^2}{\omega}b_k. \tag{197}$$

Thus, we have:

$$a_{k+1} + \chi b_{k+1} \tag{198}$$

$$\leqslant a_k + \chi\left(1 - \frac{\zeta_{\boldsymbol{\lambda},k}}{2}\omega\right)b_k \tag{199}$$

$$+ \left(2\zeta_{\boldsymbol{v},k}\left(1 + \frac{L_2}{2}\zeta_{\boldsymbol{v},k}\right)\chi\left(1 - \frac{\zeta_{\boldsymbol{\lambda},k}}{2}\omega\right)\right. \tag{200}$$

$$\left. - \frac{\zeta_{\boldsymbol{v},k}}{2}\left(1 + \chi\left(1 - \frac{\zeta_{\boldsymbol{\lambda},k}}{2}\omega\right)\right)\right)\mathbb{E}\left[\|\nabla_{\boldsymbol{v}}H_\omega(\boldsymbol{v}_k)\|_2^2\right] \tag{201}$$

$$+ \left(2\zeta_{\boldsymbol{v},k}\left(1 + \frac{L_2}{2}\zeta_{\boldsymbol{v},k}\right)\chi\left(1 - \frac{\zeta_{\boldsymbol{\lambda},k}}{2}\omega\right) + \frac{\zeta_{\boldsymbol{v},k}}{2}\left(1 + \chi\left(1 - \frac{\zeta_{\boldsymbol{\lambda},k}}{2}\omega\right)\right)\right)\frac{4L_3^2}{\omega}b_k \tag{202}$$

$$+ \frac{\zeta_{\boldsymbol{v},k}^2}{2}\left(L_H + \chi\left(1 - \frac{\zeta_{\boldsymbol{\lambda},k}}{2}\omega\right)(L_H + L_2)\right)V_{\boldsymbol{v}} + \chi\frac{\omega}{2}\zeta_{\boldsymbol{\lambda},k}^2 V_{\boldsymbol{\lambda}}. \tag{203}$$

**Part IV: apply the $\psi$-gradient domination.** Now we need to bound the term $\|\nabla H_\omega(\boldsymbol{v}_k)\|_2^2$. We consider Assumption 3.2 and we get: $\|\nabla_{\boldsymbol{v}} H_\omega(\boldsymbol{v}_k)\|_2^\psi \geqslant \alpha_1 (H_\omega(\boldsymbol{v}_k) - H^*) - \beta_1$. By defining $\widetilde{H}^* := H^* + \beta_1/\alpha_1$, we also have:

$$\|\nabla_{\boldsymbol{v}} H_\omega(\boldsymbol{v})\|_2^\psi \geqslant \alpha_1 \max\left\{0,\ H_\omega(\boldsymbol{v}) - \widetilde{H}^*\right\} \tag{204}$$
$$\implies$$
$$\|\nabla_{\boldsymbol{v}} H_\omega(\boldsymbol{v})\|_2^2 \geqslant \alpha_1^{\frac{2}{\psi}} \max\left\{0,\ H_\omega(\boldsymbol{v}) - \widetilde{H}^*\right\}^{\frac{2}{\psi}}.$$

If we apply the total expectation on both sides of the inequality, we get:

$$\mathbb{E}\left[\|\nabla_{\boldsymbol{v}} H_\omega(\boldsymbol{v})\|_2^2\right] \geqslant \alpha_1^{\frac{2}{\psi}} \mathbb{E}\left[\max\left\{0,\ H_\omega(\boldsymbol{v}) - \widetilde{H}^*\right\}^{\frac{2}{\psi}}\right] \tag{205}$$

$$\geqslant \alpha_1^{\frac{2}{\psi}} \mathbb{E}\left[\max\left\{0,\ H_\omega(\boldsymbol{v}) - \widetilde{H}^*\right\}\right]^{\frac{2}{\psi}} \tag{206}$$

$$\geqslant \alpha_1^{\frac{2}{\psi}} \max\left\{0,\ \mathbb{E}\left[H_\omega(\boldsymbol{v}) - \widetilde{H}^*\right]\right\}^{\frac{2}{\psi}}, \tag{207}$$

which is achieved by a double application of Jensen's inequality, since $z^{2/\psi}$ is convex for $\psi \in [1,2]$ and $z \geqslant 0$, and the maximum is convex. Let us start from Equation (198):

$$a_{k+1} + \chi b_{k+1} \tag{208}$$

$$\leqslant a_k + \chi\left(1 - \frac{\zeta_{\boldsymbol{\lambda},k}}{2}\omega\right)b_k \tag{209}$$

$$+ \underbrace{\left(2\zeta_{\boldsymbol{v},k}\left(1 + \frac{L_2}{2}\zeta_{\boldsymbol{v},k}\right)\chi\left(1 - \frac{\zeta_{\boldsymbol{\lambda},k}}{2}\omega\right) - \frac{\zeta_{\boldsymbol{v},k}}{2}\left(1 + \chi\left(1 - \frac{\zeta_{\boldsymbol{\lambda},k}}{2}\omega\right)\right)\right)}_{=:-C} \tag{210}$$

$$\cdot \mathbb{E}\left[\|\nabla H_\omega(\boldsymbol{v}_k)\|_2^2\right] \tag{211}$$

$$+ \left(2\zeta_{\boldsymbol{v},k}\left(1 + \frac{L_2}{2}\zeta_{\boldsymbol{v},k}\right)\chi\left(1 - \frac{\zeta_{\boldsymbol{\lambda},k}}{2}\omega\right) + \frac{\zeta_{\boldsymbol{v},k}}{2}\left(1 + \chi\left(1 - \frac{\zeta_{\boldsymbol{\lambda},k}}{2}\omega\right)\right)\right)\frac{4L_3^2}{\omega}b_k \tag{212}$$

$$+ \underbrace{\frac{\zeta_{\boldsymbol{v},k}^2}{2}\left(L_H + \chi\left(1 - \frac{\zeta_{\boldsymbol{\lambda},k}}{2}\omega\right)(L_H + L_2)\right)V_{\boldsymbol{v}} + \chi\frac{\omega}{2}\zeta_{\boldsymbol{\lambda},k}^2 V_{\boldsymbol{\lambda}}}_{=:V}. \tag{213}$$

We first enforce the negativity of $-C$. To this end:

$$-C = \left(2\zeta_{\boldsymbol{v},k}\underbrace{\left(1 + \frac{L_2}{2}\zeta_{\boldsymbol{v},k}\right)}_{\leqslant 3/2}\chi\left(1 - \frac{\zeta_{\boldsymbol{\lambda},k}}{2}\omega\right) - \frac{\zeta_{\boldsymbol{v},k}}{2}\left(1 + \chi\left(1 - \frac{\zeta_{\boldsymbol{\lambda},k}}{2}\omega\right)\right)\right) \tag{214}$$

$$\leqslant \zeta_{\boldsymbol{v},k}\left(3\chi\left(1 - \frac{\zeta_{\boldsymbol{\lambda},k}}{2}\omega\right) - \frac{1}{2}\left(1 + \chi\left(1 - \frac{\zeta_{\boldsymbol{\lambda},k}}{2}\omega\right)\right)\right) \tag{215}$$

$$\leqslant \frac{\zeta_{\boldsymbol{v},k}}{2}\left(5\chi\underbrace{\left(1 - \frac{\zeta_{\boldsymbol{\lambda},k}}{2}\omega\right)}_{\leqslant 1} - 1\right) \leqslant \frac{\zeta_{\boldsymbol{v},k}}{2}(5\chi - 1) \leqslant 0. \tag{216}$$

Thus, it is enough to enforce $5\chi - 1 \leqslant 0 \implies \chi \leqslant 1/5$. We now plug in the gradient domination inequalities:

$$a_{k+1} + \chi b_{k+1} \tag{217}$$

$$\leqslant a_k + \chi \left(1 - \frac{\zeta_{\boldsymbol{\lambda},k}}{2}\omega\right) b_k - C\alpha_1^{\frac{2}{\psi}} \max\left\{0; \, \mathbb{E}\left[H_\omega(\boldsymbol{v}) - \tilde{H}^*\right]\right\}^{\frac{2}{\psi}} + V \tag{218}$$

$$+ \left(2\zeta_{\boldsymbol{v},k}\left(1 + \frac{L_2}{2}\zeta_{\boldsymbol{v},k}\right)\chi\left(1 - \frac{\zeta_{\boldsymbol{\lambda},k}}{2}\omega\right) + \frac{\zeta_{\boldsymbol{v},k}}{2}\left(1 + \chi\left(1 - \frac{\zeta_{\boldsymbol{\lambda},k}}{2}\omega\right)\right)\right)\frac{4L_3^2}{\omega}b_k. \tag{219}$$

Now we introduce the symbol $\tilde{a}_k := \mathbb{E}\left[H_\omega(\boldsymbol{v}_k) - \tilde{H}^*\right] = a_k - \beta_1/\alpha_1$, to get:

$$\tilde{a}_{k+1} + \chi b_{k+1} \tag{220}$$

$$\leqslant \tilde{a}_k - C\alpha_1^{\frac{2}{\psi}} \max\left\{0, \tilde{a}_k\right\}^{\frac{2}{\psi}} + V \tag{221}$$

$$+ \left(\chi\left(1 - \frac{\zeta_{\boldsymbol{\lambda},k}}{2}\omega\right) + \left(2\zeta_{\boldsymbol{v},k}\left(1 + \frac{L_2}{2}\zeta_{\boldsymbol{v},k}\right)\chi\left(1 - \frac{\zeta_{\boldsymbol{\lambda},k}}{2}\omega\right)\right. \right. \tag{222}$$

$$\left. \left. + \frac{\zeta_{\boldsymbol{v},k}}{2}\left(1 + \chi\left(1 - \frac{\zeta_{\boldsymbol{\lambda},k}}{2}\omega\right)\right)\right)\frac{4L_3^2}{\omega}\right)b_k. \tag{223}$$

For what follows, we call $B$ the term that is multiplying $b_k$:

$$B := \chi\left(1 - \frac{\zeta_{\boldsymbol{\lambda},k}}{2}\omega\right) + \left(2\zeta_{\boldsymbol{v},k}\left(1 + \frac{L_2}{2}\zeta_{\boldsymbol{v},k}\right)\chi\left(1 - \frac{\zeta_{\boldsymbol{\lambda},k}}{2}\omega\right)\right. \tag{224}$$

$$\left. + \frac{\zeta_{\boldsymbol{v},k}}{2}\left(1 + \chi\left(1 - \frac{\zeta_{\boldsymbol{\lambda},k}}{2}\omega\right)\right)\right)\frac{4L_3^2}{\omega} \tag{225}$$

Let refer to $\tilde{a}_k + \chi b_k$ as $\tilde{P}_t(\chi)$ with $\chi \in (0,1)$. For the sake of clarity, we re-write our main inequality as:

$$\tilde{P}_t(\chi) = \tilde{a}_{k+1} + \chi b_{k+1} \leqslant \tilde{a}_k + Bb_k - C\max\left\{0; \, \tilde{a}_k\right\}^{\frac{2}{\psi}} + V. \tag{226}$$

Then, from Lemma F.1 having set $a \leftarrow \tilde{a}_k$ and $b \leftarrow \chi b_k$, we have:

$$\tilde{P}_{t+1}(\chi) = \tilde{a}_{k+1} + \chi b_{k+1} \leqslant \tilde{a}_k + Bb_k + C(\chi b_k)^{\frac{2}{\psi}} - 2^{1-\frac{2}{\psi}}C\max\left\{0, \, \tilde{a}_k + \chi b_k\right\}^{\frac{2}{\psi}} + V. \tag{227}$$

By choosing $\chi$ so that $\chi b_k \leqslant 1$, i.e., $\chi \leqslant 1/\max_{k\in[K]} b_k$, we have:

$$\tilde{P}_{t+1}(\chi) = \tilde{a}_{k+1} + \chi b_{k+1} \leqslant \tilde{a}_k + Bb_k + C(\chi b_k)^{\frac{2}{\psi}} - 2^{1-\frac{2}{\psi}}C\max\left\{0, \, \tilde{a}_k + \chi b_k\right\}^{\frac{2}{\psi}} + V \tag{228}$$

$$\leqslant \tilde{a}_k + (B + \chi C)b_k - 2^{1-\frac{2}{\psi}}C\max\left\{0, \, \tilde{a}_k + \chi b_k\right\}^{\frac{2}{\psi}} + V \tag{229}$$

$$= \tilde{P}_t(B + \chi C) - 2^{1-\frac{2}{\psi}}C\max\left\{0, \, \tilde{P}_t(\chi)\right\}^{\frac{2}{\psi}} + V. \tag{230}$$

To unfold the recursion, we need to ensure that $B + \chi C \leqslant \chi$, which leads to a condition relating the two learning rates:

$$B + \chi C \tag{231}$$

$$= \chi\left(1 - \frac{\zeta_{\boldsymbol{\lambda},k}}{2}\omega\right) + \left(2\zeta_{\boldsymbol{v},k}\underbrace{\left(1 + \frac{L_2}{2}\zeta_{\boldsymbol{v},k}\right)}_{\leqslant 3/2}\chi\underbrace{\left(1 - \frac{\zeta_{\boldsymbol{\lambda},k}}{2}\omega\right)}_{\leqslant 1}\right. \tag{232}$$

$$\left. + \frac{\zeta_{\boldsymbol{v},k}}{2}\left(1 + \chi\underbrace{\left(1 - \frac{\zeta_{\boldsymbol{\lambda},k}}{2}\omega\right)}_{\leqslant 1}\right)\right)\frac{4L_3^2}{\omega} \tag{233}$$

$$+ \chi\left(\underbrace{-2\zeta_{\boldsymbol{v},k}\left(1 + \frac{L_2}{2}\zeta_{\boldsymbol{v},k}\right)\chi\left(1 - \frac{\zeta_{\boldsymbol{\lambda},k}}{2}\omega\right)}_{\leqslant 0} + \frac{\zeta_{\boldsymbol{v},k}}{2}\left(1 + \chi\underbrace{\left(1 - \frac{\zeta_{\boldsymbol{\lambda},k}}{2}\omega\right)}_{\leqslant 1}\right)\right)\alpha_1^{\frac{2}{\psi}} \tag{234}$$

$$\leqslant \chi - \chi\frac{\zeta_{\boldsymbol{\lambda},k}}{2}\omega + \zeta_{\boldsymbol{v},k}\left(\frac{2L_3^2}{\omega}(1 + 7\chi) + \frac{1+\chi}{2}\alpha_2^{\frac{2}{\psi}}\right) \leqslant \chi \tag{235}$$

$$\implies \zeta_{\boldsymbol{v},k} \leqslant \frac{\omega^2 \chi \zeta_{\boldsymbol{\lambda},k}}{(1+\chi)\omega\alpha_1^{\frac{2}{\psi}} + 4L_3^2(1+7\chi)}, \tag{236}$$

where we exploited $\zeta_{\boldsymbol{v},k} \leqslant 1/L_2$ and $\zeta_{\boldsymbol{\lambda},k} \leqslant 2/\omega$. Thus, we have:

$$\widetilde{P}_{k+1}(\chi) \leqslant \widetilde{P}_k(\chi) - 2^{1-\frac{2}{\psi}}C\max\left\{0,\ \widetilde{P}_k(\chi)\right\}^{\frac{2}{\psi}} + V. \tag{237}$$

Collecting all conditions on the learning rates, we have:

$$\zeta_{\boldsymbol{v},k} \leqslant \min\left\{\frac{1}{L_H}, \frac{1}{L_2}, \frac{\omega^2\chi\zeta_{\boldsymbol{\lambda},k}}{(1+\chi)\omega\alpha_1^{\frac{2}{\psi}} + 4L_3^2(1+7\chi)}\right\}, \tag{238}$$

$$\zeta_{\boldsymbol{\lambda},k} \leqslant \min\left\{\frac{1}{\omega}, \frac{2}{\omega}\right\} = \frac{1}{\omega}. \tag{239}$$

As a further simplification, let us observe that:

$$C = \left(-2\zeta_{\boldsymbol{v},k}\underbrace{\left(1 + \frac{L_2}{2}\zeta_{\boldsymbol{v},k}\right)}_{\leqslant 3/2}\chi\left(1 - \frac{\zeta_{\boldsymbol{\lambda},k}}{2}\omega\right) + \frac{\zeta_{\boldsymbol{v},k}}{2}\left(1 + \chi\left(1 - \frac{\zeta_{\boldsymbol{\lambda},k}}{2}\omega\right)\right)\right)\alpha_1^{\frac{2}{\psi}} \tag{240}$$

$$\geqslant \frac{\zeta_{\boldsymbol{v},k}}{2}\left(1 + 5\left(1 - \frac{\zeta_{\boldsymbol{\lambda},k}}{2}\omega\right)\chi\right)\alpha_1^{\frac{2}{\psi}} \geqslant \frac{\zeta_{\boldsymbol{v},k}\alpha_1^{\frac{2}{\psi}}}{2}. \tag{241}$$

$$V = \frac{\zeta_{\boldsymbol{v},k}^2}{2}\left(L_H + \chi\left(1 - \frac{\zeta_{\boldsymbol{\lambda},k}}{2}\omega\right)(L_H + L_2)\right)V_{\boldsymbol{v}} + \chi\frac{\omega}{2}\zeta_{\boldsymbol{\lambda},k}^2 V_{\boldsymbol{\lambda}} \tag{242}$$

$$\leqslant \frac{\zeta_{\boldsymbol{v},k}^2}{2}\left((1+2\chi)L_2 + (1+\chi)\frac{L_1^2}{\omega}\right)V_{\boldsymbol{v}} + \chi\frac{\omega}{2}\zeta_{\boldsymbol{\lambda},k}^2 V_{\boldsymbol{\lambda}} =: \widetilde{V}. \tag{243}$$

Denoting with $\widetilde{C} =: 2^{1-\frac{1}{\psi}}\frac{\zeta_{\boldsymbol{v},k}\alpha_1^{\frac{2}{\psi}}}{2}$, we are going to study the recurrence:

$$\widetilde{P}_{k+1}(\chi) \leqslant \widetilde{P}_k(\chi) - \widetilde{C}\max\left\{0,\ \widetilde{P}_k(\chi)\right\}^{\frac{2}{\psi}} + \widetilde{V}. \tag{244}$$

**Part V: Rates Computation**

*Part V(a): Exact gradients* We consider the case $\widetilde{V} = 0$. Let us start with $\psi = 2$. From Lemma G.3, we have:

$$\widetilde{P}_K(\xi) \leqslant \left(1 - \widetilde{C}\right)^K \widetilde{P}_0(\xi) \leqslant \epsilon \tag{245}$$

$$\implies K \leqslant \frac{\log\frac{\widetilde{P}_0(\xi)}{\epsilon}}{\log\frac{1}{1-\widetilde{C}}} \leqslant \widetilde{C}^{-1}\log\frac{\widetilde{P}_0(\xi)}{\epsilon} = \frac{2\log\frac{\widetilde{P}_0(\xi)}{\epsilon}}{2^{1-\frac{1}{\psi}}\zeta_{\boldsymbol{\rho},t}\alpha_1^{\frac{2}{\psi}}} \tag{246}$$

The inequality on $K$ holds under the conditions:

$$\widetilde{C} \leqslant \frac{2}{\psi\widetilde{P}_0(\chi)^{\frac{2}{\psi}-1}} \implies \zeta_{\boldsymbol{v},k} \leqslant \frac{2^{1+\frac{2}{\psi}}}{\psi\alpha_1^{\frac{2}{\psi}}\widetilde{P}_0(\chi)^{\frac{2}{\psi}-1}}, \tag{247}$$

$$\zeta_{\boldsymbol{v},k} \leqslant \min\left\{\frac{1}{L_H}, \frac{1}{L_2}, \frac{\omega^2\chi\zeta_{\boldsymbol{\lambda},k}}{(1+\chi)\omega\alpha_1^{\frac{2}{\psi}} + 4L_3^2(1+7\chi)}\right\} \tag{248}$$

$$= \min\left\{\frac{1}{L_2 + \frac{L_1^2}{\omega}}, \frac{\omega^2\chi\zeta_{\boldsymbol{\lambda},k}}{(1+\chi)\omega\alpha_1^{\frac{2}{\psi}} + 4L_3^2(1+7\chi)}\right\}, \tag{249}$$

$$\zeta_{\boldsymbol{\lambda},k} \leqslant \frac{1}{\omega}, \tag{250}$$

where the first one derives from the hypothesis of Lemma G.3 and the other two from the conditions on the learning rates derived in the previous parts. We set:

$$\zeta_{\boldsymbol{\lambda},k} = \omega^{-1},$$

$$\zeta_{\boldsymbol{v},k} = \min \left\{ \frac{2^{1+\frac{2}{\psi}}}{\psi \alpha_1^{\frac{2}{\psi}} \widetilde{P}_0(\chi)^{\frac{2}{\psi}-1}}, \frac{1}{L_2 + \frac{L_1^2}{\omega}}, \frac{\omega \chi}{(1+\chi)\omega \alpha_1^{\frac{2}{\psi}} + 4L_3^2(1+7\chi)} \right\} = \mathcal{O}(\omega).$$

Thus, the sample complexity becomes $K = \mathcal{O}\left(\omega^{-1} \log \frac{1}{\epsilon}\right)$.

Consider now $\psi \in [1, 2)$. We have from Lemma G.3:

$$\widetilde{P}_K(\chi) \leqslant \left( \left(\frac{2}{\psi} - 1\right) \widetilde{C} K \right)^{-\frac{\psi}{2-\psi}} \leqslant \epsilon \tag{251}$$

$$\implies K \leqslant \frac{\psi}{2-\psi} \widetilde{C}^{-1} \epsilon^{-\frac{2}{\psi}+1} = \frac{2\psi}{(2-\psi)2^{1-\frac{1}{\psi}} \zeta_{\boldsymbol{v},k} \alpha_1^{\frac{2}{\psi}}} \epsilon^{-\frac{2}{\psi}+1}, \tag{252}$$

holding under the same conditions as before. With the same choices of learning rates, we obtain the sample complexity $K = \mathcal{O}\left(\omega^{-1} \epsilon^{-\frac{2}{\psi}+1}\right)$ as sample complexity.

*Part V(b): Estimated gradients* We consider $\widetilde{V} > 0$. In this case, from Lemma G.5, we have:

$$\widetilde{P}_K(\chi) \leqslant \left( 1 - \widetilde{C}^{1-\frac{\psi}{2}} \widetilde{V}^{\frac{\psi}{2}} \right)^K \widetilde{P}_0(\chi) + \left( \frac{\widetilde{V}}{\widetilde{C}} \right)^{\frac{\psi}{2}}. \tag{253}$$

We enforce both terms to be smaller or equal to $\epsilon/2$. With the first one, we can evaluate the sample complexity:

$$\left( 1 - \widetilde{V}^{1-\frac{\psi}{2}} \widetilde{C}^{\frac{\psi}{2}} \right)^K \widetilde{P}_0(\chi) \leqslant \frac{\epsilon}{2} \tag{254}$$

$$\implies K \leqslant \frac{\log \frac{2\widetilde{P}_0(\chi)}{\epsilon}}{\widetilde{V}^{1-\frac{\psi}{2}} \widetilde{C}^{\frac{\psi}{2}}} \tag{255}$$

$$= \frac{\log \frac{2\widetilde{P}_0(\chi)}{\epsilon}}{\left( \frac{\zeta_{\boldsymbol{v},k}^2}{2} \left((1+2\chi)L_2 + (1+\chi)\frac{L_1^2}{\omega}\right) V_{\boldsymbol{v}} + \chi \frac{\omega}{2} \zeta_{\boldsymbol{\lambda},k}^2 V_{\boldsymbol{\lambda}} \right)^{1-\frac{\psi}{2}} \left( 2^{1-\frac{1}{\psi}} \frac{\zeta_{\boldsymbol{v},k} \alpha_1^{\frac{2}{\psi}}}{2} \right)^{\frac{\psi}{2}}} \tag{256}$$

Regarding the second one, we have:

$$\left( \frac{\widetilde{V}}{\widetilde{C}} \right)^{\frac{\psi}{2}} \leqslant \frac{\epsilon}{2} \implies \left( \frac{\frac{\zeta_{\boldsymbol{v},k}^2}{2} \left((1+2\chi)L_2 + (1+\chi)\frac{L_1^2}{\omega}\right) V_{\boldsymbol{v}} + \chi \frac{\omega}{2} \zeta_{\boldsymbol{\lambda},k}^2 V_{\boldsymbol{\lambda}}}{2^{1-\frac{1}{\psi}} \frac{\zeta_{\boldsymbol{v},k} \alpha_1^{\frac{2}{\psi}}}{2}} \right)^{\frac{\psi}{2}} \leqslant \frac{\epsilon}{2} \tag{257}$$

By enforcing the relation between the two learning rates, we set $\zeta_{\boldsymbol{v},k} = \mathcal{O}(\omega^2 \zeta_{\boldsymbol{\lambda},k})$. By enforcing the previous inequality, recalling that $L_2 \leqslant \mathcal{O}(\omega^{-1})$ and $V_{\boldsymbol{v}} \leqslant \mathcal{O}(\omega^{-2})$, we obtain $\zeta_{\boldsymbol{\lambda}} = \mathcal{O}(\omega \epsilon^{2/\psi})$, from which $\zeta_{\boldsymbol{v}} = \mathcal{O}(\omega^3 \epsilon^{2/\psi})$. Substituting these values into the sample complexity upper bound, we get (highlighting the terms possibly depending on $\omega$):

$$K \leqslant \mathcal{O} \left( \frac{\log \frac{1}{\epsilon}}{((L_2 + \omega^{-1})V_{\boldsymbol{v}} \zeta_{\boldsymbol{v}}^2 + \omega \zeta_{\boldsymbol{\lambda}}^2)^{1-\psi/2} \zeta_{\boldsymbol{v}}^{\psi/2}} \right) \tag{258}$$

$$= \mathcal{O} \left( \frac{\log \frac{1}{\epsilon}}{((L_2 + \omega^{-1})V_{\boldsymbol{v}}(\omega^3 \epsilon^{2/\psi})^2 + \omega(\omega \epsilon^{2/\psi})^2)^{1-\psi/2}(\omega^3 \epsilon^{2/\psi})^{\psi/2}} \right) \tag{259}$$

$$\leqslant \mathcal{O} \left( \frac{\log \frac{1}{\epsilon}}{\omega^3 \epsilon^{4/\psi-1}} \right), \tag{260}$$

having bounded the sum at the denominator with the second addendum. $\qquad\square$

## F Technical Lemmas

**Lemma F.1.** *Let $a \in \mathbb{R}$, $b \geqslant 0$, and $\psi \in [1, 2]$. It holds that:*

$$\max\{0, a\}^{\frac{2}{\psi}} \geqslant 2^{1-\frac{2}{\psi}} \max\{0, a+b\}^{\frac{2}{\psi}} - b^{\frac{2}{\psi}}. \tag{261}$$

*Proof.* Let us consider the following derivation:

$$\max\{0, a\}^{\frac{2}{\psi}} = \begin{cases} a^{\frac{2}{\psi}} & \text{if } a > 0 \\ 0 & \text{otherwise} \end{cases} \tag{262}$$

$$\geqslant \begin{cases} 2^{1-\frac{2}{\psi}}(a+b)^{\frac{2}{\psi}} - b^{\frac{2}{\psi}} & \text{if } a > 0 \\ 0 & \text{otherwise} \end{cases} \tag{263}$$

$$= \begin{cases} 2^{1-\frac{2}{\psi}}(a+b)^{\frac{2}{\psi}} - b^{\frac{2}{\psi}} & \text{if } a > 0 \\ 0 & \text{if } -b < a \leqslant 0 \\ 0 & \text{otherwise} \end{cases} \tag{264}$$

$$\geqslant \begin{cases} 2^{1-\frac{2}{\psi}}(a+b)^{\frac{2}{\psi}} - b^{\frac{2}{\psi}} & \text{if } a > 0 \\ 2^{1-\frac{2}{\psi}}(a+b)^{\frac{2}{\psi}} - b^{\frac{2}{\psi}} & \text{if } -b < a \leqslant 0 \\ -b^{\frac{2}{\psi}} & \text{otherwise} \end{cases} \tag{265}$$

$$= \begin{cases} 2^{1-\frac{2}{\psi}}(a+b)^{\frac{2}{\psi}} - b^{\frac{2}{\psi}} & \text{if } a+b > 0 \\ -b^{\frac{2}{\psi}} & \text{otherwise} \end{cases} \tag{266}$$

$$= 2^{1-\frac{2}{\psi}} \max\{0, a+b\}^{\frac{2}{\psi}} - b^{\frac{2}{\psi}}, \tag{267}$$

$$\tag{268}$$

where the first inequality follows from $(x+y)^{\frac{2}{\psi}} \leqslant 2^{\frac{2}{\psi}-1}(x^{\frac{2}{\psi}} + y^{\frac{2}{\psi}})$ for $x, y \geqslant 0$, from Holder's inequality; the second inequality from observing that $2^{1-\frac{2}{\psi}}(a+b)^{\frac{2}{\psi}} - b^{\frac{2}{\psi}} \leqslant (2^{1-\frac{2}{\psi}} - 1)b^{\frac{2}{\psi}} \leqslant 0$ for $-b < a \leqslant 0$. $\square$

## G Recurrences

In this section, we provide auxiliary results about convergence rate of a certain class of recurrences that will be employed for the convergence analysis of the proposed algorithms. Specifically, we study the recurrence:

$$r_{k+1} \leqslant r_k - a \max\{0, r_k\}^{\phi} + b \tag{269}$$

for $a > 0$, $b \geqslant 0$, and $\phi \in [1, 2]$. To this end, we consider the helper sequence:

$$\begin{cases} \rho_0 = r_0 \\ \rho_{k+1} = \rho_k - a \max\{0, \rho_k\}^{\phi} + b \end{cases} \tag{270}$$

The line of the proof follows that of Montenegro et al. (2024). Let us start showing that for sufficiently small $a$, the sequence $\rho_k$ upper bounds $r_k$.

**Lemma G.1.** *If $a \leqslant \frac{1}{\phi \rho_k^{\phi-1}}$ for every $k \geqslant 0$, then, $r_k \leqslant \rho_k$ for every $k \geqslant 0$.*

*Proof.* By induction on $k$. For $k = 0$, the statement holds since $\rho_0 = r_0$. Suppose the statement holds for every $j \leqslant k$, we prove that it holds for $k + 1$:

$$\rho_{k+1} = \rho_k - a \max\{0, \rho_k\}^{\phi} + b \tag{271}$$

$$\geqslant r_k - a \max\{0, r_k\}^{\phi} + b \tag{272}$$

$$\geqslant r_{k+1}, \tag{273}$$

where the first inequality holds by the inductive hypothesis and by observing that the function $f(x) = x - a \max\{0, x\}^{\phi}$ is non-decreasing in $x$ when $a \leqslant \frac{1}{\phi \rho_k^{\phi-1}}$. Indeed, if $x < 0$, then $f(x) = x$, which is non-decreasing; if $x \geqslant 0$, we have $f(x) = x - ax^{\phi}$, that can be proved to be non-decreasing

in the interval $\left[0, (a\phi)^{-\frac{1}{\phi-1}}\right]$ simply by studying the sign of the derivative. Thus, we enforce the following requirement to ensure that $\rho_k$ falls in the non-decreasing region:

$$\rho_k \leqslant (a\phi)^{-\frac{1}{\phi-1}} \implies a \leqslant \frac{1}{\phi\rho_k^{\phi-1}}. \tag{274}$$

So does $r_k$ by the inductive hypothesis. $\qquad \square$

Thus, from now on, we study the properties of the sequence $\rho_k$. Let us note that, if $\rho_k$ is convergent, then it converges to the fixed-point $\overline{\rho}$ computed as follows:

$$\overline{\rho} = \overline{\rho} - a\max\{0, \overline{\rho}\}^\phi + b \implies \overline{\rho} = \left(\frac{b}{a}\right)^{\frac{1}{\phi}}, \tag{275}$$

having retained the positive solution of the equation only, since the negative one never attains the maximum $\max\{0, \overline{\rho}\}$. Let us now study the monotonicity properties of the sequence $\rho_k$.

**Lemma G.2.** *The following statements hold:*

- *If $r_0 > \overline{\rho}$ and $a \leqslant \frac{1}{\phi r_0^{\phi-1}}$, then for every $k \geqslant 0$ it holds that: $\overline{\rho} \leqslant \rho_{k+1} \leqslant \rho_k$.*

- *If $r_0 < \overline{\rho}$ and $a \leqslant \frac{1}{\phi\overline{\rho}^{\phi-1}}$, then for every $k \geqslant 0$ it holds that: $\overline{\rho} \geqslant \rho_{k+1} \geqslant \rho_k$.*

*Proof.* The proof is analogous to that of (Montenegro et al., 2024, Lemma F.3). $\qquad \square$

From now on, we focus on the case in which $r_0 \geqslant \overline{\rho}$, since, as we shall see later, the opposite case is irrelevant for the convergence guarantees. We now consider two cases: $b = 0$ and $b > 0$.

## G.1 Analysis when $b = 0$

From the policy optimization perspective, this case corresponds to the one in which the gradients are exact (no variance). Recall that here $\overline{\rho} = 0$. We have the following convergence result.

**Lemma G.3.** *If $a \leqslant \frac{1}{\phi r_0^{\phi-1}}$, $r_0 \geqslant 0$, and $b = 0$ it holds that:*

$$\rho_{k+1} \leqslant \begin{cases} (1-a)^{k+1} r_0 & \text{if } \phi = 1 \\ \min\left\{r_0, ((\phi-1)a(k+1))^{-\frac{1}{\phi-1}}\right\} & \text{if } \phi \in (1, 2] \end{cases}. \tag{276}$$

*Proof.* Since $r_0 \geqslant 0 = \overline{\rho}$, from Lemma G.2, we know that $\rho_k \geqslant 0$ and, thus, $\max\{0, \rho_k\} = \rho_k$. For $\phi = 1$, we have:

$$\rho_{k+1} = \rho_k - a\rho_k = (1-a)\rho_k = (1-a)^{k+1}\rho_0 = (1-a)^{k+1} r_0. \tag{277}$$

For $\phi \in (1, 2]$, we have:

$$\rho_{k+1} = \rho_k - a\rho_k^\phi. \tag{278}$$

We proceed by induction. For $k = 0$, the statement hold since $\rho_0 = r_0$ and $r_0 \leqslant (\phi a)^{-\frac{1}{\phi-1}} \leqslant ((\phi-1)a)^{-\frac{1}{\phi-1}}$ from the condition on the learning rate. Suppose the thesis holds for $j \leqslant k$, we prove it for $k + 1$. $\rho_{k+1} \leqslant r_0$ by monotonicity, and, from the inductive hypothesis:

$$\rho_{k+1} = \rho_k - a\rho_k^\phi \leqslant (\phi a k)^{-\frac{1}{\phi-1}} - a(\phi a k)^{-\frac{\phi}{\phi-1}} \tag{279}$$

$$= \underbrace{(\phi a k)^{-\frac{1}{\phi-1}} - (\phi a(k+1))^{-\frac{1}{\phi-1}} - a(\phi a k)^{-\frac{\phi}{\phi-1}}}_{(*)} + (\phi a(k+1))^{-\frac{1}{\phi-1}}. \tag{280}$$

We now prove that $(*)$ is non-positive:

$$(*) = ((\phi-1)a k)^{-\frac{1}{\phi-1}} - ((\phi-1)a(k+1))^{-\frac{1}{\phi-1}} - a((\phi-1)a k)^{-\frac{\phi}{\phi-1}} \tag{281}$$

$$= ((\phi-1)a)^{-\frac{1}{\phi-1}} k^{-\frac{\phi}{\phi-1}} \underbrace{\left(k - (k+1)\left(\frac{k}{k+1}\right)^{\frac{\phi}{\phi-1}}\right)}_{\leqslant \frac{1}{\phi-1}} - a^{-\frac{1}{\phi-1}}((\phi-1)k)^{-\frac{\phi}{\phi-1}} \tag{282}$$

$$\leqslant a^{-\frac{1}{\phi-1}} k^{-\frac{\phi}{\phi-1}} (\phi-1)^{-\frac{1}{\phi-1}} \left( \frac{1}{\phi-1} - \frac{1}{\phi-1} \right) \leqslant 0, \tag{283}$$

having observed that:

$$\sup_{k \geqslant 1} \left( k - (k+1) \left( \frac{k}{k+1} \right)^{\frac{\phi}{\phi-1}} \right) = \lim_{k \to +\infty} \left( k - (k+1) \left( \frac{k}{k+1} \right)^{\frac{\phi}{\phi-1}} \right) = \frac{1}{\phi-1}. \tag{284}$$

$\square$

## G.2   Analysis for $b > 0$

From the policy optimization perspective, this corresponds to the case in which the gradients are estimated, i.e., the variance is positive. In this case, we proceed considering the helper sequence:

$$\begin{cases} \eta_0 = \rho_0 \\ \eta_{k+1} = \left(1 - a\overline{\rho}^{\phi-1}\right)\eta_k + b & \text{if } k \geqslant 0 \end{cases}. \tag{285}$$

We show that the sequence $\eta_k$ upper bounds $\rho_k$ when $\rho_0 = r_0 \geqslant \overline{\rho}$.

**Lemma G.4.** *If $r_0 > \overline{\rho}$ and $a \leqslant \frac{1}{\phi r_0^{\phi-1}}$, then, for every $k \geqslant 0$, it holds that $\eta_k \geqslant \rho_k$.*

*Proof.* The proof is analogous to that of (Montenegro et al., 2024, Lemma F.4). $\square$

Thus, we can provide the convergence guarantee.

**Lemma G.5.** *If $a \leqslant \frac{1}{\phi r_0^{\phi-1}}$, $r_0 \geqslant 0$, and $b > 0$ it holds that:*

$$\eta_{k+1} \leqslant \left(1 - b^{1-\frac{1}{\phi}} a^{\frac{1}{\phi}}\right)^{k+1} + \left(\frac{b}{a}\right)^{\frac{1}{\phi}}. \tag{286}$$

*Proof.* By unrolling the recursion:

$$\eta_{k+1} = \left(1 - a\overline{\rho}^{\phi-1}\right)\eta_k + b \tag{287}$$

$$= \left(1 - a\overline{\rho}^{\phi-1}\right)^{k+1} r_0 + b \sum_{j=0}^{k} \left(1 - a\overline{\rho}^{\phi-1}\right)^j \tag{288}$$

$$\leqslant \left(1 - a\overline{\rho}^{\phi-1}\right)^{k+1} r_0 + b \sum_{j=0}^{+\infty} \left(1 - a\overline{\rho}^{\phi-1}\right)^j \tag{289}$$

$$= \left(1 - b^{1-\frac{1}{\phi}} a^{\frac{1}{\phi}}\right)^{k+1} + \frac{b}{a\overline{\rho}^{\phi-1}} \tag{290}$$

$$= \left(1 - b^{1-\frac{1}{\phi}} a^{\frac{1}{\phi}}\right)^{k+1} + \left(\frac{b}{a}\right)^{\frac{1}{\phi}}. \tag{291}$$

$\square$

# H   Experimental Details and Additional Results

## H.1   Experimental Details

### H.1.1   Employed Policies and Hyperpolicies

**Linear Gaussian Policy.**   A *linear parametric gaussian* policy $\pi_{\boldsymbol{\theta}} : \mathcal{S} \times \mathcal{A} \to \Delta(\mathcal{A})$ with variance $\sigma^2$ samples the actions as $a_t \sim \mathcal{N}(\boldsymbol{\theta}^\top \boldsymbol{s}_t, \sigma^2 I_{d_{\mathcal{S}}})$, where $\boldsymbol{s}_t$ is the observed state at time $t$ and $\boldsymbol{\theta}$ is the parameter vector.

**Tabular Softmax Policy.** A *tabular softmax* policy $\pi_{\boldsymbol{\theta}} : \mathcal{S} \times \mathcal{A} \rightarrow \Delta(\mathcal{A})$ with a temperature constant $\tau$ is such that:

$$\pi_{\boldsymbol{\theta}}(\boldsymbol{a}_j | \boldsymbol{s}_i) = \frac{\exp\left(\frac{\boldsymbol{\theta}_{i,j}}{\tau}\right)}{\sum_{z=1}^{|\mathcal{A}|} \exp\left(\frac{\boldsymbol{\theta}_{i,z}}{\tau}\right)}, \tag{292}$$

where $\boldsymbol{\theta}_{i,j}$ is the parameter associated with the $i$-th state and the $j$-th action. Notice that the total number of parameters for this kind of policy is $|\mathcal{S}||\mathcal{A}|$.

**Linear Deterministic Policy.** A *linear parametric deterministic* policy $\mu_{\boldsymbol{\theta}} : \mathcal{S} \times \mathcal{A} \rightarrow \mathcal{A}$ samples the actions as $\boldsymbol{a}_t = \boldsymbol{\theta}^\top \boldsymbol{s}_t$, where $\boldsymbol{s}_t$ is the observed state at time $t$ and $\boldsymbol{\theta}$ is the parameter vector.

**Gaussian Hyperpolicy.** A *parametric gaussian* hyperpolicy $\nu_{\boldsymbol{\rho}} : \mathcal{R} \rightarrow \Delta(\Theta)$ with variance $\sigma^2$ samples the parameters $\boldsymbol{\theta}$ for the underlying generic parametric policy $\pi_{\boldsymbol{\theta}}$ as $\boldsymbol{\theta}_t \sim \mathcal{N}(\boldsymbol{\rho}, \sigma^2 I_{d_\mathcal{R}})$, where $\boldsymbol{\rho}$ is the parameter vector for the hyperpolicy.

### H.1.2 Environments

**Discrete Grid World with Walls.** Discrete Grid World with Walls (DGWW) is a simple discrete environment we employed to compare `C-PGAE` against the sample-based versions of NPG-PD (Ding et al., 2020, Appendix H) and RPG-PD (Ding et al., 2024, Appendix C.9). DGWW is a grid-like bidimensional environment in which an agent can assume only integer coordinate positions and in which an agent can play four actions stating whether to go up, right, left, or down. The goal is to reach the center of the grid performing the minimum amount of steps, begin the initial state uniformly sampled among the four vertices of the grid. The agent is rewarded negatively and proportionally to its distance from the center, where the reward is $0$. Around the goal state there is a "U-shaped" obstacle with an opening on the top side. In particular, when the agent lands in a state in which the wall is present, it receives a cost of $1$, otherwise the cost signal is always equal to $0$. In our experiments, we employed a DGWW environment of such a kind, with $|\mathcal{S}| = 49$, i.e., with each dimension with length equal to $7$.

**Linear Quadratic Regulator with Costs.** The Linear Quadratic Regulator (LQR, Anderson and Moore, 2007) is a continuous environment we employed in the regularization sensitivity study of `C-PGAE` and `C-PGPE`, and in the comparison among the same algorithms against the sample-based version of NPG-PD2 (Ding et al., 2022, Algorithm 1) and its ridge-regularized version RPG-PD2 (not provided by the authors, but designed by us). LQR is a dynamical system governed by the following state evolution:

$$\boldsymbol{s}_{t+1} = A\boldsymbol{s}_t + B\boldsymbol{a}_t, \tag{293}$$

where $A \in \mathbb{R}^{d_\mathcal{S} \times d_\mathcal{S}}$ and $B \in \mathbb{R}^{d_\mathcal{S} \times d_\mathcal{A}}$.

In the standard version of the environment, the reward is computed at each step as:

$$r_t = -\boldsymbol{s}_t^\top R \boldsymbol{s}_t - \boldsymbol{a}_t^\top Q \boldsymbol{a}_t, \tag{294}$$

where $R \in \mathbb{R}^{d_\mathcal{S} \times d_\mathcal{S}}$ and $Q \in \mathbb{R}^{d_\mathcal{A} \times d_\mathcal{A}}$.

We modified this version of the LQR environment introducing costs. In particular, in our *CostLQR*, the state evolution is treated as in the original case, while the reward at step $t$ is computed as:

$$r_t = -\boldsymbol{s}_t^\top R \boldsymbol{s}_t, \tag{295}$$

where $R \in \mathbb{R}^{d_\mathcal{S} \times d_\mathcal{S}}$. Moreover, we added a cost signal $c$ which is computed as follows at every time step $t$:

$$c_t = \boldsymbol{a}_t^\top Q \boldsymbol{a}_t, \tag{296}$$

where $Q \in \mathbb{R}^{d_\mathcal{A} \times d_\mathcal{A}}$.

In our experiments, we consider a *CostLQR* environment whose main characteristics are reported in Table 5.

Additionally, we considered a uniform initial state distribution in $[-3, 3]$ and the following matrices:

$$A = B = 0.9 \begin{bmatrix} 1 & 0 \\ 0 & 1 \end{bmatrix}, \qquad Q = \begin{bmatrix} 0.9 & 0 \\ 0 & 0.1 \end{bmatrix}, \qquad R = \begin{bmatrix} 0.1 & 0 \\ 0 & 0.9 \end{bmatrix}. \tag{297}$$

| Environment | State Dimension $d_{\mathcal{S}}$ | Action Dimension $d_{\mathcal{A}}$ | Action Range $[a_{\min}, a_{\max}]$ | State Range $[s_{\min}, s_{\max}]$ |
|:---:|:---:|:---:|:---:|:---:|
| CostLQR | 2 | 2 | $(-\infty, +\infty)$ | $(-\infty, +\infty)$ |
| Swimmer-v4 | 8 | 2 | $[-1, 1]$ | $(-\infty, +\infty)$ |
| Hopper-v4 | 11 | 3 | $[-1, 1]$ | $(-\infty, +\infty)$ |

Table 5: Main features of *CostLQR*, *Swimmer-v4*, and *Hopper-v4*.

**MuJoCo with Costs.** For our experiments on risk minimization, we utilized environments from the MuJoCo control suite (Todorov et al., 2012), which offers a variety of continuous control environments. To tailor these environments to our specific requirements, we introduced a cost function that represents the energy associated with the control actions. In standard MuJoCo environments, a portion of the reward is typically calculated as the cost of the control action, which is proportional to the deviation of the chosen action from predefined action bounds. In our MuJoCo modification, at each time step we make the environment return a cost computed as:

$$\|\boldsymbol{a}_t - \min\{\max\{\boldsymbol{a}_t, a_{\min}\}, a_{\max}\}\|_2, \tag{298}$$

where $a_{\min}$ and $a_{\max}$ are respectively the bounds for the minimum and maximum value for each component of the action vector. Then, the action $\min\{\max\{\boldsymbol{a}_t, a_{\min}\}, a_{\max}\}$ is passed to the environment. In our experiment we consider *Swimmer-v4* and *Hopper-v4* MuJoCo environments, whose main features are summarized in Table 5.

## H.2 Details for the comparison against the baselines in DGWW

In Section 5, we compare our proposal `C-PGAE` against the sample-based versions of NPG-PD (Ding et al., 2020, Appendix H) and RPG-PD (Ding et al., 2024, Appendix C9). The environment in which the methods are tested is the Discrete Grid World with Walls (DGWW, see Appendix H) with a horizon of $T = 100$. In this experiment, the methods aim at learning the parameters of a tabular softmax policy with 196 parameters, maximizing the trajectory reward while considering a single constraint on the average trajectory cost, for which we set a threshold $b = 0.2$. All the methods were run for $K = 3000$ iterations with a batch size of $N = 10$ trajectories per iteration, and with constant learning rates. In particular, for both `C-PGAE` and NPG-PD, we employed $\zeta_{\boldsymbol{\theta}} = 0.01$ and $\zeta_{\boldsymbol{\lambda}} = 0.1$, while for RPG-PD we selected $\zeta_{\boldsymbol{\theta}} = 0.01$ and $\zeta_{\boldsymbol{\lambda}} = 0.01$. For `C-PGAE` and RPG-PD we used a regularization constant $\omega = 10^{-4}$. All the details about the experimental setting are summarized in Table 6. We would like to stress that, as prescribed by the respective convergence theorems, we chose a two time-scales learning rate approach for `C-PGAE` and a single time-scale one for RPG-PD. Figure 2a shows the performance curves (i.e., the one associated with the objective function and the cost ones). As can be noticed, `C-PGAE` manages to strike the objective of the constrained optimization problem with less trajectories. Indeed, the sample-based NPG-PD requires to estimate the value and the action-value functions for all the states and state-action pairs, resulting in analyzing $|\mathcal{S}| + |\mathcal{S}||\mathcal{A}|$ additional trajectories w.r.t. `C-PGAE` for every iteration of the algorithm. The sample-based RPG-PD also requires additional trajectories to be analyzed, which in practice, for a correct learning behavior, result to be the same in number to the extra ones analyzed by NPG-PD.

## H.3 Details for the comparison against baselines in CostLQR

In Section 5, we compare our proposals `C-PGAE` and `C-PGPE` against the continuous sample-based version of NPG-PD (Ding et al., 2022, Algorithm 1) with works with generic policy parameterizations. In the following, we refer to this version of NPG-PD as NPG-PD2. Moreover, we added a ridge-regularized version of NPG-PD2, that we call RPG-PD2, to resemble the type of regularization we employed for our proposed methods. For all the regularized methods (i.e., `C-PGAE`, `C-PGPE`, and RPG-PD2) we selected as regularization constant $\omega = 10^{-4}$. The setting for this experiment considers a bidimensional LQR environment with a single cost over the provided actions (see Appendix H) and with a fixed horizon $T = 50$. Here, the methods aim at maximizing the average reward over trajectories, while keeping the average cost over trajectories under the threshold $b = 0.9$. In particular, `C-PGAE` learns the parameters of a linear gaussian policy with a variance $\sigma_{\mathrm{A}}^2 = 10^{-3}$ and employing a learning rate schedule governed by the Adam scheduler (Kingma and Ba, 2015) with $\zeta_{\boldsymbol{\theta},0} = 0.001$ and $\zeta_{\boldsymbol{\lambda},0} = 0.01$. `C-PGPE` learns the parameters of a gaussian hyperpolicy, with a variance $\sigma_{\mathrm{P}}^2 = 10^{-3}$, which samples the parameters of a deterministic linear policy. It employs a learning rate schedule

| Details for the Comparison in DGWW Experiment | |
|---|---|
| Environment | DGWW |
| Horizon | $T = 100$ |
| Policy | Tabular Softmax |
| Constraint Threshold | $b = 0.2$ |
| Iterations | $K = 3000$ |
| Batch Size | $N = 10$ |
| Learning Rates C-PGAE | $\zeta_{\boldsymbol{\theta}} = 0.01$ and $\zeta_{\boldsymbol{\lambda}} = 0.1$ |
| Learning Rates NPG-PD | $\zeta_{\boldsymbol{\theta}} = 0.01$ and $\zeta_{\boldsymbol{\lambda}} = 0.1$ |
| Learning Rates RPG-PD | $\zeta_{\boldsymbol{\theta}} = 0.01$ and $\zeta_{\boldsymbol{\lambda}} = 0.01$ |
| Regularization C-PGAE | $\omega = 10^{-4}$ |
| Regularization RPG-PD | $\omega = 10^{-4}$ |

Table 6: Details for the comparison of C-PGAE against NPG-PD and RPG-PD in DGWW.

governed by Adam too with $\zeta_{\boldsymbol{\rho},0} = 0.001$ and $\zeta_{\boldsymbol{\lambda},0} = 0.01$. Both C-PGAE and C-PGPE were run for $K = 6000$ iterations with a batch of $N = 100$ trajectories per iteration. NPG-PD2 and RPG-PD2 are both actor-critic methods which were run for $K = 1000$ iterations with a batch size of $N = 600$ trajectories per iteration. In particular, among the trajectories of the reported batch size, $N_1 = 500$ were used for the inner critic-loop, while $N_2 = 100$ for performance and cost estimations. The inner loop step size was selected constant, as prescribed by the original algorithm, and with a value $\alpha = 10^{-5}$. Furthermore, since such methods were designed for infinite-horizon discounted environments, we tested them on the same LQR as for C-PGAE and C-PGPE, but leaving $T = +\infty$ and $\gamma = 0.98$ (the effective horizon is $(1 - \gamma)^{-1} = 50$). The step sizes for the primal and dual variables updates were governed by Adam with $\zeta_{\boldsymbol{\theta},0} = 0.003$ and $\zeta_{\boldsymbol{\lambda},0} = 0.01$. As for C-PGAE, both NPG-PD2 and RPG-PD2 aimed at learning the parameters of a linear gaussian policy, with variance $\sigma_{\mathrm{A}}^2 = 10^{-3}$. All the details about this experiment are summarized in Table 7. Figure 2b reports the learning curves for the average return and the cost over trajectories. As can be seen, our methods manage to solve the constrained optimization problem at hand by leveraging on less trajectories. Indeed, NPG-PD2 and RPG-PD2 suffer the inner critic loop, which add additional trajectories to be analyzed per iteration (in this specific case $N_1 = 500$). We would like to stress that the actor-critic methods were very sensible to the hyperparameters selection, especially the length and the step size of the inner loop.

### H.4   Details for the risk-constrained experiment on Swimmer-v4

In Section 5, we presented an experiment comparing what happens when considering the two exploration approaches of C-PGAE and C-PGPE and on different risk-constrained optimization problems on the cost version of *Swimmer-v4* (details in Appendix H). In particular, we considered such an environment with a horizon of $T = 100$. Both the algorithms were run for $K = 3000$ iterations, with a batch size of $N = 100$ trajectories per iteration, and with step sizes governed by the Adam scheduler. Moreover, they had a regularization constant $\omega = 10^{-4}$.

C-PGAE employed a *linear gaussian policy* with variance $\sigma_{\mathrm{A}}^2 = 1$. In Table 8, we list the characteristics for all the experiments with all the risk measures.

On the other hand, C-PGPE employed a *gaussian hyperpolicy* with variance $\sigma_{\mathrm{P}}^2 = 0.01$. Such an hyperpolicy sampled parameters for an underlying *linear deterministic policy*. In Table 9, we list the characteristics for all the experiments with all the risk measures.

As also highlighted in the main paper, C-PGPE delivers an hyperpolicy which samples (after having sampled the parameters for the underlying policy) actions whose costs are always under the fixed threshold. Moreover, by considering risk measures different from the *average cost* one, the empirical distribution of costs shows that these are way lighter w.r.t. the ones associated with the deployment of

| Details for the Comparison in CostLQR Experiment | |
|---|---|
| Environment | Bidimensional CostLQR |
| Horizon | $T = 50$ |
| (Hyper)Policy | Linear Gaussian with $\sigma^2 = 10^{-3}$ |
| Constraint Threshold | $b = 0.9$ |
| Iterations (C-PGPE and C-PGAE) | $K = 6000$ |
| Iterations (NPG-PD2 and RPG-PD2) | $K = 1000$ |
| Batch Size (C-PGPE and C-PGAE) | $N = 100$ |
| Batch Size (NPG-PD2 and RPG-PD2) | $N = 600$ |
| Learning Rate (Adam) C-PGPE | $\zeta_{\rho,0} = 10^{-3}$ and $\zeta_\lambda = 10^{-2}$ |
| Learning Rates (Adam) C-PGAE | $\zeta_{\theta,0} = 10^{-3}$ and $\zeta_\lambda = 10^{-2}$ |
| Learning Rates (Adam) NPG-PD2 | $\zeta_\theta = 3 \cdot 10^{-3}$ and $\zeta_\lambda = 10^{-2}$ |
| Learning Rates (Adam) RPG-PD2 | $\zeta_\theta = 3 \cdot 10^{-3}$ and $\zeta_\lambda = 10^{-2}$ |
| Regularization C-PGPE | $\omega = 10^{-4}$ |
| Regularization C-PGAE | $\omega = 10^{-4}$ |
| Regularization RPG-PD2 | $\omega = 10^{-4}$ |

Table 7: Details for the comparison of C-PGPE and C-PGAE against NPG-PD2 and RPG-PD2 in a bodomensional CostLQR.

| Risk Measure | Risk Parameter | $b$ | $\zeta_{\theta,0}$ | $\zeta_{\lambda,0}$ | $\zeta_{\eta,0}$ |
|---|---|---|---|---|---|
| Average Cost | ✗ | 50 | $10^{-3}$ | $10^{-2}$ | ✗ |
| $\text{CVaR}_\alpha$ | $\alpha = 0.95$ | 50 | $10^{-4}$ | $10^{-1}$ | $10^{-1}$ |
| MV | $\kappa = 0.5$ | 50 | $10^{-4}$ | $10^{-1}$ | $10^{-1}$ |
| Chance | $n = 50$ | 0.05 | $10^{-4}$ | $10^{-1}$ | $10^{-1}$ |

Table 8: Parameters of the risk measures employed in the experiment on *Swimmer-v4* with C-PGAE.

| Risk Measure | Risk Parameter | $b$ | $\zeta_{\rho,0}$ | $\zeta_{\lambda,0}$ | $\zeta_{\eta,0}$ |
|---|---|---|---|---|---|
| Average Cost | ✗ | 50 | $10^{-3}$ | $10^{-2}$ | ✗ |
| $\text{CVaR}_\alpha$ | $\alpha = 0.95$ | 50 | $10^{-3}$ | $10^{-2}$ | $10^{-3}$ |
| MV | $\kappa = 0.5$ | 50 | $10^{-3}$ | $10^{-2}$ | $10^{-3}$ |
| Chance | $n = 50$ | 0.05 | $10^{-3}$ | $10^{-2}$ | $10^{-3}$ |

Table 9: Parameters of the risk measures employed in the experiment on *Swimmer-v4* with C-PGPE.

an hyperpolicy learned considering average cost constraints. C-PGAE also shows results of this kind. However, the displacement between the final empirical cost distributions is not as marked as the one shown with C-PGPE. The exception is from the MV risk measure side, which seems to make C-PGAE learn a policy able to pay lighter costs, but resulting in poor-performing policy. All the other found (hyper)policies, instead, provide similar performance scores.

## H.5 Risk-constrained experiment on Hopper-v4

Here, we present a similar experiment to the one shown on *Swimmer-v4*. Also in this case, we compare what happens when considering the two exploration approaches of `C-PGAE` and `C-PGPE` and on different risk-constrained optimization problems, but on the cost version of *Hopper-v4* (details in Appendix H). The experimental setting is quite the same of the one considered above (i.e., $T = 100$, $K = 3000$, $N = 100$, $\omega = 10^{-4}$).

`C-PGAE` employed a *linear gaussian policy* with variance $\sigma_A^2 = 1$. The characteristics for all the risk measures' experiments are presented in Table 10. For the learning rates schedules, we employed the Adam scheduler.

| Risk Measure | Risk Parameter | $b$ | $\zeta_{\boldsymbol{\theta},0}$ | $\zeta_{\boldsymbol{\lambda},0}$ | $\zeta_{\boldsymbol{\eta},0}$ |
|---|---|---|---|---|---|
| Average Cost | ✗ | 100 | $10^{-2}$ | $10^{-1}$ | ✗ |
| $\text{CVaR}_\alpha$ | $\alpha = 0.95$ | 100 | $10^{-3}$ | $10^{-1}$ | $10^{-1}$ |
| MV | $\kappa = 0.5$ | 100 | $10^{-4}$ | $10^{-1}$ | $10^{-1}$ |
| Chance | $n = 100$ | 0.05 | $10^{-4}$ | $10^{-1}$ | $10^{-1}$ |

Table 10: Parameters of the risk measures employed in the experiment on *Hopper-v4* with `C-PGAE`.

`C-PGPE` employed a *gaussian hyperpolicy* with variance $\sigma_P^2 = 0.1$ over an underlying *linear deterministic policy*. The characteristics for all the experiments with all the risk measures are summarized in Table 11. For the learning rates schedules, we employed the Adam scheduler.

| Risk Measure | Risk Parameter | $b$ | $\zeta_{\boldsymbol{\rho},0}$ | $\zeta_{\boldsymbol{\lambda},0}$ | $\zeta_{\boldsymbol{\eta},0}$ |
|---|---|---|---|---|---|
| Average Cost | ✗ | 100 | $10^{-2}$ | $10^{-1}$ | ✗ |
| $\text{CVaR}_\alpha$ | $\alpha = 0.95$ | 100 | $10^{-2}$ | $10^{-1}$ | $10^{-2}$ |
| MV | $\kappa = 0.5$ | 100 | $10^{-2}$ | $10^{-1}$ | $10^{-2}$ |
| Chance | $n = 100$ | 0.05 | $10^{-2}$ | $10^{-1}$ | $10^{-2}$ |

Table 11: Parameters of the risk measures employed in the experiment on *Hopper-v4* with `C-PGPE`.

Figure 5 shows the empirical distributions of costs over 100 trajectories of the learned (hyper)policies via `C-PGPE` and `C-PGAE`. `C-PGPE` shows a behavior that is quite the same as the one shown in the *Swimmer-v4* environment. Indeed, by considering constraints on the $\text{CVaR}_\alpha$, the MV, or on the Chance, the learned hyperpolicy selects parameters for the underlying policy selecting actions that pay a way lighter cost w.r.t. the ones provided by an hyperpolicy learned by considering constraints on average costs. It is worth noticing that the lightest costs are observed when considering the $\text{CVaR}_\alpha$ or the MV risk measures. In this case, `C-PGAE` shows a similar behavior to `C-PGPE` for what concern the distance in costs between the ones observed by learning considering constraints on the average cost and the ones observed by learning with other risk measures. The lightest costs here can be observed by learning with chance constraints, while, by considering constraints on the MV, the observed costs exceed the ones observed under average cost constraints. For what concern the performances of the learned (hyper)policies (see Table 4c, `C-PGPE` exhibit high-performing hyperpolicies under average cost or chance constraints, while the ones under $\text{CVaR}_\alpha$ and MV shows similar (low) performance scores. On the other hand, for `C-PGAE` the learned policy under $\text{CVaR}_\alpha$ exhibits the same good-performing behavior as the one of the policy learned under average cost constraints.

## H.6 Regularization Sensitivity Study

Here, we study the sensitivity of `C-PGAE` and `C-PGPE` w.r.t. the regularization term $\omega$. We tried the algorithms on a bidimensional LQR environment which has been modified to output a cost signal based on the selected action. For the environment at hand, we considered a horizon $T = 50$. We run

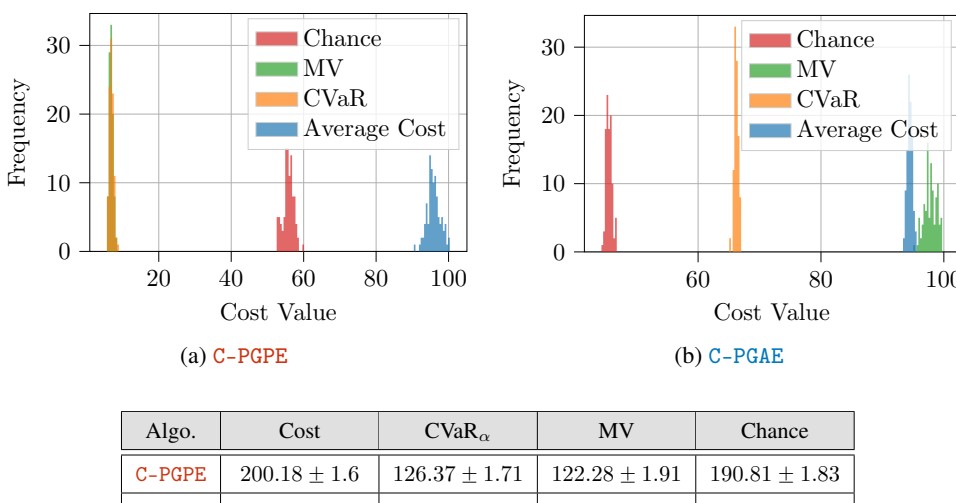

(a) C-PGPE

(b) C-PGAE

| Algo. | Cost | CVaR$_\alpha$ | MV | Chance |
|-------|------|---------------|-----|--------|
| C-PGPE | $200.18 \pm 1.6$ | $126.37 \pm 1.71$ | $122.28 \pm 1.91$ | $190.81 \pm 1.83$ |
| C-PGAE | $228.04 \pm 0.48$ | $216.81 \pm 0.46$ | $136.45 \pm 1.35$ | $159.25 \pm 2.06$ |

(c) Avg. Ret. (5 runs, mean $\pm$ std).

Figure 5: Cost distributions with (hyper)policies learned considering different risk measures (5 runs).

| Details for the Regularization Sensitivity Study in CostLQR Experiment | |
|---|---|
| Environment | Bidimensional CostLQR |
| Horizon | $T = 50$ |
| (Hyper)Policy | Linear Gaussian with $\sigma^2 = 10^{-3}$ |
| Constraint Threshold | $b = 0.2$ |
| Iterations | $K = 10^4$ |
| Batch Size | $N = 100$ |
| Learning Rate (Adam) C-PGPE | $\zeta_{\rho,0} = 10^{-3}$ and $\zeta_\lambda = 10^{-2}$ |
| Learning Rates (Adam) C-PGAE | $\zeta_{\theta,0} = 10^{-3}$ and $\zeta_\lambda = 10^{-2}$ |
| Regularization | $\omega \in \left\{0, 10^{-4}, 10^{-2}\right\} 10^{-4}$ |

Table 12: Details for the reguarization sensitivity study of C-PGPE and C-PGAE in a bodomensional CostLQR.

both the algorithms for $K = 10^4$ iterations, with a batch size $N = 100$ trajectories per iteration, and with a varying regularization term such that $\omega \in \{0, 10^{-4}, 10^{-2}\}$. We considered a single constraint on the average trajectory cost, for which we set a threshold $b = 0.2$. For the step size schedules, we employed Adam (Kingma and Ba, 2015) with initial rates $\zeta_{\rho,0} = \zeta_{\theta,0} = 10^{-3}$ and $\zeta_{\lambda,0} = 10^{-2}$. Moreover, in this specific experiment C-PGAE employed a linear gaussian policy with a variance $\sigma_A^2 = 10^{-3}$. On the other hand, C-PGPE employed a linear gaussian hyperpolicy with a variance $\sigma_P^2 = 10^{-3}$ over a linear deterministic policy. The experimental environment is summarized in Table 12. Figures 7 and 6 show the Lagrangian curves, the performance ones (i.e., the one associated with the objective function), and the cost-related ones. From the shown curves it is possible to notice that, for both C-PGAE and C-PGPE, a higher regularization ($\omega = 10^{-2}$) corresponds to a higher bias w.r.t. the constraint satisfaction. This bias is compliant with what shown by Theorem 3.1, indeed, the higher the regularization, the higher the constraint threshold should be made stricter. Finally, we report in Figure 8 the evolution of the values of the Lagrangian multipliers $\lambda$ during the learning. As expected from the theory, for both C-PGAE and C-PGPE a higher regularization leads to have smaller values of $\lambda$. Moreover, we empirically notice that C-PGAE reaches higher value of $\lambda$ w.r.t. the ones seen by C-PGPE.

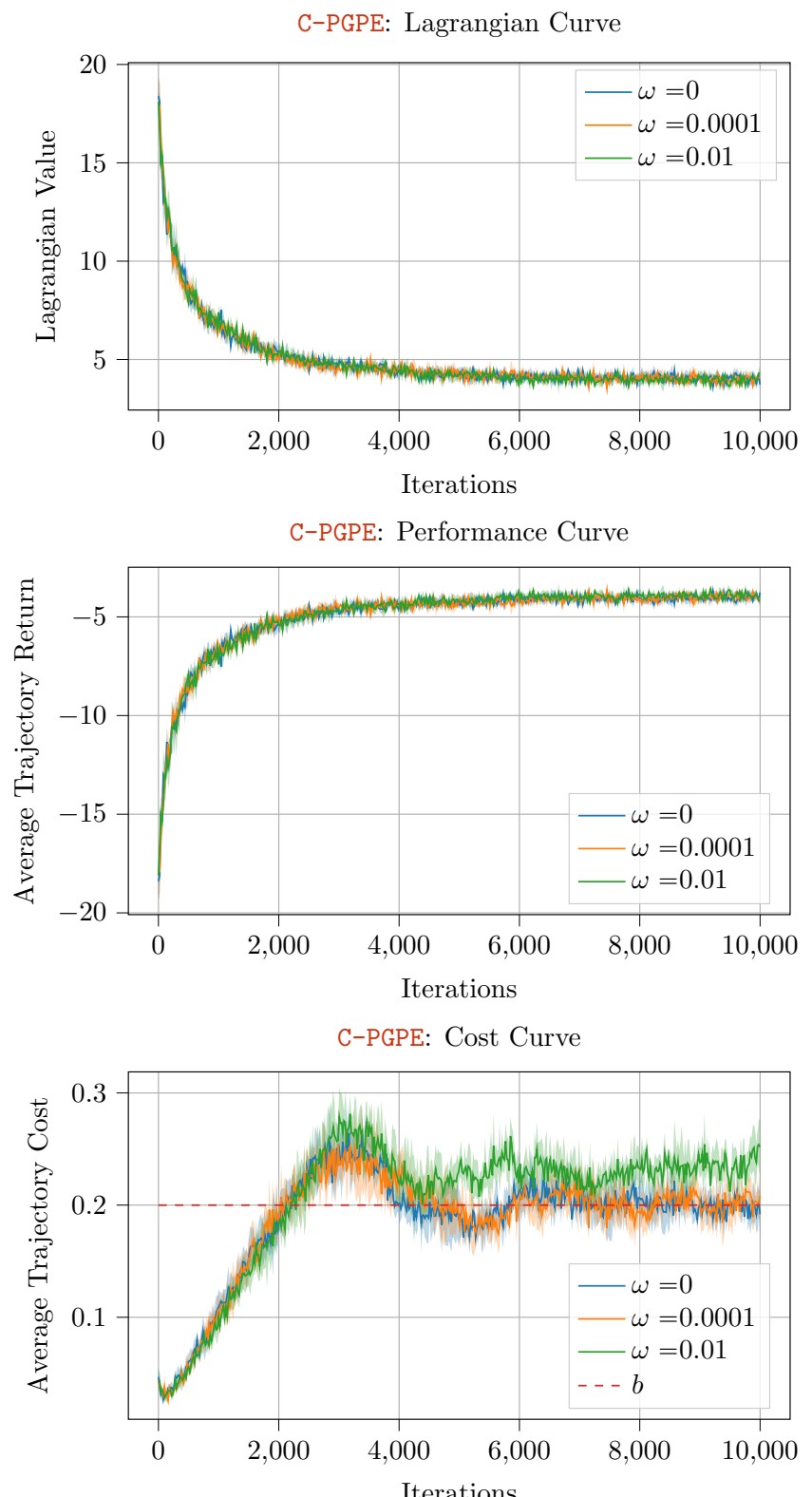

Figure 6: Lagrangian, performance and cost curves for C-PGPE over *CostLQR* with regularization values $\omega \in \{0, 10^{-4}, 10^{-2}\}$ (5 runs, mean $\pm$ 95% C.I.).

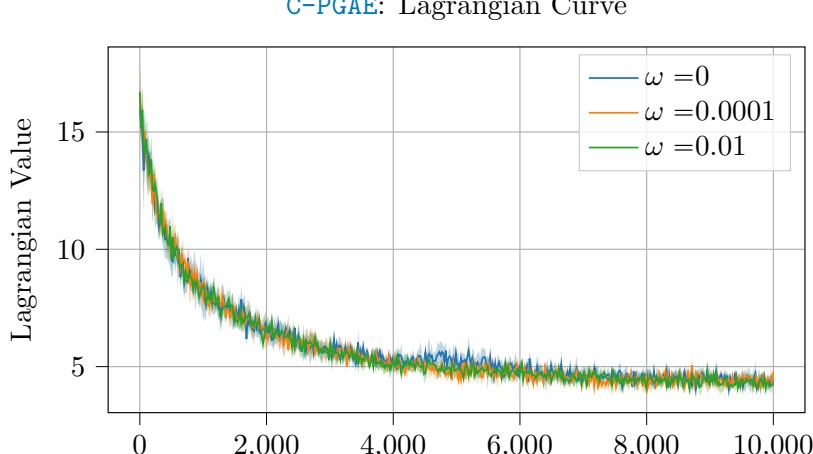

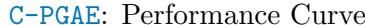

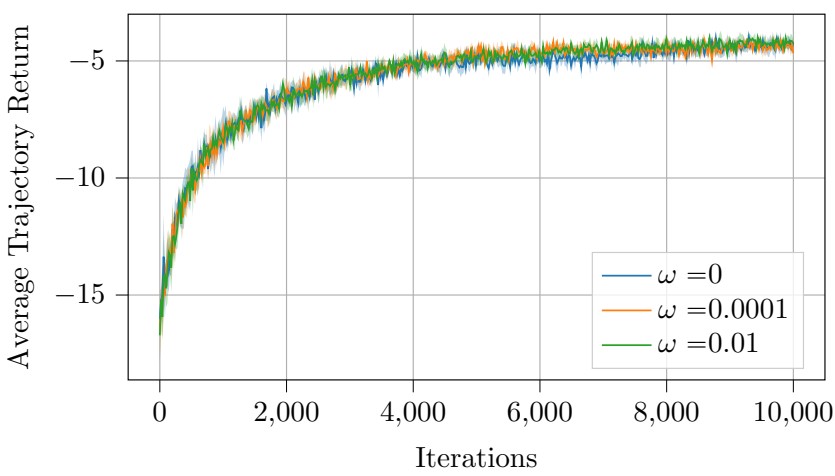

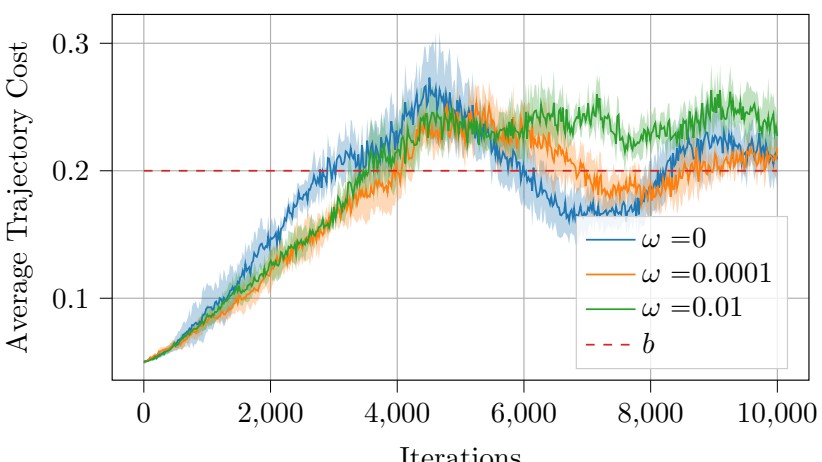

Figure 7: Lagrangian, performance and cost curves for `C-PGAE` over *CostLQR* with regularization values $\omega \in \{0, 10^{-4}, 10^{-2}\}$ (5 runs, mean $\pm$ 95% C.I.).

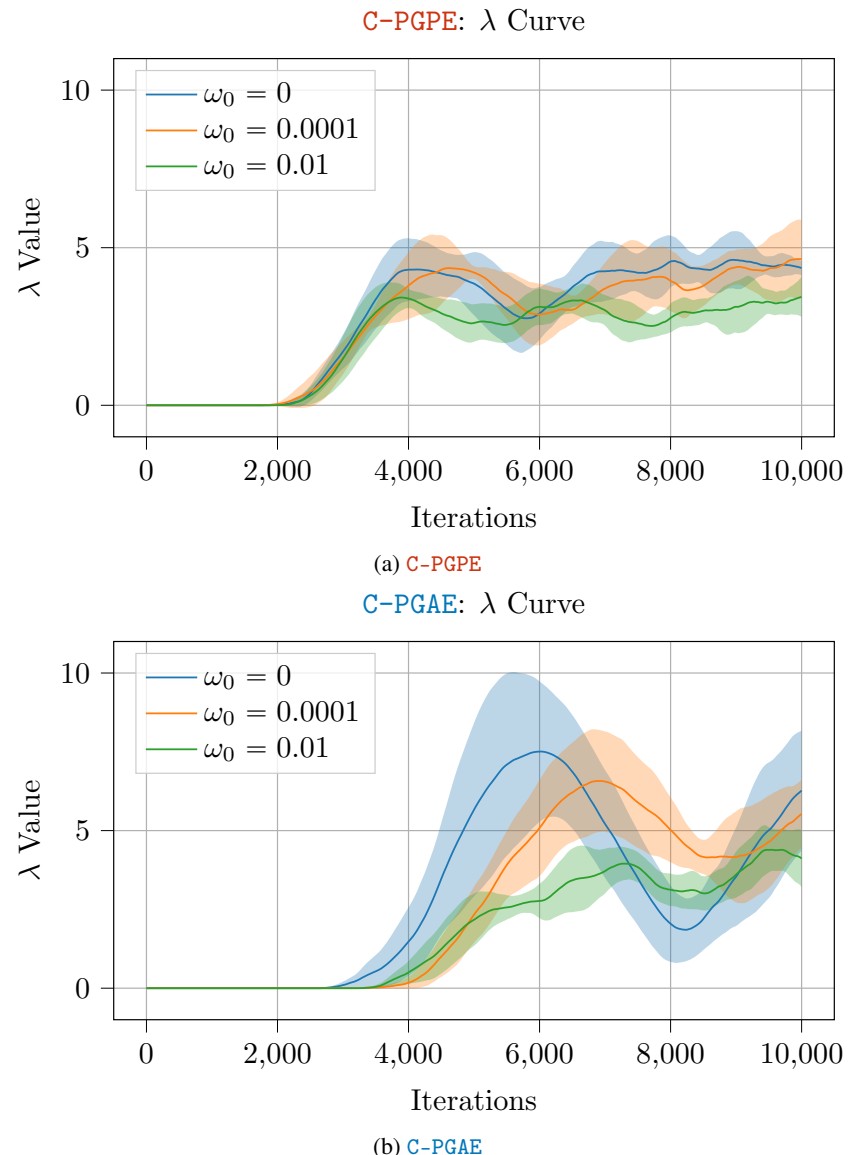

(a) C-PGPE

(b) C-PGAE

Figure 8: $\lambda$ curves for C-PGPE and C-PGAE over *CostLQR* with regularization values $\omega \in \{0, 10^{-4}, 10^{-2}\}$ (5 runs, mean $\pm$ 95% C.I.).

## H.7 Computational Resources

All the experiments were run on a 2019 16-inches MacBook Pro. The machine was equipped as follows:

| CPU | RAM | GPU |
|---|---|---|
| Intel Core i7 (6 cores, 2.6 GHz) | 16 GB 2667 MHz DDR4 | Intel UHD Graphics 630 1536 MB |

In particular, $N = 100$ trajectories of the MuJoCo environments with $T = 100$ scored $\approx 2$ iterations per second for C-PGAE, while $\approx 3$ iterations per second for C-PGPE. $N = 100$ trajectories of the *CostLQR* environment with $T = 100$ scored $\approx 5$ and $\approx 8$ iterations per second respectively for C-PGAE and C-PGPE. All the performances are to be considered with a parallelization over 10 CPU cores.

