# OpenReview forum: "Last-Iterate Global Convergence of Policy Gradients for Constrained Reinforcement Learning"
_NeurIPS.cc/2024/Conference — NeurIPS 2024 poster_

### Official Review · Reviewer_zSqV · 2024-06-12

**Soundness:** 3
**Presentation:** 3
**Contribution:** 2
**Rating:** 4
**Confidence:** 4

**Summary:**

The paper targets the constrained RL problem and provides a primal-dual method C-PG to solve it. The proposed method is extended to C-PGAE and C-PGPE to handle the constraint cases with risk measures.  The paper provides a theoretical analysis of the global last-iterate convergence guarantees toward C-PG and empirically tests the C-PGAE and C-PGPE.

**Strengths:**

The paper is well-written. All the assumptions are standard in literatures or justified. The theoretical result is rigorous.

**Weaknesses:**

The main weakness of the paper comes from the technical novelty. From the algorithm design, the primal-dual method is widely used in constrained optimization problems and constrained RL problems, and the regularization term is also not new in constrained optimization problems [1]. The proposed algorithm basically follows the existing methods without a significant novel design. The algorithm can be naturally adapted to C-PGAE and C-PGPE to handle the constraint cases with risk measures due to the well-established policy gradient for the case. From the theoretical analysis, assumptions 1,2, 3, and 4 are standard assumptions for constrained optimization problems. With these assumptions, the constrained RL problem is transferred to a pure constrained optimization problem. Although the assumptions are justified in RL, I believe there are lots of theoretical results for constrained optimization problems with these assumptions, and can be directly used in constrained RL. Therefore, the theoretical novelty may also be marginal.

Although the paper claims that it addresses some theoretical limitations of previous works, these limitations are somehow avoided, rather than being addressed. For example, the paper does not require the softmax policy, but assumption 2 is verified under the softmax policy. The provided convergence rates do not depend on the problem dimension, but the problem dimension may be included in the constants of assumptions 2, 3, and 4.

The experiment scenarios are relatively simple, but it is acceptable if the theoretical result is sufficiently solid.





[1] Khuzani, Masoud Badiei, and Na Li. "Distributed regularized primal-dual method: Convergence analysis and trade-offs."

**Questions:**

As mentioned in Weaknesses, as the dimension-free property of the proposed algorithm is claimed many times in the paper, have you proven that the problem dimension is independent of the constants of assumptions 2, 3, and 4? What is the reason and the theoretical intuition that it is dimension-free?

How does the convergence rate change after introducing the regularization term for the last-iteration convergence? As it introduces an extra error term of $w$ in Theorem 3.1, it may deteriorate the convergence rate.

I am willing to increase my score if weaknesses and questions are addressed.

**Limitations:**

No limitations of the paper are explicitly stated by the author.

---

> ### Author Rebuttal · Authors · 2024-08-04
>
> We thank the Reviewer for the time spent reviewing our paper and for recognizing that our work is well-written and that our theoretical results are rigorous. Below, we address the raised issues.
>
> **General novelty.** To the best of our knowledge, our paper is the first to introduce a **general framework** for CRL with primal-dual PGs, that is able to address simultaneously: (i) **continuous action and state spaces** (more general than the tabular case considered by some previous works), (ii) multiple constraints, (iii) **generic parameterizations of (hyper)policies** (strictly generalizing the approaches based on softmax), (iv) **inexact gradients** (avoiding irrealistic assumption of exact gradients of some previous works), (v) **parameter-based exploration** (action-based only was considered in previous works); (vi) delivering **global last-iterate convergence guarantees** (guarantee stronger than convergence *on average*).
>
> **Algorithmic novelty and assumptions role.** Algorithmically, while the primal-dual PG approach is well-explored [1,2,4,5], **PB exploration has not been addressed until now**. Our main contribution is the **theoretical analysis**, which provides state-of-the-art results and addresses some limitations discussed earlier.
> Under our assumptions (which are standard as recognized by the Reviewer [1,4,5,6,7]), we remark that **CRL does not reduce to a "pure constrained optimization" problem**. Indeed, with inexact gradients, the problem remains a learning one due to the need for **gradient estimation**, which affects algorithmic choices like the learning rates tuning.
>
> **Technical novelty.** We stress that our paper provides novel technical contributions, which we comment on in the following.
> We employ the **$\psi$-gradient domination** (asm. 3.2) on the Lagrangian function w.r.t. the parameters of the (hyper)policy to be learned, rather than focusing on specific classes of policies or on approximation error assumptions (asm. 2 of [1]). Moreover, we study the convergence of a new and different **potential function** $\mathcal{P}_k$ (Sec. 3.3) and we prove that when it is bounded, then both the objective function gap and the constraint violations are bounded (**thr. 3.1**). Finally, a new technical challenge we had to face was the study of the recursion appearing in Eq. (215). For this, we derived the technical **lem. F.1** and conducted a quite extensive study of the recurrences of the form $r\_{k+1} \le r\_k - a \\max\\{0,r\_k\\}^\phi + b$ in **apx. G**. These results and theoretical tools are, in our opinion, novel and of potential *independent interest*.
>
>
> **Addressing of theoretical limitations of other works.** We address a **more general** setting than that of softmax policies. While our asm. 3.2, as noted by the Reviewer, is satisfied by the softmax policy, **Remark 3.1 shows that it applies beyond softmax policies**, thereby **generalizing and unifying** the setting of previous works. Additionally, we tackle other challenges, including the use of inexact gradients, handling multiple constraints, and operating in a continuous setting with $|\mathcal{S}| = |\mathcal{A}| = \infty$.
>
> **Dimension-free property.** Thank you for highlighting this potential source of confusion. To clarify, dimension-free theoretical guarantees [1,4,5,8] mean our results apply to **continuous state and action spaces without dependence on the cardinality** of these spaces (i.e., $|\mathcal{S}|$ and $|\mathcal{A}|$). As shown by [3], in this setting, new problem dimensions must be considered: **$d_{\Theta}$ (parameterization dimension), $d_{\mathcal{A}}$ (action vector dimension), and $T$ or $(1-\gamma)^{-1}$ (trajectory length)**. While the constants in our assumptions may depend on these dimensions, they do not depend on $|\mathcal{S}|$ or $|\mathcal{A}|$. We will clarify this in the final version of the paper.
>
> **Regularization and convergence rate.** The Reviewer is right. Introducing the regularization term with magnitude $\omega$ affects convergence properties, indeed, Table 1 details complexities w.r.t. $\varepsilon$ and $\omega$. However, the ridge-regularization of the Lagrangian with respect to $\lambda$ makes it **strongly convex (quadratic) in $\lambda$**, allowing last-iterate convergence **without additional assumptions**. In contrast, existing works on CRL with **non-regularized Lagrangians provide convergence guarantees on average (not last-iterate)** [4,5,8].
>
> **References**
> [1] Ding et al. (2024). Last-iterate convergent policy gradient primal-dual methods for constrained mdps.
>
> [2] Khuzani et al. (2016). Distributed regularized primal-dual method: Convergence analysis and trade-offs.
>
> [3] Montenegro et al. (2024). Learning Optimal Deterministic Policies with Stochastic Policy Gradients.
>
> [4] Ding et al. (2020). Natural policy gradient primal-dual method for constrained markov decision processes.
>
> [5] Ding et al. (2022). Convergence and sample complexity of natural policy gradient primal-dual methods for constrained MDPs.
>
> [6] Bai et al. (2023). Achieving zero constraint violation for constrained reinforcement learning via conservative natural policy gradient primal-dual algorithm.
>
> [7] Yang et al. (2020). Global convergence and variance reduction for a class of nonconvex-nonconcave minimax problems.
>
> [8] Liu et al. (2021). Policy optimization for constrained mdps with provable fast global convergence.

---

> > ### Comment · Reviewer_zSqV · 2024-08-13
> >
> > Thank you for your detailed responses. The main contribution of the paper is the derivation of theoretical results that simultaneously offer several advantages (i)-(vi). However, some of these advantages ((i)(ii)(iv)(vi)) have been addressed by previous works. A concern is that assumptions like gradient domination, regularity of the objective function, and the existence of saddle points are quite similar to those in standard (non-convex) optimization, which has been extensively studied in the field of optimization.
> >
> > Overall, I think the theoretical results are solid, but the experiments and comparisons with baselines are relatively simple. As I am not fully familiar with the theory bar of this conference, I will finalize my score after discussing it with the AC. If the paper meets the requirements, I am inclined to vote for acceptance.

---

### Official Review · Reviewer_Vs5i · 2024-07-01

**Soundness:** 3
**Presentation:** 3
**Contribution:** 3
**Rating:** 6
**Confidence:** 4

**Summary:**

The paper studies policy-based methods in constrained RL. The author first establishes the last-iterate convergence of the algorithm C-PG under a form a gradient domination assumptions. Then, the author further designs action-based and parameter-based versions of C-PG to handle constraints defined in terms of risk measures over the costs. Finally, the proposed algorithms are validated by numerical examples of control problems.

**Strengths:**

* Despite not being the first paper to establish the last-iterate convergence of primal-dual type algorithms, I believe the convergence results for C-PG is also a meaningful contribution to the literature.
* More general constrained RL formulations are studied in the paper, where constrains are defined in terms of risk measures. The author adapts C-PG to another two sample-based variants for these formulations.
* Overall, the paper is well-written and easy to follow. The author uses highlights for the two different algorithms/settings, making them easy to distinguish.

**Weaknesses:**

* The role of Section 4 in the paper is unclear to me. Good theoretical results are established for Section 3, yet, there is no general theoretical results for Section 4, besides some cases where results from Section 3 can be directly applied. This makes Section 4 looks like an ''add-on'' to Section 3.
* For theoretical results, e.g., Theorems 3.1 and 3.2, it would be better for the author to briefly discuss the proof idea in the main paper.

Minor comments: it would be better to make figure legends exactly align with the names of algorithms, e.g., change CPGAE to C-PGAE.

**Questions:**

* I hope the author could further clarify the contribution/role of Section 4 in the paper.
* Could the author also discusses how the techniques used in the paper different from that in Ding et al., 2024?

**Limitations:**

The paper has stated assumptions clearly in the paper. Yet, although the author mentions "The limitations of the work emerges in the final section of the work", I failed to find that section in the paper.

---

> ### Author Rebuttal · Authors · 2024-08-04
>
> We thank the Reviewer for having appreciated the contribution our work brings and its presentation. In the following, our answers to the Reviewer's concerns.
>
> **Section 4.** We thank the Reviewer for having raised this point. First of all, **Section 3** introduces the **exploration-agnostic** algorithm C-PG which exhibits global last-iterate convergence **guarantees** (Theorems 3.1 and 3.2). C-PG is thought to be employed for constrained RL problems in which the constraints are enforced in terms of **expected values over cost** functions.
> In **Section 4**, we particularize C-PG in two versions, C-PGAE and C-PGPE, depending on the employed **exploration approach**. Moreover, in Section 4 we **extend** the **framework** presented in Section 3 to handle constraints formulated in terms of **risk measures** over cost functions, employing a **unified formulation** for risk measures. In Table 3 (that we will move to the main paper leveraging the additional page), we show the mapping of the functions $f$ and $g$ (which form the unified risk measure formulation) to several risk measures, in which also the **expectation over costs** is **included**. While we agree that no further theoretical guarantees are presented in Section 4, Remark 4.1 comments on whether the theoretical guarantees of Section 3 applies also for risk measures different from the expected values over costs.
> Besides making the framework more general, the introduction of the constraints over risk measures allow to (empirically) appreciate the **semantic difference** of enforcing constraints while considering different **exploration** approaches (parameter-based vs action-based). In this sense, Section 4 has the goal to highlight the generality of the approach beyond the constraints formulated as expected costs.
>
> **Thr. 3.1 and 3.2.** Thank you for the raising this point. We have deferred the proofs to Appendix E due to space constraints. We will leverage the additional page to add the **sketches of the proofs**.
>
> **Plot labels.** We agree with the Reviewer and we will fix these typos in the final version of the paper.
>
> **Comparison with Ding et al. 2024.** We are happy to clarify this point.
> * From a **technical** point of view, to achieve global last-iterate convergence guarantees, we employ the **$\psi$-gradient domination** (Assumption 3.2) on the Lagrangian function w.r.t. the parameters of the (hyper)policy to be learned, rather than focusing on specific classes of policies or on approximation error assumptions (Assumption 2 of [1]). Moreover, we study the convergence of a different **potential function** $\mathcal{P}_k$ (Section 3.3) and we prove that when it is bounded, then both the objective function gap and the constraint violations are bounded (**Theorem 3.1**). Finally, a new technical challenge we had to face was the study of the recursion appearing in Equation (215). For this, we derived the technical **Lemma F.1** and conducted a quite extensive study of the recurrences of the form $r\_{k+1} \le r\_k - a \\max\\{0,r\_k\\}^\phi + b$ in **Appendix G**.
> * From an **algorithmic** perspective, we propose a *meta algorithm* C-PG that (i) requires a **regularization only on $\lambda$** ([1] performs also entropy regularization w.r.t. the policy), (ii) that is able to handle **multiple constraints** ([1] handles the single constraint case only), (iii) that requires learning rates on **two different time scales** ([1] is single time scale), and (iv) that leverages on an **alternate ascent-descent** scheme. Moreover, [1] proposes also an optimistic method which works only with exact gradients. We particularize C-PG with **parameter-based** and **action-based** exploration approaches ([1] uses just the action-based one) both working with **generic parameterizations** and with **inexact gradients** in **continuous action** and **state** spaces.
>
> **Limitations.** The Reviewer is right, the reported sentence is misleading. We meant that the limitations of our work are to be considered as the points that future works should address (Section 6). For instance, the development of C-PG variants ensuring the same guarantees shown in this paper, but with a single time scale learning rates. We will make more explicit the limitations of our work in the final version of the paper.
>
> **References**
> [1] Ding, D., Wei, C. Y., Zhang, K., & Ribeiro, A. (2024). Last-iterate convergent policy gradient primal-dual methods for constrained mdps. Advances in Neural Information Processing Systems, 36.

---

> > ### Comment · Reviewer_Vs5i · 2024-08-09
> >
> > I thank the reviewer for the explanations. I will maintain my score.

---

### Official Review · Reviewer_ZDP8 · 2024-07-07

**Soundness:** 2
**Presentation:** 3
**Contribution:** 2
**Rating:** 5
**Confidence:** 4

**Summary:**

This paper proposes a general framework for addressing safe RL problems via gradient-based primal-dual algorithms. The authors show that the proposed algorithm exhibit global last-iterate convergence guarantees under gradient domination assumptions. Additionally, the authors validate their algorithms on several constrained control problems.

**Strengths:**

1. This paper is well-written and easy to follow.
2. The paper is technically sound with most claims supported sufficiently.
3. The theoretical analysis seems novel.

**Weaknesses:**

Quality:

1. In Theorem 3.1, it requires w=O(\epsilon) to enforce an overall \epsilon error, but w should be a fixed number in C-PG algorithm.
2. It is better to have some experimental results about cost constraints on MuJoCo and include more baselines, e.g., [Zhang et al., 2020, First Order Constrained Optimization in Policy Space].

Clarity:

1. The range of reward value is [-1, 0], which is a bit weird compared to normal settings.
2. The advantage of parameter-based hyperpolicy is not clearly mentioned in the paper.

Significance:

One relevant literature is missing. In [Liu et al. 2021, Policy Optimization for Constrained MDPs with Provable Fast Global Convergence], a fast convergence \tilde{O}(1/\epsilon) result is proved without multiple assumptions in this paper.

**Questions:**

Please see the details in "weakness".

**Limitations:**

There is no potential negative social impact of this work.

---

> ### Author Rebuttal · Authors · 2024-08-04
>
> We thank the Reviewer for reviewing our work and for appreciating the clarity, the soundness, and the theoretical novelty.
>
> **The role of $\omega$.** The desired accuracy $\varepsilon$ is **decided before** employing the algorithms. As prescribed by theory, $\varepsilon$ appears in the expression of the **learning rates** for the variables to learn and of the **regularization** amount $\omega$. Tuning hyperparameters based on $\varepsilon$ is standard in convergence studies of PGs ([1, thr. C.2] for learning rate and horizon; [2, thr. 6.1] for learning rate) and primal-dual PGs ([3, cor. 3] for learning rate and regularization; [6, thr. 3] for algorithm parameters).
>
> **MuJoCo experiments and baselines.** We present experiments in MuJoCo envs. with costs in Sec. 5 and Apx H, focusing on comparing our algorithms under risk measures over costs. All the presented experiments aim to validate our theoretical results. Indeed, we chose as baselines state-of-the-art PGs for CMDPs [4, 5] coming with convergence guarantees comparable to our results. Specifically, [4] considers tabular CMDPs, while [5] addresses continuous CMDPs but loses last-iterate guarantees. Instead, the paper "Zhang et al., 2020" proposed by the Reviewer solves an approximated constrained optimization problem and its performance is characterized by worst-case policy improvement only and no convergence guarantee. It has a more practical perspective, thus we believe that a comparison with it is beyond the scope of our work.
>
> **Reward range.** This choice is based on formulating the constrained optimization problem for costs, which are positive. Thus, we use a cost $c_0$ in place of the reward $r$ for the optimization part of the problem. The cost function to minimize is defined at line 116 as $c_0(s,a) = - r(s,a)$. We consider $r$ ranging in $[-1,0]$, so that $c_0$ ranges in $[0,1]$. We stress that this choice for the range induces no loss of generality w.r.t. a range $[R_{\min}, R_{\max}]$.
>
> **Hyperpolicy advantage.** We are happy to clarify this point.
> 1. **Trade-off between exploration approaches**. There is a trade-off between PB and AB explorations [2,5]. The PB approach struggles with **large parameterizations** (large $d_{\Theta}$), but its gradient estimator has **lower variance** compared to AB. The AB approach struggles with **long-lasting interactions** (large $T$ or $\gamma \to 1$) and **large action vectors** (large $d_{\mathcal{A}}$), but comes with a **higher variance** gradient estimator than PB. Thus, PB exploration can be preferable in some cases.
> 2. **PB exploration and risk constraints**: In Sec. 5, we show that PB exploration, when considering risk constraints over costs, can learn policies **inducing safer behaviors compared to AB exploration**. This arises from the semantic differences in how the two approaches enforce risk constraints (lines 288-293).
>
> Finally, we note that PB exploration has not been addressed in the CRL literature (to our knowledge). We thank the Reviewer for pointing out this lack of clarity, will make it more explicit in the final version of the paper.
>
> **Comparison with Liu et al. 2021.** The paper proposed by the Reviewer considers mainly the setting of: (i) *tabular* CMDPs; (ii) *softmax* policies; (iii) *exact gradients* provided by an oracle; and (iv) provides an *average* (not last-iterate) convergence rate of order $\tilde{\mathcal{O}}(\epsilon^{-1})$. As shown in Table 1, in the same setting, our convergence rate is $\mathcal{O}(\varepsilon^{-2})$, i.e., when: (1) GD condition holds $\psi = 1$ (which happens under (i) *tabular* CMDPs; (ii) *softmax* policies; see Remark 3.1); (2) exact gradients are available; (3) the regularization $\omega$ is $\mathcal{O}(\varepsilon)$. Liu et al. 2021 faster convergence is justified by the fact that their approach **relies to the softmax policy model**, indeed, **applies to tabular problems only**, and **does not reach a last-iterate convergence**.
> Moreover, Liu et al. 2021 propose a sample-based variant which requires $\tilde{\mathcal{O}}(\varepsilon^{-3})$ (still for the tabular case and with an average convergence guarantee). As shown in Table 1, in such case (i.e., $\psi=1$ and inexact gradients) our last-iterate rate is $\tilde{\mathcal{O}}(\omega^{-4}\varepsilon^{-3})$ (being $\tilde{\mathcal{O}}(\varepsilon^{-7})$ with $\omega=\mathcal{O}(\varepsilon)$). However, the setting considered by Liu et al. 2021 assumes the access to a **generative model generating independent trajectories from any arbitrary pair $(s,a)$**, a demanding requirement that we overcome.
> Finally, we stress that in the setting considered by Liu et al. 2021, most of our assumptions hold. Indeed, asm. 3.1 holds in the form of the Slater's condition; asm. 3.2 holds with $\psi=1$ for tabular CMDPs with softmax policies (see Remark 3.1); asm. 3.3 holds for tabular softmax that is known to lead to bounded gradients and smooth objectives [7]; asm. 3.4 holds because the mentioned paper considers exact gradients or it uses good events considering the differences between the estimations and the real values of $V$ and $Q$ bounded (their Definition 3). We thank the Reviewer for suggesting this paper; we will add it in the final version of the paper, especially in the comparison table (Table 2).
>
> **References**
> [1] Yuan et al. (2022). A general sample complexity analysis of vanilla policy gradient.
>
> [2] Montenegro et al. (2024). Learning Optimal Deterministic Policies with Stochastic Policy Gradients.
>
> [3] Ding et al (2024). Last-iterate convergent policy gradient primal-dual methods for constrained mdps.
>
> [4] Ding et al. (2022). Convergence and sample complexity of natural policy gradient primal-dual methods for constrained MDPs.
>
> [5] Metelli et al. (2018). Policy optimization via importance sampling.
>
> [6] Liu et al. (2021). Policy optimization for constrained mdps with provable fast global convergence.
>
> [7] Papini et al. (2022). Smoothing policies and safe policy gradients.

---

> > ### Comment · Reviewer_ZDP8 · 2024-08-13
> >
> > Thank you for your clarification! I have increased the score.

---

### Official Review · Reviewer_AKt3 · 2024-07-10

**Soundness:** 3
**Presentation:** 3
**Contribution:** 3
**Rating:** 7
**Confidence:** 1

**Summary:**

This paper studies the problem of constrained MDP. To solve this problem, this paper adopts the policy gradient methods, and specifically they considered the action-based policy gradient method and parameter-based policy gradient method.

The algorithm proposed in this paper is a type of primal-dual method. Under certain assumptions, this algorithm is shown to have last iterate convergence.

This paper also executes numerical experiments on various environment with their algorithm, and the experiments validate the results in this paper.

**Strengths:**

This paper is a well-written paper. The description of problem setup, theorems, assumptions are clear.

Even though this is mainly a theory paper, there are numerical experiments which validates the theoretical results.

This paper has results on the last iterate convergence, which is a property not possessed by most of stochastic optimization algorithms.

**Weaknesses:**

I don't see significant weaknesses in this paper.

**Questions:**

Do you have lower bounds showing that these rates are tight?

**Limitations:**

Yes. The authors addressed all the limitations listed in the guidelines.

---

> ### Author Rebuttal · Authors · 2024-08-04
>
> We thank the Reviewer for appreciating our work and recognizing its novelty. Below, we respond to the Reviewer's questions.
>
> > Do you have lower bounds showing that these rates are tight?
>
> The derivation of a lower bound for Constrained MDPs with **continuous state and/or action spaces** is still an **open research problem**. Nevertheless, [1] presents a sample complexity lower bound for the *tabular case* under a generative model, which is of order ${\Omega}(|\mathcal{S}| |\mathcal{A}| (1-\gamma)^{-5} \zeta^{-2} \varepsilon^{-2})$, where $\zeta$ is the Slater's constant. However, such a lower bound does not apply to our case, since it has been derived for the tabular setting.
>
> **Reference**
> [1] Vaswani, S., Yang, L., & Szepesvári, C. (2022). Near-optimal sample complexity bounds for constrained MDPs. Advances in Neural Information Processing Systems, 35, 3110-3122.

---

> > ### Comment · Reviewer_AKt3 · 2024-08-12
> >
> > Thank you very much for your response. I do not have further questions.

---

### Decision · Program_Chairs · 2024-09-25

**Decision:**

Accept (poster)

**Comment:**

The paper makes a significant contribution to the field of constrained RL by establishing the last-iterate convergence of C-PG, a result that strengthens the theoretical foundation of policy-based methods in this domain. Furthermore, the authors extend their analysis to encompass more general constrained RL formulations, demonstrating the adaptability of their approach. Authors have done a amazing work presenting the paper in a clear and organized way, making the complex ideas accessible to readers.

However, as reviewers pointed out, the theoretical contribution of Section 4 will benefit from additional revisions, with the authors enhancing the paper by providing a more thorough discussion of the proof techniques employed. While there is potential for further refinement, the paper's merits outweigh its shortcomings, justifying an acceptance decision. The reviewers anticipate that the authors will address the constructive feedback provided, leading to a stronger final version of the paper.